# Causality Meets Locality: Provably Generalizable and Scalable Policy Learning for Networked Systems

**Hao Liang[1,†], Shuqing Shi[1,*], Yudi Zhang[2], Biwei Huang[3], Yali Du[1,4]**

[1]King's College London    [2]Eindhoven University of Technology
[3]University of California, San Diego    [4]The Alan Turing Institute

## Abstract

Large-scale networked systems, such as traffic, power, and wireless grids, challenge reinforcement-learning agents with both scale and environment shifts. To address these challenges, we propose GSAC (**G**eneralizable and **S**calable **A**ctor-**C**ritic), a framework that couples causal representation learning with meta actor-critic learning to achieve both scalability and domain generalization. Each agent first learns a sparse local causal mask that provably identifies the minimal neighborhood variables influencing its dynamics, yielding exponentially tight approximately compact representations (ACRs) of state and domain factors. These ACRs bound the error of truncating value functions to $\kappa$-hop neighborhoods, enabling efficient learning on graphs. A meta actor-critic then trains a shared policy across multiple source domains while conditioning on the compact domain factors; at test time, a few trajectories suffice to estimate the new domain factor and deploy the adapted policy. We establish finite-sample guarantees on causal recovery, actor-critic convergence, and adaptation gap, and show that GSAC adapts rapidly and significantly outperforms learning-from-scratch and conventional adaptation baselines.

## 1 Introduction

Large-scale networked systems, such as traffic networks [41], power grids [3], and wireless communication systems [1, 44], present significant challenges for reinforcement learning (RL) due to their massive scale, sparse local interactions, and structural heterogeneity. These characteristics impose two fundamental difficulties: scalability and generalizability. On one hand, the joint state-action space grows exponentially with the number of agents, making conventional RL algorithms [29] computationally intractable. On the other hand, real-world networked systems often experience environment shifts and structural changes, necessitating algorithms that can generalize and adapt efficiently across different environments. Therefore, a natural question arises:

*Is it feasible to design a provably generalizable and scalable MARL algorithm for networked system?*

While this question has attracted increasing attention in single-agent RL domain generalization literature [39, 10, 6], its resolution remains open in the multi-agent reinforcement learning (MARL) context, especially for learning in networked system. In this work, we provide an affirmative answer by developing a causality-inspired framework, GSAC (**G**eneralizable and **S**calable **A**ctor-**C**ritic), which couples causal representation learning with meta actor-critic learning to achieve both scalability and generalizability. We summarize our main contributions as follows.

- We establish *structural identifiability* in networked MARL, providing the first sample complexity results of causal mask recovery and domain factor estimation.

---

[*]Equal contribution.
[†]Correspondence to: Hao Liang (`hao.liang@kcl.ac.uk`).

39th Conference on Neural Information Processing Systems (NeurIPS 2025).

- We introduce efficient algorithms to construct *approximately compact representations* (ACRs) of states and domain factors. ACR improves both scalability, by significantly reducing the input dimensionality required for learning and computation, and generalizability, by isolating the minimal and most informative components. This approach may be of independent interest.

- We propose a meta actor-critic algorithm, which jointly learns scalable localized policies across multiple source domains, conditioning on the compact domain factors to generalize effectively.

- We provide rigorous theoretical guarantees on finite-sample convergence and adaptation gap, and empirically validate our method on two benchmarks demonstrating rapid adaptation and superior performance over learning-from-scratch and conventional adaptation baselines.

## 1.1 Related works

**Networked MARL with guarantees.** Our work is closely related to the line of research on *MARL in networked systems* [23, 17, 24]. These studies leverage local interactions to enable scalability, proposing decentralized policy optimization algorithms that learn local policies for each agent with finite-time convergence guarantees. In particular, [24] provides the first provably efficient MARL framework for networked systems under the discounted reward setting. The works in [23] and [17] extend this framework to the average-reward setting and to stochastic and non-local network structures, respectively. However, to the best of our knowledge, no existing methodology addresses both the design and theoretical analysis of policies that are simultaneously generalizable across domains and scalable in networked MARL. Our work fills this gap by introducing a principled framework that achieves provable generalization and scalability via causal representation learning and domain-conditioned policy optimization.

**Domain generalization and adaptation in RL.** RL agents often encounter *environmental shifts* between training and deployment, prompting recent efforts to improve generalization and adaptation. [10] proposes learning factored representations along with individual change factors across domains, while [7] extends this approach to handle non-stationary environments. [22, 33] aim to enhance generalization to unseen states by eliminating redundant dependencies between state and action variables in causal dynamics models. Beyond causal approaches, a substantial body of work focuses on learning domain-invariant representations without causal modeling, such as bisimulation metrics [39] that preserve decision-relevant structure while filtering out nuisance features. Causal representation learning has also been applied to model goal-conditioned transitions [6], offering theoretical guarantees via structure-aware meta-learning [10], and credit assignment [42, 32]. However, these works focus solely on single-agent settings or multi-agent settings with a limited number of agents. In contrast, we provide the first sample complexity guarantees for structural identifiability in networked MARL and introduce ACRs, a novel mechanism that enables both scalable learning and provable generalization across domains.

A detailed discussion of related work is provided in Appendix A.

## 2 Preliminaries

For clarity, we provide a complete table of notation in Appendix B.

**Networked MARL.** We consider networked MARL represented as a graph $\mathscr{G} = (\mathscr{N}, \mathscr{E})$, where $\mathscr{N} = 1, \ldots, n$ denotes the set of agents and $\mathscr{E} \subseteq \mathscr{N} \times \mathscr{N}$ encodes the interaction edges. Each agent $i \in \mathscr{N}$ observes a local state $\mathbf{s}_i \in \mathcal{S}_i$ and selects an action $\mathbf{a}_i \in \mathcal{A}_i$, forming the global state $\mathbf{s} = (\mathbf{s}_1, \ldots, \mathbf{s}_n) \in \mathcal{S} := \mathcal{S}_1 \times \cdots \times \mathcal{S}_n$ and the joint action $\mathbf{a} = (\mathbf{a}_1, \ldots, \mathbf{a}_n) \in \mathcal{A} := \mathcal{A}_1 \times \cdots \times \mathcal{A}_n$. At each time step $t$, the system evolves according to the following decentralized transition dynamics:

$$P(\mathbf{s}(t+1) \mid \mathbf{s}(t), \mathbf{a}(t)) = \prod_{i=1}^{n} P_i\big(\mathbf{s}_i(t+1) \mid \mathbf{s}_{\mathcal{N}_i}(t), \mathbf{a}_i(t)\big), \tag{1}$$

where $\mathcal{N}_i \subseteq \mathscr{N}$ denotes the set of neighbors of agent $i$ in the interaction graph, including $i$ itself, and $\mathbf{s}_{\mathcal{N}_i}(t)$ collects the states of agents in $\mathcal{N}_i$ at time $t$. Each agent $i$ adopts a localized policy $\pi_i^{\theta_i}$, parameterized by $\theta_i \in \Theta_i$, which specifies a distribution over local actions conditioned on its local neighborhood state: $\pi_i(\mathbf{a}_i \mid \mathbf{s}_{\mathcal{N}_i})$. Agents act independently according to their respective policies. We write $\theta := (\theta_1, \ldots, \theta_n)$ to denote the tuple of all local policy parameters, and define the joint

policy as $\pi^\theta(\mathbf{a} \mid \mathbf{s}) := \prod_{i=1}^{n} \pi_i^{\theta_i}(\mathbf{a}_i \mid \mathbf{s}_{\mathcal{N}_i})$. We only consider localized policy and use $\theta$ and $\pi$ interchangeably throughout the paper when referring to policies.

Each agent receives a local reward $r_i(\mathbf{s}_i, \mathbf{a}_i)$ depending on its local state and action, and the global reward is defined as the average across all agents: $r(\mathbf{s}, \mathbf{a}) := \frac{1}{n} \sum_{i=1}^{n} r_i(\mathbf{s}_i, \mathbf{a}_i)$. The goal is to learn a set of localized policies $\theta$ that maximize the expected discounted sum of global rewards, starting from an initial state distribution $\rho_0$:

$$\max_{\theta \in \Theta} J(\theta) := \mathbb{E}_{\mathbf{s} \sim \rho_0} \mathbb{E}_{\mathbf{a}(t) \sim \pi^\theta(\cdot | \mathbf{s}(t)), \, \mathbf{s}(t+1) \sim P(\cdot | \mathbf{s}(t), \mathbf{a}(t))} \left[ \sum_{t=0}^{\infty} \gamma^t r(\mathbf{s}(t), \mathbf{a}(t)) \Big| \mathbf{s}(0) = \mathbf{s} \right]. \quad (2)$$

For clarity, we assume the reward function is known; however, our framework and analysis readily extend to the setting with unknown rewards.

**Truncation as efficient approximation in networked MARL**   A central challenge in applying reinforcement learning to networked systems is the curse of dimensionality: while each agent's local state and action spaces are relatively small, the global state and action spaces grow exponentially with the number of agents $n$. This renders standard RL methods computationally intractable at scale. [24] exploit the local interaction structure and demonstrate that the $Q$-function exhibits exponential decay with respect to graph distance. This property enables a principled truncation of the $Q$-function for scalable approximation. Specifically, for a state-action pair $(\mathbf{s}, \mathbf{a})$ and a policy $\pi$:

$$Q^\pi(\mathbf{s}, \mathbf{a}) := \mathbb{E}_{\mathbf{a}(t) \sim \pi^\theta(\cdot | \mathbf{s}(t))} \left[ \sum_{t=0}^{\infty} \gamma^t r(\mathbf{s}(t), \mathbf{a}(t)) \Big| \mathbf{s}(0) = \mathbf{s}, \, \mathbf{a}(0) = \mathbf{a} \right]$$

$$= \frac{1}{n} \sum_{i=1}^{n} \mathbb{E}_{\mathbf{a}(t) \sim \pi^\theta(\cdot | \mathbf{s}(t))} \left[ \sum_{t=0}^{\infty} \gamma^t r_i(\mathbf{s}_i(t), \mathbf{a}_i(t)) \Big| \mathbf{s}(0) = \mathbf{s}, \, \mathbf{a}(0) = \mathbf{a} \right] := \frac{1}{n} \sum_{i=1}^{n} Q_i^\pi(\mathbf{s}, \mathbf{a}). \quad (3)$$

For an integer $\kappa \geq 0$, let $\mathcal{N}_i^\kappa$ denote the $\kappa$-hop neighborhood of agent $i$, and define $\mathcal{N}_{-i}^\kappa := \mathcal{N} \backslash \mathcal{N}_i^\kappa$ as the set of agents outside this neighborhood. We write the global state and action as $\mathbf{s} = (\mathbf{s}_{\mathcal{N}_i^\kappa}, \mathbf{s}_{\mathcal{N}_{-i}^\kappa})$ and $\mathbf{a} = (\mathbf{a}_{\mathcal{N}_i^\kappa}, \mathbf{a}_{\mathcal{N}_{-i}^\kappa})$, respectively. [24] show that for any $\pi$, agent $i \in \mathcal{N}$, and any tuples $\mathbf{s}_{\mathcal{N}_i^\kappa}, \mathbf{s}_{\mathcal{N}_{-i}^\kappa}, \mathbf{s}_{\mathcal{N}_{-i}^\kappa}', \mathbf{a}_{\mathcal{N}_i^\kappa}, \mathbf{a}_{\mathcal{N}_{-i}^\kappa}, \mathbf{a}_{\mathcal{N}_{-i}^\kappa}', Q_i^\pi$ satisfies the exponential decay property:

$$\left| Q_i^\pi(\mathbf{s}_{\mathcal{N}_i^\kappa}, \mathbf{s}_{\mathcal{N}_{-i}^\kappa}, \mathbf{a}_{\mathcal{N}_i^\kappa}, \mathbf{a}_{\mathcal{N}_{-i}^\kappa}) - Q_i^\pi(\mathbf{s}_{\mathcal{N}_i^\kappa}, \mathbf{s}_{\mathcal{N}_{-i}^\kappa}', \mathbf{a}_{\mathcal{N}_i^\kappa}, \mathbf{a}_{\mathcal{N}_{-i}^\kappa}') \right| \leq \frac{\bar{r}}{1 - \gamma} \gamma^{\kappa+1}, \quad (4)$$

where $\bar{r}$ is the upper bound on the reward functions. Thus the influence of distant agents on $Q_i^\pi$ diminishes rapidly with graph distance. This motivates the use of a truncated $Q$-function:

$$\hat{Q}_i^\pi \left( s_{\mathcal{N}_i^\kappa}, \mathbf{a}_{\mathcal{N}_i^\kappa} \right) := Q_i^\pi \left( \mathbf{s}_{\mathcal{N}_i^\kappa}, \bar{\mathbf{s}}_{\mathcal{N}_{-i}^\kappa}, \mathbf{a}_{\mathcal{N}_i^\kappa}, \bar{\mathbf{a}}_{\mathcal{N}_{-i}^\kappa} \right), \quad (5)$$

where $\bar{\mathbf{s}}_{\mathcal{N}_{-i}^\kappa}$ and $\bar{\mathbf{a}}_{\mathcal{N}_{-i}^\kappa}$ are fixed (and arbitrary) placeholders for the unobserved components. Due to exponential decay, the truncated $Q$-function approximates the true $Q$-function with bounded error: $\sup_{(\mathbf{s}, \mathbf{a}) \in \mathcal{S} \times \mathcal{A}} \left| \hat{Q}_i^\pi \left( s_{\mathcal{N}_i^\kappa}, a_{\mathcal{N}_i^\kappa} \right) - Q_i^\pi(\mathbf{s}, \mathbf{a}) \right| \leq \frac{\bar{r}}{1-\gamma} \gamma^{\kappa+1}$. Furthermore, this truncated approximation can be directly used in actor-critic updates, enabling scalable policy gradient estimation. The approximation error in the truncated policy gradient is also exponentially small[3]. Crucially, the truncation technique has significantly reduced input dimensionality, making it much more efficient to compute, store, and optimize in large-scale networked settings.

**Networked MARL under domain generalization**   The standard formulation of networked MARL, as reviewed above, assumes a fixed environment or domain. We extend this framework to incorporate domain generalization, where the environment may shift between training and deployment. Each environment $\mathcal{E}(\boldsymbol{\omega})$ is characterized by its own transition dynamics (see Equation 6), and is uniquely specified by a latent domain factor $\boldsymbol{\omega} = (\omega_i)_{i \in \mathcal{N}} \in \Omega$, where $\omega_i$ encodes the domain-specific dynamics for agent $i$. To model structured variation across environments, we define the local generative process for each agent $i$. Let $\mathbf{s}_i = (s_{i,1}, \ldots, s_{i, d_i^s})$, where $d_i^s$ is its dimensionality. The state and reward for agent $i$ evolve as:

$$s_{i,j}(t+1) = f_{i,j} \left( \mathbf{c}_{\mathcal{N}_{i,j}}^{\mathbf{s} \to \mathbf{s}} \odot \mathbf{s}_{\mathcal{N}_i}(t), \mathbf{c}_{i,j}^{\mathbf{a} \to \mathbf{s}} \odot \mathbf{a}_i(t), \mathbf{c}_{i,j}^{\boldsymbol{\omega} \to \mathbf{s}} \odot \omega_i, \epsilon_{i,j}^{\mathbf{s}}(t) \right), \quad (6)$$

---

[3]See Appendix C for a detailed review of key concepts in networked MARL.

where $\odot$ denotes element-wise multiplication, and $\epsilon$ denotes i.i.d. stochastic noise. The binary vectors $\mathbf{c}^{\cdot\rightarrow\cdot}$ are causal masks, indicating structural dependencies between variables. Crucially, the functions $f_{i,j}$ and the causal masks $\mathbf{c}^{\cdot\rightarrow\cdot}$ are shared across all environments, and only $\boldsymbol{\omega}_i$ varies. This decomposition allows us to disentangle invariant causal structure from domain-specific variations.

We consider a training setup with $M$ source domains $\langle \mathcal{E}_1, \dots, \mathcal{E}_M \rangle$ and a target domain $\mathcal{E}_{M+1}$, each drawn i.i.d. from an unknown domain distribution $\boldsymbol{\omega}_m \sim \mathcal{D}$ for $m = 1, \cdots, M+1$. The goal is to learn networked MARL policies from the source domains that generalize effectively to new, unseen target domains, using as little adaptation data as possible.

## 3 Approximately compact representation

In networked MARL, each agent's truncated value function and localized policy are given by $\hat{Q}_i^\pi : \mathcal{S}_{\mathcal{N}_i^\kappa} \times \mathcal{A}_{\mathcal{N}_i^\kappa} \to \mathbb{R}$ and $\pi_i : \mathcal{S}_{\mathcal{N}_i} \to \Delta(\mathcal{A})$, respectively. The input dimensionality of the value function is $\dim(\mathcal{S}_{\mathcal{N}_i^\kappa} \times \mathcal{A}_{\mathcal{N}_i^\kappa}) = \sum_{j \in \mathcal{N}_i^\kappa}(d_j^{\mathbf{s}} + d_i^{\mathbf{a}})$, and that of the policy is $\dim(\mathcal{S}_{\mathcal{N}_i}) = \sum_{j \in \mathcal{N}_i} d_j^{\mathbf{s}}$. While truncation to the $\kappa$-hop neighborhood significantly reduces the input size compared to the global space $\dim(\mathcal{S} \times \mathcal{A}) = \sum_{j \in \mathcal{N}}(d_j^{\mathbf{s}} + d_j^{\mathbf{a}})$, the input dimensionality can still be large, especially when $\kappa$ or node degrees are high. To further mitigate this issue, we propose constructing ACRs by leveraging the identifiable causal masks $\mathbf{c}_i$ for each agent $i$, as defined in Equation 6. Specifically, we use the causal structure to extract a reduced set of relevant variables from $\mathbf{s}_{\mathcal{N}_i^\kappa}$, denoted $\mathbf{s}_{\mathcal{N}_i^\kappa}^\circ \subset \mathbf{s}_{\mathcal{N}_i^\kappa}$, which preserves the predictive information for the value function. This leads to a strictly smaller input space $\dim(\mathbf{s}_{\mathcal{N}_i^\kappa}^\circ) < \dim(\mathbf{s}_{\mathcal{N}_i^\kappa})$, with approximation error that decays exponentially in $\kappa$.

Moreover, we extend this ACR framework to include the policy inputs and domain-specific factors, enabling efficient transfer across domains. The resulting ACR framework improves both scalability, by significantly reducing the input dimensionality required for learning and computation, and generalizability, by isolating the minimal and most informative domain-specific components relevant to each agent. In the following subsections, we detail the algorithmic construction of ACRs, beginning with the fixed-environment setting. For clarity, we assume the causal masks and domain factors are given; their identification and the estimation of domain factors are addressed in Section 5.

### 3.1 ACR for fixed environment

We begin by formalizing the notion of an ACR for truncated $Q$-functions.

**Definition 1** (Value ACR). Given the graph $\mathscr{G}$ encoded by the binary masks $\mathbf{c}$, for each agent $i$ and its $\kappa$-hop neighborhood $\mathcal{N}_i^\kappa$, we recursively define the $\kappa$-hop ACR $\mathbf{s}_{\mathcal{N}_i^\kappa}^\circ(t)$ as the set of state components $s_{\mathcal{N}_i^\kappa,j}$ satisfying at least one of the following:

- has a direct link to agent $i$'s reward $r_i$, i.e., $c_{i,j}^{s \rightarrow r} = 1$, and $\mathbf{s}_{\mathcal{N}_i^0}^\circ := \cup_{j:c_{i,j}^{s \rightarrow r}=1} s_{i,j}$.

- for $1 \le \kappa' \le \kappa$: $s_{\mathcal{N}_i^{\kappa'},j}(t)$ has a link to $s_{\mathcal{N}_i^{\kappa'-1},l}(t+1)$ such that $s_{\mathcal{N}_i^{\kappa'-1},l}(t) \in \mathbf{s}_{\mathcal{N}_i^{\kappa'-1}}^\circ$.

Figure 1 illustrates how the proposed ACR framework identifies the key state components influencing an agent's reward through local causal structure. At each time step, the agent's reward $r_i$ depends directly on a small subset of neighboring state variables at the previous time step . These relevant components are recursively traced backward along causal edges, yielding a compact representation $s_{\mathcal{N}_i^\kappa}^\circ$ within the $\kappa$-hop neighborhood. The figure also visualizes how the cumulative discounted rewards $r_i(t), \gamma r_i(t+1), \dots, \gamma^\kappa r_i(t+\kappa)$ are aggregated, emphasizing that only the identified influential components contribute significantly to the value function. This provides the structural foundation for the exponentially tight approximation guarantees in Proposition 1. Based on this, we define the approximately compact $Q$-function as:

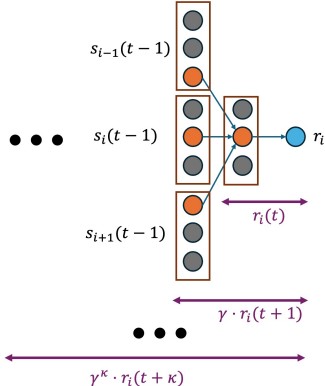

Figure 1: Illustration of ACR.

$$\tilde{Q}_i^\pi\left(\mathbf{s}_{\mathcal{N}_i^\kappa}^\circ, \mathbf{a}_{\mathcal{N}_i^\kappa}\right) := \hat{Q}_i^\pi\left(\mathbf{s}_{\mathcal{N}_i^\kappa}^\circ, \bar{\mathbf{s}}_{\mathcal{N}_i^\kappa}/\mathbf{s}_{\mathcal{N}_i^\kappa}^\circ, \mathbf{a}_{\mathcal{N}_i^\kappa}\right), \qquad (7)$$

where $\bar{s}_{\mathcal{N}_i^\kappa}/\mathbf{s}_{\mathcal{N}_i^\kappa}^\circ$ denotes fixed, arbitrary values for the irrelevant components. The approximation error of $\tilde{Q}_i^\pi$ compared to the full $Q$-function is exponentially small in $\kappa$, as shown below.

**Proposition 1** (Approximation error of value ACR). For any agent $i$, policy $\pi$, and $\kappa \geq 0$, the approximation error between $\tilde{Q}_i^\pi$ and $Q_i^\pi$ satisfies:

$$\sup_{(\mathbf{s},\mathbf{a}) \in \mathcal{S} \times \mathcal{A}} \left| \tilde{Q}_i^\pi \left( \mathbf{s}_{\mathcal{N}_i^\kappa}^\circ, \mathbf{a}_{\mathcal{N}_i^\kappa} \right) - Q_i^\pi(\mathbf{s},\mathbf{a}) \right| \leq \frac{2\bar{r}}{1-\gamma} \gamma^{\kappa+1}.$$

The proof is deferred to Appendix E. We now define ACRs for policies.

**Definition 2** (Policy ACR). Given the graph $\mathscr{G}$ encoded by binary causal masks $\mathbf{c}$, the ACR for agent $i$'s policy over its neighborhood $\mathcal{N}_i$ is defined recursively using Algorithm 3 (see Appendix D).

Algorithm 3 takes as input the causal masks $\mathbf{c}_{\mathcal{N}_i^\kappa}$ and the local states $\mathbf{s}_{\mathcal{N}_i}$, and outputs a compact representation $\mathbf{s}_{\mathcal{N}_i}^\circ \subset \mathbf{s}_{\mathcal{N}_i}$ such that $\dim(\mathbf{s}_{\mathcal{N}_i}^\circ) < \dim(\mathbf{s}_{\mathcal{N}_i})$. We then define the *approximately compact policy* $\tilde{\pi}_i : \mathcal{S}_{\mathcal{N}_i}^\circ \mapsto \Delta_{\mathcal{A}_i}$ by $\tilde{\pi}_i \left( \cdot \mid \mathbf{s}_{\mathcal{N}_i}^\circ \right) := \pi_i \left( \cdot \mid \mathbf{s}_{\mathcal{N}_i}^\circ, \bar{s}_{\mathcal{N}_i}/\mathbf{s}_{\mathcal{N}_i}^\circ \right)$, where $\bar{s}_{\mathcal{N}_i}/\mathbf{s}_{\mathcal{N}_i}^\circ$ are fixed values for the non-influential components. By combining the value and policy ACRs, we define the doubly compact $Q$-function $\tilde{Q}_i^{\tilde{\pi}} \left( \mathbf{s}_{\mathcal{N}_i^\kappa}^\circ, \mathbf{a}_{\mathcal{N}_i^\kappa} \right)$, which reduces input complexity while retaining performance guarantees.

**Proposition 2** (Approximation error). For any policy $\pi$, let $\tilde{\pi}$ be the corresponding approximately compact policy constructed using Algorithm 3, for any $i$, and $\kappa \geq 0$, we have

$$\sup_{(\mathbf{s},\mathbf{a}) \in \mathcal{S} \times \mathcal{A}} \left| \tilde{Q}_i^{\tilde{\pi}} \left( \mathbf{s}_{\mathcal{N}_i^\kappa}^\circ, \mathbf{a}_{\mathcal{N}_i^\kappa} \right) - Q_i^\pi(\mathbf{s},\mathbf{a}) \right| \leq 3 \frac{\bar{r}}{1-\gamma} \gamma^{\kappa+1}.$$

The proof of Proposition 2 is deferred to Appendix E. These results show that identifying ACRs enables dramatic reductions in the state-space dimensionality for each agent, making learning and computation tractable even in large-scale networks. In particular, the effective input dimension satisfies $|\mathbf{s}_{\mathcal{N}_i^\kappa}^\circ| \ll |\mathbf{s}_{\mathcal{N}_i^\kappa}|$. In the next subsection, we extend ACR construction to domain factors, enabling generalization across multiple environments.

## 3.2 ACR for efficient domain generalization

We now extend the ACR framework to support generalization across environments characterized by latent domain-specific factors. Given a policy $\pi$ and environment $\boldsymbol{\omega} = (\boldsymbol{\omega}_i)_{i \in \mathcal{N}}$, the *environment-conditioned Q-function* is defined as

$$Q_i^\pi(\mathbf{s},\mathbf{a},\boldsymbol{\omega}) := \mathbb{E}_{\mathbf{a}(t) \sim \pi(\cdot|\mathbf{s}(t)), \mathbf{s}(t+1) \sim P_{\boldsymbol{\omega}}(\cdot|\mathbf{s}(t),\mathbf{a}(t))} \left[ \sum_{t=0}^{\infty} \gamma^t r_i(\mathbf{s}_i(t),\mathbf{a}_i(t)) \mid \mathbf{s}(0) = \mathbf{s}, \mathbf{a}(0) = \mathbf{a} \right].$$

We show (Lemma 7) that this $\boldsymbol{\omega}$-conditioned $Q$-function retains the exponential decay property. This motivates defining a truncated $\boldsymbol{\omega}$-conditioned $Q$-function by fixing irrelevant variables:

$$\hat{Q}_i^\pi \left( \mathbf{s}_{\mathcal{N}_i^\kappa}, \mathbf{a}_{\mathcal{N}_i^\kappa}, \boldsymbol{\omega}_{\mathcal{N}_i^\kappa} \right) := Q_i^\pi \left( \mathbf{s}_{\mathcal{N}_i^\kappa}, \bar{\mathbf{s}}_{\mathcal{N}_{-i}^\kappa}, \mathbf{a}_{\mathcal{N}_i^\kappa}, \bar{\mathbf{a}}_{\mathcal{N}_{-i}^\kappa}, \boldsymbol{\omega}_{\mathcal{N}_i^\kappa}, \bar{\boldsymbol{\omega}}_{\mathcal{N}_{-i}^\kappa} \right). \tag{8}$$

We consider $\boldsymbol{\omega}$-conditioned policies $\pi = (\pi_1, \ldots, \pi_n)$ such that each $\pi_i : \mathcal{S}_{\mathcal{N}_i} \times \Omega_{\mathcal{N}_i} \mapsto \Delta_{\mathcal{A}_i}$, i.e., $\mathbf{a}_i(t) \sim \pi_i(\cdot|\mathbf{s}_{\mathcal{N}_i}(t), \boldsymbol{\omega}_{\mathcal{N}_i})$. To reduce the input space further, we define a domain-factor ACR $\boldsymbol{\omega}^\circ$ (Definition 5), which identifies the minimal subset of $\boldsymbol{\omega}_{\mathcal{N}_i^\kappa}$ influencing the reward. Using both the value ACR and domain ACR, we define the approximately compact $\boldsymbol{\omega}$-conditioned $Q$-function $\tilde{Q}_i^\pi \left( \mathbf{s}_{\mathcal{N}_i^\kappa}^\circ, \mathbf{a}_{\mathcal{N}_i^\kappa}, \boldsymbol{\omega}_{\mathcal{N}_i^\kappa}^\circ \right)$ (Definition 6). Similarly, we define policy ACRs for $\boldsymbol{\omega}$-conditioned policies (Definition 7), and let $\tilde{\pi}$ denote the corresponding approximately compact policy. We then obtain a fully compact representation $\tilde{Q}_i^{\tilde{\pi}}$ of $Q$-function, which provably approximates the true $Q$-function.

**Proposition 3.** For any policy $\pi$, and any $i$, we have

$$\sup_{(\mathbf{s},\mathbf{a},\boldsymbol{\omega}) \in \mathcal{S} \times \mathcal{A} \times \Omega} \left| \tilde{Q}_i^{\tilde{\pi}} \left( \mathbf{s}_{\mathcal{N}_i^\kappa}^\circ, \mathbf{a}_{\mathcal{N}_i^\kappa}, \boldsymbol{\omega}_{\mathcal{N}_i^\kappa}^\circ \right) - Q_i^\pi(\mathbf{s},\mathbf{a},\boldsymbol{\omega}) \right| \leq \frac{3\bar{r}}{1-\gamma} \gamma^{\kappa+1}. \tag{9}$$

The proof of Proposition 3 is deferred to Appendix E. Proposition 3 shows that it suffices to operate on the key components of the $\kappa$-hop state and latent domain factors, which substantially reduces input dimensionality. As a result, both training and test-time inference become significantly more scalable—without sacrificing theoretical guarantees. In the next section, we present our main algorithm, which leverages these ACRs to achieve generalizable and scalable networked policy learning. For simplicity, we will omit "approximately compact" when referring to the functions and policies $Q$, knowing that all components operate on the identified ACRs.

# 4 Generalizable and scalable actor-critic

## 4.1 Roadmap

We propose GSAC (**G**eneralizable and **S**calable **A**ctor-**C**ritic), a principled framework for scalable and generalizable networked MARL. Our GSAC framework (Algorithm 1) integrates causal discovery, representation learning, and meta actor-critic optimization into a unified pipeline, with each phase supported by theoretical results. Phase 1 (causal discovery and domain factor estimation) is underpinned by Theorem 1 and Propositions 4-5, which establish structural identifiability and sample complexity guarantees for recovering causal masks and latent domain factors. Phase 2 (construction of ACRs) leverages the causal structure to build compact representations of value functions and policies, with bounded approximation errors rigorously characterized by Propositions 1-3. Phase 3 (meta actor-critic learning) performs scalable policy optimization across multiple source domains, with convergence of the critic and actor updates guaranteed by Theorem 2 (critic error bound) and Theorem 3 (policy gradient convergence). Finally, Phase 4 (fast adaptation to new domains) exploits the learned meta-policy and compact domain factors to achieve rapid adaptation, where the adaptation performance gap is formally controlled by Theorem 4. Together, these results demonstrate that each algorithmic component is theoretically justified and collectively leads to provable scalability and generalization in networked MARL. Figure 3 visually illustrates the GSAC pipeline.

## 4.2 Algorithm overview

GSAC (Algorithm 1) consists of four sequential phases:

**Phase 1: Causal discovery and domain factor estimation.** In each source environment, agents estimate their local causal masks and latent domain factors to disentangle invariant structure from domain-specific variations; details are deferred to Section 5.

**Phase 2: Constructing ACRs.** Using the recovered causal masks, each agent constructs ACRs for value functions and policies as described in Section 3. These ACRs significantly reduce the input dimensionality while preserving decision-relevant information for the following phases.

**Phase 3: Meta-learning via actor-critic optimization.** Agents are trained on across $M$ source environments by optimizing policies through local actor-critic updates using ACR-based inputs. We provide a detailed description of this procedure in the following Section 4.3.

**Phase 4: Fast adaptation to target domain.** In a new, unseen environment, each agent collects a few trajectories and estimates its domain factor $\hat{\omega}_i^{M+1}$. The learned meta-policy $\pi_i^{\theta(K)}$ is then conditioned on the adapted ACR input $(\mathbf{s}_{\mathcal{N}_i}^\circ, \hat{\omega}_{\mathcal{N}_i}^{M+1})$ for immediate deployment. This process allows efficient generalization without requiring further policy training from scratch.

## 4.3 Key idea: meta-learning via actor-critic optimization

At each outer iteration $k$, a source domain $m(k)$ is sampled and each agent $i$ roll out trajectories using their current policies $\pi_i^{\theta(k)}$, which are conditioned on the compact ACR inputs $(\mathbf{s}_{\mathcal{N}_i}^\circ, \hat{\omega}_{\mathcal{N}_i}^\circ)$. These interactions are used to update both the critic $\hat{Q}$ and the actor $\pi^\theta$ in a decentralized yet coordinated manner, enabling policy generalization across domains.

*Critic update.* Each agent maintains a local tabular critic $\hat{Q}_i$ over its the ACR input space $\mathcal{S}_{\mathcal{N}_i^\kappa}^\circ \times \mathcal{A}_{\mathcal{N}_i^\kappa} \times \Omega_{\mathcal{N}_i^\kappa}^\circ$. At every inner iteration $t$, the critic is updated using temporal-difference (TD) learning (Line 16): the $Q$-value for the current state-action-domain triple is updated toward a bootstrap target composed of the received reward and the next-step value. All other $Q$-values remain unchanged. This TD update leads to an estimate of a truncated $Q$-function for current domain.

*Actor update.* After completing an episode, each agent aggregates the $Q$-values of all agents within its $\kappa$-hop neighborhood and weights them by the gradient of the log-policy at each timestep. The actor parameters $\theta$ are then updated using stochastic gradient ascent with stepsize $\eta_k = \eta/\sqrt{k+1}$.

We present the finite-time convergence and adaptation error bounds of GSAC in Section 6. Prior to that, we discuss causal discovery and domain factor estimation in *Phase 1* in Section 5.

---

**Algorithm 1** GENERALIZABLE AND SCALABLE ACTOR-CRITIC

---

1: **Input:** $\theta_i(0)$; parameter $\kappa$; $T$, length of each episode; stepsize parameters $h, t_0, \eta$.
2: **for** source domain index $m = 1, 2, \ldots, M$ **do**    ▷ *P1: causal recovery and domain estimation*
3:     Sample $\boldsymbol{\omega}^m \sim \mathcal{D}$, each agent $i$ estimate the causal mask $\mathbf{c}_i$ and domain factor $\hat{\boldsymbol{\omega}}_i^m$
4: **end for**
5: **for** each agent $i$ **do**                      ▷ *P2: approximately compact representation*
6:     Identify $\mathbf{s}_{\mathcal{N}_i^\kappa}^\circ \leftarrow \text{ACR}_Q(\mathbf{c}, i, \kappa)$ and $\mathbf{s}_{\mathcal{N}_i}^\circ \leftarrow \text{ACR}_\pi(\mathbf{c}, i)$
7:     Identify $\boldsymbol{\omega}_{\mathcal{N}_i^\kappa}^{m,\circ} \leftarrow \text{ACR}_Q(\mathbf{c}, i, \kappa)$ and $\boldsymbol{\omega}_{\mathcal{N}_i}^{m,\circ} \leftarrow \text{ACR}_\pi(\mathbf{c}, i)$ for each $m = 1, 2, \cdots, M$
8: **end for**
9: **for** $k = 0, 1, 2, \ldots, K - 1$ **do**                      ▷ *P3: meta-learning*
10:     Sample domain index $m(k) \sim \{1, \ldots, K\}$, set $\hat{\boldsymbol{\omega}}^\circ \leftarrow \hat{\boldsymbol{\omega}}^{m(k),\circ}$, sample $\mathbf{s}(0) \sim \rho_0$
11:     Each agent $i$ takes action $\mathbf{a}_i(0) \sim \pi_i^{\theta_i(k)}(\cdot \mid \mathbf{s}_{\mathcal{N}_i}^\circ(0), \hat{\boldsymbol{\omega}}_{\mathcal{N}_i}^\circ)$, and receive reward $r_i(0)$
12:     Initialize critic $\hat{Q}_i^0 \in \mathbb{R}^{\mathcal{S}_{\mathcal{N}_i^\kappa}^\circ \times \mathcal{A}_{\mathcal{N}_i^\kappa}^\circ \times \Omega_{\mathcal{N}_i^\kappa}^\circ}$ to be all zeros
13:     **for** $t = 1$ to $T$ **do**
14:         Get state $\mathbf{s}_i(t)$, take action $\mathbf{a}_i(t) \sim \pi_i^{\theta_i(k)}(\cdot \mid \mathbf{s}_{\mathcal{N}_i}^\circ(t), \hat{\boldsymbol{\omega}}_{\mathcal{N}_i}^\circ)$, get reward $r_i(t)$
15:         Update $Q$-function with stepsize $\alpha_{t-1} \leftarrow \frac{h}{t-1+t_0}$:

$$\hat{Q}_i^t(\mathbf{s}_{\mathcal{N}_i^\kappa}^\circ(t-1), \mathbf{a}_{\mathcal{N}_i^\kappa}(t-1), \hat{\boldsymbol{\omega}}_{\mathcal{N}_i^\kappa}^\circ) \leftarrow (1 - \alpha_{t-1})\hat{Q}_i^{t-1}(\mathbf{s}_{\mathcal{N}_i^\kappa}^\circ(t-1), \mathbf{a}_{\mathcal{N}_i^\kappa}(t-1), \hat{\boldsymbol{\omega}}_{\mathcal{N}_i^\kappa}^\circ)$$
$$+ \alpha_{t-1}\left(r_i(t-1) + \gamma\hat{Q}_i^{t-1}(\mathbf{s}_{\mathcal{N}_i^\kappa}^\circ(t), \mathbf{a}_{\mathcal{N}_i^\kappa}(t), \hat{\boldsymbol{\omega}}_{\mathcal{N}_i^\kappa}^\circ)\right),$$
$$\hat{Q}_i^t(\mathbf{s}_{\mathcal{N}_i^\kappa}^\circ, \mathbf{a}_{\mathcal{N}_i^\kappa}, \hat{\boldsymbol{\omega}}_{\mathcal{N}_i^\kappa}^\circ) \leftarrow \hat{Q}_i^{t-1}(\mathbf{s}_{\mathcal{N}_i^\kappa}^\circ, \mathbf{a}_{\mathcal{N}_i^\kappa}, \hat{\boldsymbol{\omega}}_{\mathcal{N}_i^\kappa}^\circ), \text{ for } (\mathbf{s}_{\mathcal{N}_i^\kappa}^\circ, \mathbf{a}_{\mathcal{N}_i^\kappa}) \neq (\mathbf{s}_{\mathcal{N}_i^\kappa}^\circ(t-1), \mathbf{a}_{\mathcal{N}_i^\kappa}(t-1))$$

16:     **end for**
17:     Each agent $i$ estimates policy gradient:

$$\hat{g}_i(k) \leftarrow \sum_{t=0}^T \gamma^t \cdot \frac{1}{n} \sum_{j \in \mathcal{N}_i^\kappa} \hat{Q}_j^T(\mathbf{s}_{\mathcal{N}_j^\kappa}^\circ(t), \mathbf{a}_{\mathcal{N}_j^\kappa}(t), \hat{\boldsymbol{\omega}}_{\mathcal{N}_j^\kappa}^\circ)\nabla_{\theta_i}\log\pi_i^{\theta_i(k)}(\mathbf{a}_i(t) \mid \mathbf{s}_{\mathcal{N}_i}^\circ(t), \hat{\boldsymbol{\omega}}_{\mathcal{N}_i}^\circ)$$

18:     Update policy: $\theta_i(k+1) \leftarrow \theta_i(k) + \eta_k\hat{g}_i(k)$ with stepsize $\eta_k \leftarrow \frac{\eta}{\sqrt{k+1}}$
19: **end for**
20: Collect few trajectories $\{(\mathbf{s}(t), \mathbf{a}(t), \mathbf{s}(t+1))\}_{t=0}^{T_a}$ in the new domain    ▷ *P4: generalization*
21: Each agent $i$ estimates new domain factor $\hat{\boldsymbol{\omega}}_i^{M+1}$, and deploy policy $\pi_i^{\theta(K)}(\cdot | \mathbf{s}_{\mathcal{N}_i}^\circ, \hat{\boldsymbol{\omega}}_{\mathcal{N}_i}^{M+1})$

---

## 5 Causal recovery and domain factor estimation

In this section, we first discuss the computational overhead of causal discovery and ACR construction, and then provide the theoretical guarantees for causal recovery and domain factor estimation in Theorem 1 and Proposition 4-5. Phase 1 (causal discovery) and Phase 2 (ACR construction) introduce upfront costs that are only one-time, local, and amortized. In particular, they are one-time preprocessing steps for each source domain and do not need to be repeated during meta-training or adaptation. Both steps are local per agent and over small neighborhoods, parallel across agents, and the results are re-used for the entire meta-training horizon and for adaptation.

**Theorem 1** (Structural identifiability in networked MARL). Under the standard faithfulness assumption, the structural matrices $\mathbf{c}_i$ in 6 are identifiable from the observed data.

Theorem 1 guarantees that the underlying structure, that encodes how neighboring states, local actions, and latent domain factors affect local transitions, can be uniquely recovered from trajectories under standard causal discovery assumptions. The proof is deferred to Appendix F. Furthermore, Proposition 4 provides a finite-sample guarantee for recovering the local structural dependencies.

**Proposition 4** (Informal). Under standard assumptions, including faithfulness, minimum mutual information for true causal links, bounded in-degree $d_{\max}$, sub-Gaussian noise, and Lipschitz continuity of the transition function, the sample complexity to recover the causal masks satisfies

$$\mathcal{O}\left(\frac{\dim(\mathbf{s}_{\mathcal{N}_i}) \cdot d_{\max}\log(\dim(\mathbf{s}_{\mathcal{N}_i}) \cdot n/\delta)}{\lambda^2}\right),$$

with probability at least $1 - \delta$, where $\lambda$ quantifies signal strength.

The required sample size scales almost linearly with the number of observed variables $\dim(\mathbf{s}_{\mathcal{N}_i}$, the sparsity level $d_{\max}$, and decays quadratically with the strength of causal signals $\lambda$. The formal statement and proof of Proposition 4, and discussions on the imposed assumptions are deferred to Appendix F.

**Proposition 5** (Informal)**.** Suppose the causal masks are recovered and the domain-dependent transition dynamics are identifiable. Assume that distinct domain factors induce distinguishable state transitions in total variation, and that $\Omega_i$ is compact with diameter $D_\Omega$. Then, with probability at least $1 - \delta$, the estimated factor $\hat{\boldsymbol{\omega}}$ given a trajectory of length $T_e$ generated from true factor $\boldsymbol{\omega}^*$ satisfies

$$\|\hat{\boldsymbol{\omega}} - \boldsymbol{\omega}^*\|_2 \le \delta_\omega(T_e) = \mathcal{O}\left(\sqrt{\frac{D_\Omega \log(nT_e/\delta)}{T_e}}\right). \tag{10}$$

The estimation error decays as $O(1/\sqrt{T_e})$ with high probability, and depends logarithmically on the number of agents. The result highlights that domain generalization can be performed efficiently with only a few samples. The formal statement and proof of Proposition 5 are deferred to Appendix F.

## 6 Convergence results and adaptation gap

For clarity of analysis, we first establish convergence and adaptation guarantees for an *ACR-free* variant of GSAC (Algorithm 5, detailed in Appendix D). Theoretical results for GSAC with ACR follow as corollaries. We define the expected return of a policy parameterized by $\theta$ using an estimated domain factor $\boldsymbol{\omega}'$ in a true environment $\boldsymbol{\omega}$:

$$J(\theta, \boldsymbol{\omega}'; \boldsymbol{\omega}) := \mathbb{E}_{\mathbf{s} \sim \rho_0} \mathbb{E}_{\mathbf{a}(t) \sim \pi^\theta(\cdot|\mathbf{s}(t), \boldsymbol{\omega}'), \mathbf{s}(t+1) \sim P_{\boldsymbol{\omega}}(\mathbf{s}(t), \mathbf{a}(t))} \left[\sum_{t=0}^{\infty} \gamma^t r(\mathbf{s}(t), \mathbf{a}(t)) \middle| \mathbf{s}(0) = \mathbf{s}\right], \tag{11}$$

where $\pi^\theta(\mathbf{a}|\mathbf{s}, \boldsymbol{\omega}) = \prod_{i=1}^n \pi_i^{\theta_i}(\mathbf{a}_i|\mathbf{s}_{\mathcal{N}_i}, \boldsymbol{\omega}_{\mathcal{N}_i})$ is the joint domain-conditioned policy. For notational simplicity, we write $J(\theta, \boldsymbol{\omega}) := J(\theta, \boldsymbol{\omega}; \boldsymbol{\omega})$ when the estimated and true domain factors coincide. To this end, for domain generalization our objective under domain distribution $\mathcal{D}$ is:

$$\max_{\theta \in \Theta} J(\theta) := \mathbb{E}_{\boldsymbol{\omega} \sim \mathcal{D}}[J(\theta, \boldsymbol{\omega})].$$

### 6.1 Convergence

We begin by introducing standard assumptions (Assumption 1-4) used in networked MARL [24], as well as additional ones (Assumption 5-6) for the domain generalization setting.

**Assumption 1.** Rewards are bounded: $0 \le r_i(\mathbf{s}_i, \mathbf{a}_i) \le \bar{r}$ for all $i, \mathbf{s}_i, \mathbf{a}_i$. The local state and action spaces satisfy $|\mathcal{S}_i| \le S$ and $|\mathcal{A}_i| \le A$.

**Assumption 2.** There exist $\tau \in \mathbb{N}$ and $\sigma \in (0, 1)$ such that, for any $\theta$ and $\boldsymbol{\omega}$, the local transition probabilities satisfy $\mathbb{P}_{\boldsymbol{\omega}}\left((\mathbf{s}_{\mathcal{N}_i^\kappa}(\tau), \mathbf{a}_{\mathcal{N}_i^\kappa}(\tau)) = (\mathbf{s}', \mathbf{a}') \mid (\mathbf{s}(1), \mathbf{a}(1)) = (\mathbf{s}, \mathbf{a})\right) \ge \sigma$ for all $i$ and all $((\mathbf{s}', \mathbf{a}'), (\mathbf{s}, \mathbf{a}))$ in the appropriate product spaces.

**Assumption 3.** For all $i$, $\mathbf{s}_{\mathcal{N}_i}$, $\boldsymbol{\omega}_{\mathcal{N}_i}$, $\mathbf{a}_i$, and $\theta_i$, $\|\nabla_{\theta_i} \log \pi_i^{\theta_i}(\mathbf{a}_i|\mathbf{s}_{\mathcal{N}_i}, \boldsymbol{\omega}_{\mathcal{N}_i})\| \le L_i$, $\|\nabla_\theta \log \pi^\theta(\mathbf{a}|\mathbf{s}, \boldsymbol{\omega})\| \le L := \sqrt{\sum_{i=1}^n L_i^2}$, and $\nabla J(\theta)$ is $L'$-Lipschitz continuous in $\theta$.

**Assumption 4.** Each agent's parameter space $\Theta_i \subset \mathbb{R}^{d_i^\theta}$ is compact with diameter bounded by $D_\Theta$.

**Assumption 5.** For all $i$: (i) $P_i(\mathbf{s}_i(t+1)|\mathbf{s}_{\mathcal{N}_i}(t), \mathbf{a}_i(t), \boldsymbol{\omega}_i)$ is $L_P$-Lipschitz in $\boldsymbol{\omega}_i$; (ii) $Q_i^\theta(\mathbf{s}, \mathbf{a}, \boldsymbol{\omega})$ is $L_Q$-Lipschitz in $\boldsymbol{\omega}$; (iii) $\nabla_{\theta_i} J(\theta, \boldsymbol{\omega})$ is $L_J$-Lipschitz in $\boldsymbol{\omega}$.

**Assumption 6.** The domain factor space $\Omega_i$ is compact with diameter bounded by $D_\Omega$.

**Discussion on assumptions.** Assumption 1-4 are standard for proving convergence of networked MARL algorithms, without consideration of domain generalization/adaptation [23, 17, 24]. To account for generalizability across domains in networked systems, we introduce additional Assumption 5-6 regarding the latent domain factor. Assumption 5 is similar with Assumption 3, which imposes the smoothness w.r.t. the domain factor, while Assumption 3 imposes the smoothness w.r.t. the actor parameter $\theta$. Assumption 6 is a regularity assumption to ensure the compactness of domain factor space. We provide a detailed discussion on these assumption in Appendix G.

**Critic error bound.** We first analyze the inner-loop critic update. Fixing any outer iteration $k$, and omitting $k$ from the notation, we establish the following result. Theorem 2 shows that the inner loop converges to an estimate of $Q_i$ with steady-state error decaying with $1/\sqrt{T_e}$ and exponentially in $\kappa$. The proof and formal statement of Theorem 2 are deferred to Appendix G.

**Theorem 2** (Critic error bound, informal). Under Assumptions 1–6, and for any $\delta \in (0,1)$, if the critic stepsize is set as $\alpha_t = h/(t + t_0)$ and the domain factor $\hat{\omega}$ is estimated from $T_e$ trajectories, then with probability at least $1 - \delta$, the critic estimate satisfies:

$$|Q_i(\mathbf{s}, \mathbf{a}, \boldsymbol{\omega}) - \hat{Q}_i^T(\mathbf{s}_{\mathcal{N}_i^\kappa}, \mathbf{a}_{\mathcal{N}_i^\kappa}, \hat{\boldsymbol{\omega}}_{\mathcal{N}_i^\kappa})| \leq \frac{C_a}{\sqrt{T + t_0}} + \frac{C_a'}{T + t_0} + \frac{2c\rho^{\kappa+1}}{(1-\gamma)^2} + C_{\boldsymbol{\omega}}'\sqrt{\frac{\log(nT_e/\delta)}{T_e}},$$

where $C_a$, $C_a'$, and $C_{\boldsymbol{\omega}}$ are constants. These will be further characterized and discussed in Section 6.3, where we analyze the additional benefits of incorporating ACR.

**Actor convergence.** Based on the critic bound, we derive the bound on policy gradient updates. The first term, of order $\mathcal{O}(1/\sqrt{K+1})$, vanishes as the number of outer iterations $K$ increases. The remaining three terms correspond to different sources of error: the second arises from neighborhood truncation and decays exponentially with $\kappa$; the third stems from estimation of the domain factor $\boldsymbol{\omega}$, with error decreasing as $1/\sqrt{T_e}$; and the fourth reflects the approximation of the domain distribution $\mathcal{D}$ using only $M$ sampled source domains, decaying with $1/\sqrt{M}$. The formal statement and proof of Theorem 3 are deferred to Appendix G.

**Theorem 3** (Policy gradient convergence, informal). Under Assumptions 1–6, for $K \geq 3$ and sufficiently large $T$, if the actor and critic stepsizes are chosen appropriately and domain factors are estimated from $T_e$ samples, then with probability at least $1 - \delta$:

$$\frac{\sum_{k=0}^{K-1} \eta_k \|\nabla J(\theta(k))\|^2}{\sum_{k=0}^{K-1} \eta_k} \leq \tilde{\mathcal{O}}\left(\frac{1}{\sqrt{K+1}} + \rho^{\kappa+1} + \sqrt{\frac{1}{T_e}} + \sqrt{\frac{1}{M}}\right).$$

## 6.2 Generalization

In *Phase 4*, for a new domain $\boldsymbol{\omega}^{M+1} \sim \mathcal{D}$, `ACR-free GSAC` collects $T_a$ trajectories, estimates $\hat{\boldsymbol{\omega}}^{M+1}$, and deploys the policy $\pi^{\theta(K)}(\cdot|\mathbf{s}, \hat{\boldsymbol{\omega}}^{M+1})$. The expected return is:

$$J(\theta(K), \hat{\boldsymbol{\omega}}^{M+1}; \boldsymbol{\omega}^{M+1}) = \mathbb{E}_{\mathbf{s}(0)\sim\rho_0}\mathbb{E}_{\mathbf{a}(t)\sim\pi^\theta(\cdot|\mathbf{s}(t),\hat{\boldsymbol{\omega}}^{M+1}),\mathbf{s}(t+1)\sim P_{\boldsymbol{\omega}^{M+1}}(\mathbf{s}(t),\mathbf{a}(t))}\left[\sum_{t=0}^{\infty} \gamma^t r_t\right].$$

**Theorem 4** (Adaptation guarantee). Under Assumptions 1–6, with probability at least $1 - \delta$:

$$\mathbb{E}\left[J(\theta(K), \hat{\boldsymbol{\omega}}^{M+1}; \boldsymbol{\omega}^{M+1}) \mid \theta(K)\right] \geq J(\theta(K)) - L_{\boldsymbol{\omega}'}C_{\boldsymbol{\omega}}\sqrt{\frac{\log(n/\delta)}{T_a}}.$$

Theorem 4 establishes that the adaptation gap decreases at a rate of $\mathcal{O}(1/\sqrt{T_a})$, relative to the return of the meta-trained policy. The proof is deferred to Appendix G.

## 6.3 Additional gains from ACR

Notably, ACR introduces only a constant-factor increase in approximation error over truncation (multiplicative factor 3), and thus all ACR-free convergence and generalization results naturally extend to `GSAC`. Beyond computational gains discussed in Section 3, ACR also improves *sample efficiency*. In Theorem 2, the explicit constants are defined as $C_a := \frac{6\bar{\epsilon}}{1-\sqrt{\gamma}}\sqrt{\frac{\tau h}{\sigma}\left[\log(\frac{2\tau T^2}{\delta}) + f(\kappa)\log SA\right]}$ with $\bar{\epsilon} := 4\frac{\bar{r}}{1-\gamma} + 2\bar{r}$, $C_a' := \frac{2}{1-\sqrt{\gamma}}\max(\frac{16\bar{\epsilon}h\tau}{\sigma}, \frac{2\bar{r}}{1-\gamma}(\tau + t_0))$, and $C_{\boldsymbol{\omega}}' := L_Q C_{\boldsymbol{\omega}}\sqrt{D_\Omega}$, and the stepsize $\alpha_t = \frac{h}{t+t_0}$ satisfying $h \geq \frac{1}{\sigma}\max(2, \frac{1}{1-\sqrt{\gamma}})$, $t_0 \geq \max(2h, 4\sigma h, \tau)$. By leveraging ACR, the effective dimensionality of the $\kappa$-hop state space is dramatically reduced, leading to smaller $\tau$ and larger $\sigma$ in Assumption 2. This reduces $C_a$, $C_a'$, and $t_0$, enabling faster convergence and reducing the required inner-loop iterations $T$ to achieve a given accuracy in $Q$-value estimation of within critic, thereby decreasing the total sample complexity.

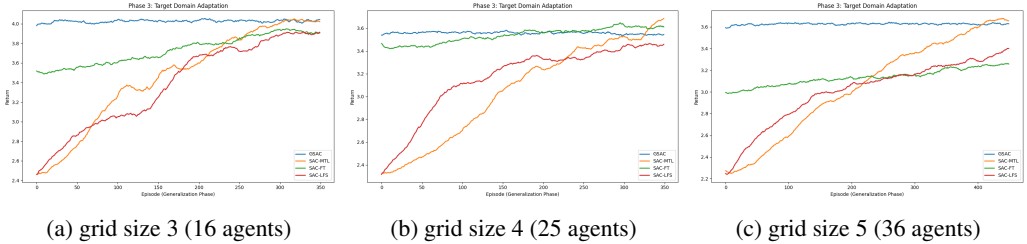

| (a) grid size 3 (16 agents) | (b) grid size 4 (25 agents) | (c) grid size 5 (36 agents) |

Figure 2: Adaptation comparison for different grid sizes in wireless communication benchmarks.

# 7 Numerical experiments

We evaluate `GSAC` on two standard benchmarks for networked MARL algorithms: wireless communications [44, 24, 36] and traffic control [30, 24] (the latter deferred to Appendix H.2). A detailed description of the experimental setup and comprehensive results are provided in Appendix H. We consider $M = 3$ source domains, each with domain factors $\boldsymbol{\omega} \in \{0.2, 0.5, 0.8\}$, while the target domain uses $\boldsymbol{\omega}_{\text{target}} = 0.65$ unless otherwise specified. In each domain, `GSAC` is trained for $K$ outer iterations (depending on convergence), with a inner loop horizon $T = 10$. For domain factor estimation, we collect $T_e = 20$ trajectories per domain. To evaluate adaptation performance in the target domain, we compare `GSAC` against three baselines:

- `GSAC` (**ours**): Learn $\pi_\theta(a|s, \omega)$ by optimizing $\theta$ over source domains. At test time, estimate $\omega'$ and directly deploy $\pi_\theta(a|s, \omega')$ in the target domain without further training.
- `SAC-MTL` (**multi-task**): Learn $\pi_\theta(a|s, z)$, where $z$ denotes the *pre-specified* one-hot encoding of each source domain, and jointly optimize $\theta$ across domains [37]; deploy $\pi_\theta(a|s, z')$ in the target domain.
- `SAC-FT` (**fine-tune**): Train a single policy $\pi_\theta(a|s)$ across source domains without domain-factor conditioning and fine-tune $\theta$ in the target domain.
- `SAC-LFS` (**learning from scratch**): Train $\pi_\theta(a|s)$ in the target domain without prior meta-training.

To evaluate both scalability and generalizability, we vary the grid size of the wireless communication network (grid size $\in \{3, 4, 5\}$), which affects both the number of users and the connectivity structure (see Figure 4-5). As shown in Figure 2 and Figure 6, `GSAC` consistently maintains high training and adaptation performance across grid sizes, demonstrating its scalability and generalizability. In particular, `GSAC` achieves the best few-shot performance (Episodes 1–30), reflecting rapid adaptation from minimal target data. In contrast, `SAC-MTL` exhibits moderate performance during the early adaptation phase but improves steadily over time. This is because multi-task learning leverages shared structure across source domains, yet lacks explicit domain-factor conditioning, leading to slower adaptation compared to `GSAC`. `SAC-FT` performs worse in the early phase due to its need to fine-tune policy parameters directly in the target domain, resulting in higher sample complexity. `SAC-LFS` suffers the slowest convergence and lowest overall return, underscoring the importance of meta-training and domain factor conditioning for efficient cross-domain adaptation.

# 8 Conclusions and future work

We presented `GSAC`, a causality-aware MARL framework that integrates ACR construction with meta actor–critic learning, achieving provable scalability, fast adaptation, and strong generalization across domains. We established quasi-linear sample complexity and finite-sample convergence guarantees, and showed empirically that `GSAC` consistently outperforms competitive baselines on challenging networked MARL benchmarks.

The main limitation of our current work is that `GSAC` has only been evaluated on tabular and fully observed benchmarks. Nonetheless, it establishes a principled foundation for practical control of large-scale networked systems. Promising future directions include: (i) extending `GSAC` to continuous state and action spaces with function approximation based on ACRs [10]; (ii) broadening empirical evaluation to more diverse and large-scale networked systems [19]; and (iii) incorporating partial observability into the framework.

## Acknowledgments

We would like to thank all the anonymous reviewers for their careful proofreading and constructive feedback, which have greatly improved the quality of this work. In particular, we thank one reviewer for the insightful suggestions on adding the roadmap linking theorems and algorithms in Section 4, as well as for recommending additional discussions on Assumptions 1–6. We also thank the other reviewers for suggesting the inclusion of two additional baseline methods and an extra benchmark, along with further discussions on the assumptions underlying Theorem 1 and Proposition 4. We are grateful to Chengdong Ma for valuable initial discussions. This work was supported by the Engineering and Physical Sciences Research Council [grant number EP/Y003187/1 and UKRI849].

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

# A Related works

**Networked MARL.**   Networked systems are prevalent across a wide range of real-world domains, including power grids [3, 4], transportation networks [41, 9], and wireless communications [1, 44]. Our work is closely related to the rapidly growing body of research on *multi-agent reinforcement learning in networked systems* [23, 17, 24]. These studies exploit localized interactions to improve scalability, typically by designing decentralized policy optimization algorithms that learn per-agent policies with theoretical convergence guarantees. Notably, [24] provide the first provably efficient MARL framework for networked systems under the discounted reward setting. Subsequent extensions address average-reward formulations [23] and stochastic, non-local network topologies [17]. However, none of these methods simultaneously address *both* generalization across domains and scalability in large networked systems. Our work fills this gap by proposing a principled framework—built on causal representation learning and domain-conditioned policy optimization—that achieves provable generalization and scalability in networked MARL.

A complementary but distinct line of work focuses on *fully decentralized MARL via neighbor-to-neighbor communication*. [38] propose a fully decentralized actor–critic algorithm, where agents update local policies using only local observations and consensus-based messages from neighbors, and prove convergence under linear function approximation. Other strategies such as mean-field MARL [35] approximate interactions by aggregating neighborhood effects, trading some fidelity for tractability. Graph neural networks and message-passing schemes have also gained traction [13].

More recent theoretical efforts in decentralized learning emphasize convergence speed and communication complexity under peer-to-peer protocols. [15] show that message passing improves regret bounds over non-communicative baselines. [28] establish almost-sure convergence for consensus-based multi-task actor–critic algorithms. [34] propose a communication protocol for distributed adaptive control with provable bounds on consensus error and policy suboptimality. [16] provide the first finite-sample analysis of distributed tabular Q-learning, linking sample complexity to spectral properties of the network and Markov mixing times. Finally, [8] design a communication-efficient decentralized TD-learning algorithm that matches optimal sample complexity while reducing communication overhead.

**Domain Generalization and Adaptation in RL.**   RL agents frequently face *environmental shifts* between training and deployment, motivating research into policy generalization and fast adaptation. One line of work leverages *causal representation learning* to identify invariant features and disentangle spurious correlations. For example, [18] introduce a framework for learning invariant causal representations, with provable generalization across environments. Similarly, [40] trace causal origins of non-stationarity to construct stable representations. [10] propose AdaRL, which learns parsimonious graphical models that pinpoint minimal domain-specific differences, enabling efficient adaptation from limited target data. Related efforts seek *domain-invariant representations* through bisimulation metrics [39], which preserve decision-relevant dynamics while filtering nuisance factors. Causal modeling has also been applied to goal-conditioned transitions [6] and structure-aware meta-learning [10]. However, these methods are primarily limited to single-agent settings.

Complementary to representation-based generalization, a growing body of work studies *risk-aware* or *robust* RL in the single-agent setting, designed to generalize under uncertainty, distribution shift, or tail-risk objectives. Representative approaches include risk-aware RL such as CVaR and entropic risk [27, 5, 14], and robust MDP formulations that optimize worst-case or ambiguity-aware performance across environments [12, 20]. While these methods provide principled guarantees against uncertainty, they primarily focus on single-agent decision making and do not address scalability or structural generalization in multi-agent, networked systems.

Additional approaches not directly aligned with ours include *meta-RL* and *sim-to-real transfer*. Meta-RL methods such as PEARL [25], VariBAD [43], and ProMP [26] remain foundational for few-shot adaptation by optimizing meta-objectives or latent context embeddings. In sim-to-real transfer, [31] demonstrate provable improvements in exploration efficiency by leveraging simulated environments during training. In imitation learning, [2] propose ICIL, which learns invariant causal features to replicate expert behavior across different domains. While effective in their respective domains, these works do not address the unique structural and scalability challenges posed by large-scale networked MARL.

# B  Table of notation

| Symbol | Definition |
|---|---|
| *Networked MARL Setting* | |
| $\mathscr{N}$ | Set of agents, indexed by $i \in \mathscr{N}$ |
| $\mathscr{G} = (\mathscr{N}, \mathscr{E})$ | Agent interaction graph |
| $\mathcal{N}_i, \mathcal{N}_i^\kappa$ | 1-hop and $\kappa$-hop neighborhood of agent $i$ |
| $\mathcal{N}_{-i}^\kappa$ | Agents not in $\mathcal{N}_i^\kappa$, i.e., $\mathscr{N} \setminus \mathcal{N}_i^\kappa$ |
| $\mathcal{S}, \mathcal{S}_i$ | Global and agent $i$'s state space |
| $\mathcal{A}, \mathcal{A}_i$ | Global and agent $i$'s action space |
| $\mathbf{s}, \mathbf{s}_i, \mathbf{s}_{\mathcal{N}_i}$ | Global state, local state of agent $i$, and neighbor states |
| $\mathbf{a}, \mathbf{a}_i, \mathbf{a}_{\mathcal{N}_i}$ | Global action, agent $i$'s action, and neighborhood actions |
| $\pi^\theta, \pi_i^{\theta_i}$ | Joint policy, $\prod_{i=1}^n \pi_i^{\theta_i}$; Agent $i$'s local policy, parameterized by $\theta_i$ |
| $\theta, \theta_i$ | Joint and local policy parameters |
| $r_i(\mathbf{s}_i, \mathbf{a}_i), r(\mathbf{s}, \mathbf{a})$ | Local reward function, Global reward: $\frac{1}{n}\sum_i r_i(\mathbf{s}_i, \mathbf{a}_i)$ |
| $\rho_0$ | Initial state distribution |
| $\gamma$ | Discount factor |
| *Causal Modeling and Domain Generalization* | |
| $\boldsymbol{\omega}_i, \boldsymbol{\omega}$ | Latent domain factor for agent $i$, and full domain vector |
| $\Omega, \Omega_i$ | Domain factor space and local domain space |
| $\mathcal{D}$ | Distribution over domain factors $\boldsymbol{\omega}$ |
| $\mathbf{c}^{\cdot \to \cdot}$ | Binary causal masks (e.g., $\mathbf{c}^{\mathbf{a} \to \mathbf{s}}$) |
| $f_{i,j}(\cdot)$ | Transition function for $j$-th state dimension of agent $i$ |
| $\epsilon_{i,j}^{\mathbf{s}}(t)$ | Random noise in transition dynamics |
| $s_{i,j}$ | $j$-th component of agent $i$'s state |
| $d_i^{\mathbf{s}}, d_i^{\mathbf{a}}$ | Dimensionality of state and action space of agent $i$ |
| *Value Functions and ACR* | |
| $Q^\pi(\mathbf{s}, \mathbf{a}), Q_i^\pi(\mathbf{s}, \mathbf{a})$ | Global action-value function, Local value contribution of agent $i$ |
| $\hat{Q}_i^\pi, \tilde{Q}_i^\pi$ | Truncated $Q$-function, ACR-based approximate $Q$-function |
| $\mathbf{s}_{\mathcal{N}_i^\kappa}^\circ, \mathbf{s}_{\mathcal{N}_i}^\circ$ | ACR of $\kappa$-hop state, ACR of agent $i$'s local neighborhood state |
| $\boldsymbol{\omega}_{\mathcal{N}_i^\kappa}^\circ, \boldsymbol{\omega}_{\mathcal{N}_i}^\circ$ | ACR of $\kappa$-hop domain factor, ACR of local domain factor |
| $\tilde{\pi}_i$ | ACR-based approximately compact local policy |
| $\bar{\mathbf{s}}, \bar{\mathbf{a}}, \bar{\boldsymbol{\omega}}$ | Fixed placeholders for unobserved variables in truncation |
| *Algorithm and Learning* | |
| $T$ | Inner-loop episode horizon |
| $K$ | Number of outer-loop meta-training iterations |
| $T_e$ | Number of trajectories used for estimating $\boldsymbol{\omega}$ during meta-training |
| $T_a$ | Number of trajectories used to estimate $\boldsymbol{\omega}$ in target domain |
| $\alpha_t, \eta_k$ | Critic and actor learning rates |
| $\hat{Q}_i^t$ | Estimated critic for agent $i$ at time $t$ |
| $\hat{g}_i(k)$ | Estimated policy gradient at outer iteration $k$ |
| $h, t_0$ | Critic learning rate parameters: $\alpha_t = \frac{h}{t+t_0}$ |
| *Theoretical Constants and Bounds* | |
| $\bar{r}$ | Upper bound on local rewards |
| $\lambda$ | Minimum signal strength for causal discovery |
| $D_\Omega$ | Diameter of domain factor space |
| $L_P, L_Q, L_J$ | Lipschitz constants for dynamics, $Q$-function, gradient in $\boldsymbol{\omega}$ |
| $L, L'$ | Gradient and smoothness bounds of policy $\pi^\theta$ |
| $\sigma, \tau$ | Minimum visitation probability and mixing time constants |
| $C_a, C_a'$ | Constants in critic approximation bounds |
| $\delta, \delta_\omega$ | Confidence level and domain factor estimation error |
| $\rho$ | Exponential decay rate in $Q$-function approximation |
| $\kappa$ | Truncation radius (number of hops) |
| $M$ | Number of source domains |
| $n$ | Number of agents |
| $S, A$ | Maximum size of local state/action space |

## C  Background in networked MARL

In this section, we provide background on networked MARL, beginning with the definition of the exponential decay property introduced in [23].

**Definition 3** (Exponential decay property). The $(c, \rho)$-exponential decay property holds if, for any localized policy $\theta$, for any $i \in \mathcal{N}, \mathbf{s}_{\mathcal{N}_i^\kappa} \in \mathcal{S}_{\mathcal{N}_i^\kappa}, \mathbf{s}_{\mathcal{N}_{-i}^\kappa}, \mathbf{s}'_{\mathcal{N}_{-i}^\kappa} \in \mathcal{S}_{\mathcal{N}_{-i}^\kappa}, \mathbf{a}_{\mathcal{N}_i^\kappa} \in \mathcal{A}_{\mathcal{N}_i^\kappa}, \mathbf{a}_{\mathcal{N}_{-i}^\kappa}, \mathbf{a}'_{\mathcal{N}_{-i}^\kappa} \in \mathcal{A}_{\mathcal{N}_{-i}^\kappa}, Q_i^\pi$ satisfies,

$$\left| Q_i^\pi \left( \mathbf{s}_{\mathcal{N}_i^\kappa}, \mathbf{s}_{\mathcal{N}_{-i}^\kappa}, \mathbf{a}_{\mathcal{N}_i^\kappa}, \mathbf{a}_{\mathcal{N}_{-i}^\kappa} \right) - Q_i^\pi \left( \mathbf{s}_{\mathcal{N}_i^\kappa}, \mathbf{s}'_{\mathcal{N}_{-i}^\kappa}, a_{\mathcal{N}_i^\kappa}, \mathbf{a}'_{\mathcal{N}_{-i}} \right) \right| \leq c\rho^{\kappa+1}.$$

Furthermore, [24] proves that the exponential decay property holds generally with $\rho = \gamma$ for discounted MDPs.

**Lemma 1.** Assume $\forall i, r_i \leq \bar{r}$. Then the $\left( \frac{\bar{r}}{1-\gamma}, \gamma \right)$-exponential decay property holds.

The exponential decay property implies that the dependence of $Q_i^\pi$ on other agents shrinks quickly as the distance grows, which motivates the truncated $Q$-functions [24],

$$\hat{Q}_i^\pi \left( \mathbf{s}_{\mathcal{N}_i^\kappa}, \mathbf{a}_{\mathcal{N}_i^\kappa} \right) := \sum_{\mathbf{s}_{\mathcal{N}_{-i}^\kappa} \in \mathcal{S}_{\mathcal{N}_{-i}^\kappa}, \mathbf{a}_{\mathcal{N}_{-i}^\kappa} \in \mathcal{A}_{\mathcal{N}_{-i}^\kappa}} w_i \left( \mathbf{s}_{\mathcal{N}_{-i}^\kappa}, \mathbf{a}_{\mathcal{N}_{-i}^\kappa}; \mathbf{s}_{\mathcal{N}_i^\kappa}, \mathbf{a}_{\mathcal{N}_i^\kappa} \right) Q_i^\pi \left( s_{\mathcal{N}_i^\kappa}, \mathbf{s}_{\mathcal{N}_{-i}^\kappa}, \mathbf{a}_{\mathcal{N}_i^\kappa}, \mathbf{a}_{\mathcal{N}_{-i}^\kappa} \right),$$

where $w_i \left( \mathbf{s}_{\mathcal{N}_{-i}^\kappa}, \mathbf{a}_{\mathcal{N}_{-i}}; \mathbf{s}_{\mathcal{N}_i^\kappa}, \mathbf{a}_{\mathcal{N}_i^\kappa} \right)$ are any non-negative weights satisfying

$$\sum_{\mathbf{s}_{\mathcal{N}_{-i}} \in \mathcal{S}_{\mathcal{N}_{-i}^\kappa}, \mathbf{a}_{\mathcal{N}_{-i}^\kappa} \in \mathcal{A}_{\mathcal{N}_{-i}^\kappa}} w_i \left( \mathbf{s}_{\mathcal{N}_{-i}^\kappa}, \mathbf{a}_{\mathcal{N}_{-i}}; \mathbf{s}_{\mathcal{N}_i^\kappa}, \mathbf{a}_{\mathcal{N}_i^\kappa} \right) = 1, \quad \forall \left( \mathbf{s}_{\mathcal{N}_i^\kappa}, \mathbf{a}_{\mathcal{N}_i^\kappa} \right) \in \mathcal{S}_{\mathcal{N}_i^\kappa} \times \mathcal{A}_{\mathcal{N}_i^\kappa}.$$

With the definition of the truncated $Q$-function, when the exponential decay property holds, the truncated $Q$-function approximates the full $Q$-function with high accuracy and can be used to approximate the policy gradient.

**Lemma 2.** Under the $(c, \rho)$-exponential decay property, the following holds: (a) Any truncated $Q$-function satisfies,

$$\sup_{(\mathbf{s},\mathbf{a}) \in \mathcal{S} \times \mathcal{A}} \left| \hat{Q}_i^\pi \left( \mathbf{s}_{\mathcal{N}_i^\kappa}, \mathbf{a}_{\mathcal{N}_i^\kappa} \right) - Q_i^\pi(\mathbf{s}, \mathbf{a}) \right| \leq c\rho^{\kappa+1}.$$

(b) Given $i$, define the following truncated policy gradient,

$$\hat{h}_i(\theta) = \frac{1}{1-\gamma} \mathbb{E}_{\mathbf{s} \sim \rho^\theta, \mathbf{a} \sim \pi^\theta(\cdot|\mathbf{s})} \left[ \frac{1}{n} \nabla_{\theta_i} \log \pi_i^{\theta_i} \left( \mathbf{a}_i \mid \mathbf{s}_i \right) \sum_{j \in \mathcal{N}_i^\kappa} \hat{Q}_j^\pi \left( \mathbf{s}_{\mathcal{N}_j^\kappa}, \mathbf{a}_{\mathcal{N}_j^\kappa} \right) \right].$$

Then, if $\left\| \nabla_{\theta_i} \log \pi_i^{\theta_i} \left( \mathbf{a}_i \mid \mathbf{s}_i \right) \right\| \leq L_i, \forall \mathbf{a}_i, \mathbf{s}_i$, we have $\left\| \hat{h}_i(\theta) - \nabla_{\theta_i} J(\theta) \right\| \leq \frac{cL_i}{1-\gamma} \rho^{\kappa+1}$.

Lemma 2 shows that the truncated $Q$-function has a much smaller dimension than the true $Q$-function, and is thus scalable to compute and store. However, despite the reduction in dimension, the error resulting from the approximation is small.

## D  Algorithm design

### D.1  Roadmap and pipeline

Figure 3 visually illustrates the GSAC pipeline, complementing the roadmap. It complements Algorithm 1 by showing how causal recovery and ACR construction (Phases 1–2) compress the state-action space, enabling efficient meta-policy training (Phase 3), and how this meta-policy rapidly adapts to new domains (Phase 4), as guaranteed by Theorems 2-4.

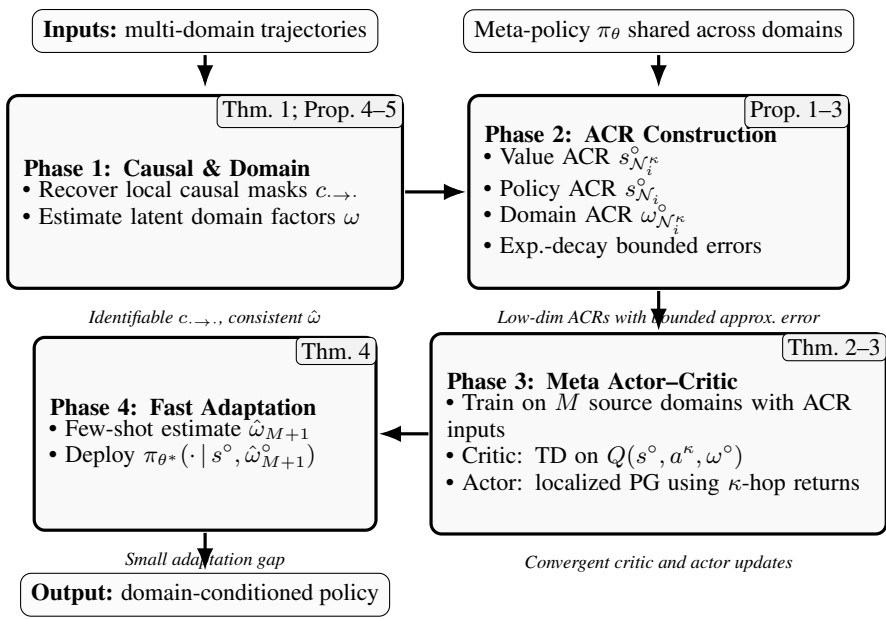

Figure 3: GSAC pipeline (cf. Algorithm 1). Each phase is annotated with its supporting results.

## D.2 Policy ACR

We present the complete pseudocode of the algorithm used in defining the policy ACR in Definition 2. We begin with Algorithm 2, which constructs the policy ACR using a variable number of steps. As shown in Proposition 6, Algorithm 2 precisely identifies the relevant components of $s_i$ for all agents while preserving the values of the $Q$-function. However, the number of steps required by Algorithm 2 is problem-dependent and may vary. To address this, we propose Algorithm 3, which constructs the policy ACR with a fixed step budget $\kappa$. At a high level, Algorithm 3 identifies all components of $s_i$ that influence future rewards within $\kappa$ time steps, resulting in an approximation error that decays with increasing $\kappa$. This approximation guarantee is formally established in Proposition 7.

---

**Algorithm 2** Policy ACR

---

**Input: causal mask $\mathbf{c} = \mathbf{c}_{i \in \mathcal{N}}$**

1: *// Initialization*:
2: $\mathbf{s}_i^\circ \leftarrow \varnothing$ for all $i \in \mathcal{N}$
3: **for** agent $i = 1, 2, \ldots,$ **do**
4:     **for** component $j = 1, 2, \ldots, d_i^s$ **do**
5:         **if** there is a direct link from $s_{i,j}$ to $r_i$, i.e., $c_{i,j}^{s \to r} = 1$ **then** $\mathbf{s}_i^\circ \leftarrow \mathbf{s}_i^\circ \cup \{s_{i,j}\}$
6:         **end if**
7:     **end for**
8: **end for**
9: *// Recursively identify with local communication*:
10: **while** $\mathbf{s}_i^\circ$ not converged for some $i$ **do**
11:     **for** agent $i = 1, 2, \ldots,$ **do**
12:         agent $i$ receives $(\mathbf{s}_{i'}^\circ)_{i' \in N_i/i}$ from its neighbors and $\mathbf{s}_{\mathcal{N}_i}^\circ \leftarrow \cup_{i' \in N_i} \{\mathbf{s}_{i'}^\circ\}$
13:         **for** component $j = 1, 2, \ldots, d_i^s$ **do**
14:             **if** there is an edge from $s_{i,j}$ to (component of) $\mathbf{s}_{\mathcal{N}_i}^\circ$ **then** $\mathbf{s}_i^\circ \leftarrow \mathbf{s}_i^\circ \cup \{s_{i,j}\}$
15:             **end if**
16:         **end for**
17:     **end for**
18: **end while**

**Output: policy ACR for each agent $\mathbf{s}_{\mathcal{N}_i}^\circ$**

---

---

**Algorithm 3** Policy ACR with finite steps

---

**Input: causal mask $\mathbf{c} = (\mathbf{c}_i)_{i \in \mathcal{N}}$**

1: // *Initialization*:
2: $\mathbf{s}_i^\circ(1) \leftarrow \varnothing$ for all $i \in \mathcal{N}$
3: **for** agent $i = 1, 2, \ldots,$ **do**
4:     **for** component $j = 1, 2, \ldots, d_i^s$ **do**
5:         **if** there is a direct link from $s_{i,j}$ to $r_i$, i.e., $c_{i,j}^{s \to r} = 1$ **then** $\mathbf{s}_i^\circ(1) \leftarrow \mathbf{s}_i^\circ(1) \cup \{s_{i,j}\}$
6:         **end if**
7:     **end for**
8: **end for**
9: // *Recursive identification with local communication in finite steps*:
10: **for** step $l = 1, 2, \ldots, \kappa - 1$ **do**
11:     **for** agent $i = 1, 2, \ldots,$ **do**
12:         agent $i$ receives $(\mathbf{s}_{i'}^\circ(l))_{i' \in N_i/i}$ from its neighbors and $\mathbf{s}_{\mathcal{N}_i}^\circ(l) \leftarrow \cup_{i' \in N_i} \{\mathbf{s}_{i'}^\circ(l)\}$
13:         $\mathbf{s}_i^\circ(l+1) \leftarrow \mathbf{s}_i^\circ(l)$
14:         **for** component $j = 1, 2, \ldots, d_i^s$ **do**
15:             **if** there is an edge from $s_{i,j}$ to $\mathbf{s}_{\mathcal{N}_i}^\circ(l)$ **then** $\mathbf{s}_i^\circ(l+1) \leftarrow \mathbf{s}_i^\circ(l+1) \cup \{s_{i,j}\}$
16:             **end if**
17:         **end for**
18:     **end for**
19: **end for**
20: **for** agent $i = 1, 2, \ldots,$ **do**
21:     agent $i$ receives $(\mathbf{s}_{i'}^\circ(\kappa))_{i' \in \mathcal{N}_i/i}$ from its neighbors and $\mathbf{s}_{\mathcal{N}_i}^\circ \leftarrow \cup_{i' \in \mathcal{N}_i} \{\mathbf{s}_{i'}^\circ(\kappa)\}$
22: **end for**

**Output: policy ACR $\mathbf{s}_{\mathcal{N}_i}^\circ$ for each agent $i$**

---

## D.3 GSAC

We provide the complete pseudo-code of GSAC in Algorithm 4.

## D.4 ACR-free GSAC

We provide the complete pseudo-code of ACR-free GSAC in Algorithm 5.

# E Missing proofs in Section 3

## E.1 Value ACR

Definition 1 recursively identifies the key state components within the $\kappa$-hop neighborhood that influence an agent's reward, yielding a compact subset $\mathbf{s}_{\mathcal{N}_i^\kappa}^\circ \subset \mathbf{s}_{\mathcal{N}_i^\kappa}$ with an exponentially small approximation error in the $Q$-function. This definition assumes a fixed number of time steps $\kappa$. In contrast, Definition 4 exactly identifies all state components in $\mathbf{s}_{\mathcal{N}_i^\kappa}$ that influence the agent's reward but requires a variable number of time steps. Notably, Definition 4 is stricter and implies Definition 1. Specifically, the former identifies only the components that influence the reward within $\kappa$ steps, while the latter includes all components in $\mathbf{s}_{\mathcal{N}_i^\kappa}$ that have a direct or indirect influence on the reward, possibly through longer dependencies.

**Definition 4** (Value ACR). Given the graph $\mathcal{G}$ encoded by the binary masks $\mathbf{c}$, for each agent $i$ and its $\kappa$-hop neighborhood $\mathcal{N}_i^\kappa$, we recursively define the $\kappa$-hop ACR $\mathbf{s}_{\mathcal{N}_i^\kappa}^\circ(t)$ as the set of state components $s_{\mathcal{N}_i^\kappa, j}$ satisfying at least one of the following:

- $s_{i,j}$ has a direct link to agent $i$'s reward $r_i$, i.e., $c_{i,j}^{s \to r} = 1$, or

- $s_{\mathcal{N}_i^\kappa, j}(t)$ has an edge to another state component $s_{l, \mathcal{N}_i^\kappa}(t+1)$ such that $s_{l, \mathcal{N}_i^\kappa}(t) \in \mathbf{s}_{\mathcal{N}_i^\kappa}^\circ$,

We provide the proof of Proposition 1 here.

**Algorithm 4** GENERALIZABLE AND SCALABLE ACTOR-CRITIC

1: **Input:** $\theta_i(0)$; parameter $\kappa$; $T$, length of each episode; stepsize parameters $h, t_0, \eta$.
2: // *Phase 1: local causal model learning*
3: **for** source domain index $m = 1, 2, \ldots, M$ **do**
4:     Sample $\boldsymbol{\omega}^m \sim \mathcal{D}$
5:     Collect trajectories $\tau^m = (\mathbf{s}^m(t), \mathbf{a}^m(t), r^m(t), \mathbf{s}^m(t+1))_{t=1}^{T_e}$
6:     Split $\tau^m$ into $\tau_i^m = \left(\mathbf{s}_{\mathcal{N}_i}^m(t), \mathbf{a}_i^m(t), r_i(t), \mathbf{s}_i^m(t+1)\right)_{t=1}^{T_e}$ for each agent $i$
7:     Each agent $i$ estimate the causal mask $\mathbf{c}_i$, model parameter $\psi_i$, and domain factor $\hat{\boldsymbol{\omega}}_i^m$
8: **end for**
9: // *Phase 2: approximately compact representation*
10: **for** each agent $i$ **do**
11:     Identify $\mathbf{s}_{\mathcal{N}_i^\kappa}^\circ \leftarrow \mathtt{ACR}_v(\mathbf{c}, i, \kappa)$ and $\mathbf{s}_{\mathcal{N}_i}^\circ \leftarrow \mathtt{ACR}_\pi(\mathbf{c}, i)$
12:     Identify $\boldsymbol{\omega}_{\mathcal{N}_i^\kappa}^{m,\circ} \leftarrow \mathtt{ACR}_v(\mathbf{c}, i, \kappa)$ and $\boldsymbol{\omega}_{\mathcal{N}_i}^{m,\circ} \leftarrow \mathtt{ACR}_\pi(\mathbf{c}, i)$ for each $m = 1, 2, \cdots, M$
13: **end for**
14: // *Phase 3: meta-learning*
15: **for** $k = 0, 1, 2, \ldots, K-1$ **do**
16:     Sample domain index $m(k)$ uniformly from $\{1, \ldots, K\}$
17:     Set $\hat{\boldsymbol{\omega}} = \hat{\boldsymbol{\omega}}^{m(k)}$                                        ▷ Current domain factors
18:     Sample initial state $\mathbf{s}(0) \sim \rho_0$
19:     **for** each agent $i$ **do**
20:         Take action $\mathbf{a}_i(0) \sim \pi_i^{\theta_i(k)}(\cdot \mid \mathbf{s}_i(0), \hat{\boldsymbol{\omega}}_i)$
21:         Receive reward $r_i(0) = r_i(\mathbf{s}_i(0), \mathbf{a}_i(0))$
22:         Initialize critic $\hat{Q}_i^0 \in \mathbb{R}^{\mathcal{S}_{\mathcal{N}_i^\kappa} \times \mathcal{A}_{\mathcal{N}_i^\kappa} \times \Omega_{\mathcal{N}_i^\kappa}}$ to zero
23:     **end for**
24:     **for** $t = 1$ to $T$ **do**
25:         **for** each agent $i$ **do**
26:             Get state $\mathbf{s}_i(t)$, take action $\mathbf{a}_i(t) \sim \pi_i^{\theta_i(k)}(\cdot \mid \mathbf{s}_i^\circ(t), \hat{\boldsymbol{\omega}}_i^\circ)$
27:            Get reward $r_i(t) = r_i(\mathbf{s}_i(t), \mathbf{a}_i(t))$
28:            $\alpha_{t-1} = \frac{h}{t-1+t_0}$
29:            Update TD target:

$$\hat{Q}_i^t(\mathbf{s}_{\mathcal{N}_i^\kappa}^\circ(t-1), \mathbf{a}_{\mathcal{N}_i^\kappa}(t-1), \hat{\boldsymbol{\omega}}_{\mathcal{N}_i^\kappa}^\circ) \leftarrow (1 - \alpha_{t-1})\hat{Q}_i^{t-1}(\mathbf{s}_{\mathcal{N}_i^\kappa}^\circ(t-1), \mathbf{a}_{\mathcal{N}_i^\kappa}(t-1), \hat{\boldsymbol{\omega}}_{\mathcal{N}_i^\kappa}^\circ)$$
$$+ \alpha_{t-1}\left(r_i(t-1) + \gamma \hat{Q}_i^{t-1}(\mathbf{s}_{\mathcal{N}_i^\kappa}^\circ(t), \mathbf{a}_{\mathcal{N}_i^\kappa}(t), \hat{\boldsymbol{\omega}}_{\mathcal{N}_i^\kappa}^\circ)\right)$$

30:            For all other $(\mathbf{s}_{\mathcal{N}_i^\kappa}^\circ, \mathbf{a}_{\mathcal{N}_i^\kappa}) \neq (\mathbf{s}_{\mathcal{N}_i^\kappa}^\circ(t-1), \mathbf{a}_{\mathcal{N}_i^\kappa}(t-1))$, set:

$$\hat{Q}_i^t(\mathbf{s}_{\mathcal{N}_i^\kappa}^\circ, \mathbf{a}_{\mathcal{N}_i^\kappa}, \hat{\boldsymbol{\omega}}_{\mathcal{N}_i^\kappa}^\circ) = \hat{Q}_i^{t-1}(\mathbf{s}_{\mathcal{N}_i^\kappa}^\circ, \mathbf{a}_{\mathcal{N}_i^\kappa}, \hat{\boldsymbol{\omega}}_{\mathcal{N}_i^\kappa}^\circ)$$

31:         **end for**
32:     **end for**
33:     **for** each agent $i$ **do**
34:         Estimate policy gradient:

$$\hat{g}_i(k) = \sum_{t=0}^{T} \gamma^t \cdot \frac{1}{n} \sum_{j \in \mathcal{N}_i^\kappa} \hat{Q}_j^T(\mathbf{s}_{\mathcal{N}_j^\kappa}^\circ(t), \mathbf{a}_{\mathcal{N}_j^\kappa}(t), \hat{\boldsymbol{\omega}}_{\mathcal{N}_j^\kappa}^\circ)\nabla_{\theta_i} \log \pi_i^{\theta_i(k)}(\mathbf{a}_i(t) \mid \mathbf{s}_i^\circ(t), \hat{\boldsymbol{\omega}}_i^\circ)$$

35:         Update policy: $\theta_i(k+1) = \theta_i(k) + \eta_k \hat{g}_i(k)$, where $\eta_k = \frac{\eta}{\sqrt{k+1}}$
36:     **end for**
37: **end for**
38: // *Phase 4: adaptation to new domain*
39: Collect few trajectories $\tau^{M+1} = \{(\mathbf{s}(t), \mathbf{a}(t), \mathbf{s}(t+1))\}_{t=0}^{T_a}$ in the new domain
40: **for** each agent $i$ **do**
41:     Infer domain factor $\hat{\boldsymbol{\omega}}_i^{M+1}$
42:     Deploy policy $\pi_i^{\theta(K)}(\cdot|\mathbf{s}_i^\circ, \hat{\boldsymbol{\omega}}_i^{M+1,\circ})$
43: **end for**

---

**Algorithm 5** GENERALIZABLE AND SCALABLE ACTOR-CRITIC WITHOUT ACR

---

1: **Input:** $\theta_i(0)$; parameter $\kappa$; $T$, length of each episode; stepsize parameters $h, t_0, \eta$.
2: *// Phase 1: domain estimation*
3: **for** source domain index $m = 1, 2, \ldots, M$ **do**
4:     Sample $\boldsymbol{\omega}^m \sim \mathcal{D}$
5:     Collect trajectories $\tau^m = (\mathbf{s}^m(t), \mathbf{a}^m(t), r^m(t), \mathbf{s}^m(t+1))_{t=1}^{T_e}$
6:     Split $\tau^m$ into $\tau_i^m = \left(\mathbf{s}_{\mathcal{N}_i}^m(t), \mathbf{a}_i^m(t), r_i(t), \mathbf{s}_i^m(t+1)\right)_{t=1}^{T_e}$ for each agent $i$
7:     Each agent $i$ estimate the domain factor $\hat{\boldsymbol{\omega}}_i^m$
8: **end for**
9: *// Phase 2: meta-learning*
10: **for** $k = 0, 1, 2, \ldots, K-1$ **do**
11:     Sample domain index $m(k)$ uniformly from $\{1, \ldots, K\}$
12:     Set $\hat{\boldsymbol{\omega}} = \hat{\boldsymbol{\omega}}^{m(k)}$                                                  ▷ Current domain factors
13:     Sample initial state $\mathbf{s}(0) \sim \rho_0$
14:     **for** each agent $i$ **do**
15:         Take action $\mathbf{a}_i(0) \sim \pi_i^{\theta_i(k)}(\cdot \mid \mathbf{s}_i(0), \hat{\boldsymbol{\omega}}_i)$
16:         Receive reward $r_i(0) = r_i(\mathbf{s}_i(0), \mathbf{a}_i(0))$
17:         Initialize critic $\hat{Q}_i^0 \in \mathbb{R}^{\mathcal{S}_{\mathcal{N}_i^\kappa} \times \mathcal{A}_{\mathcal{N}_i^\kappa} \times \Omega_{\mathcal{N}_i^\kappa}}$ to zero
18:     **end for**
19:     **for** $t = 1$ to $T$ **do**
20:         **for** each agent $i$ **do**
21:             Get state $\mathbf{s}_i(t)$, take action $\mathbf{a}_i(t) \sim \pi_i^{\theta_i(k)}(\cdot \mid \mathbf{s}_i(t), \hat{\boldsymbol{\omega}}_i)$
22:             Get reward $r_i(t) = r_i(\mathbf{s}_i(t), \mathbf{a}_i(t))$
23:             $\alpha_{t-1} = \frac{h}{t-1+t_0}$
24:             Update TD target:

$$\hat{Q}_i^t(\mathbf{s}_{\mathcal{N}_i^\kappa}(t-1), \mathbf{a}_{\mathcal{N}_i^\kappa}(t-1), \hat{\boldsymbol{\omega}}_{\mathcal{N}_i^\kappa}) \leftarrow (1 - \alpha_{t-1})\hat{Q}_i^{t-1}(\mathbf{s}_{\mathcal{N}_i^\kappa}(t-1), \mathbf{a}_{\mathcal{N}_i^\kappa}(t-1), \hat{\boldsymbol{\omega}}_{\mathcal{N}_i^\kappa})$$
$$+ \alpha_{t-1}\left(r_i(t-1) + \gamma\hat{Q}_i^{t-1}(\mathbf{s}_{\mathcal{N}_i^\kappa}(t), \mathbf{a}_{\mathcal{N}_i^\kappa}(t), \hat{\boldsymbol{\omega}}_{\mathcal{N}_i^\kappa})\right)$$

25:             For all other $(\mathbf{s}_{\mathcal{N}_i^\kappa}, \mathbf{a}_{\mathcal{N}_i^\kappa}) \neq (\mathbf{s}_{\mathcal{N}_i^\kappa}(t-1), \mathbf{a}_{\mathcal{N}_i^\kappa}(t-1))$, set:

$$\hat{Q}_i^t(\mathbf{s}_{\mathcal{N}_i^\kappa}, \mathbf{a}_{\mathcal{N}_i^\kappa}, \hat{\boldsymbol{\omega}}_{\mathcal{N}_i^\kappa}) = \hat{Q}_i^{t-1}(\mathbf{s}_{\mathcal{N}_i^\kappa}, \mathbf{a}_{\mathcal{N}_i^\kappa}, \hat{\boldsymbol{\omega}}_{\mathcal{N}_i^\kappa})$$

26:         **end for**
27:     **end for**
28:     **for** each agent $i$ **do**
29:         Estimate policy gradient:

$$\hat{g}_i(k) = \sum_{t=0}^{T} \gamma^t \cdot \frac{1}{n} \sum_{j \in \mathcal{N}_i^\kappa} \hat{Q}_j^T(\mathbf{s}_{\mathcal{N}_j^\kappa}(t), \mathbf{a}_{\mathcal{N}_j^\kappa}(t), \hat{\boldsymbol{\omega}}_{\mathcal{N}_j^\kappa})\nabla_{\theta_i} \log \pi_i^{\theta_i(k)}(\mathbf{a}_i(t) \mid \mathbf{s}_i(t), \hat{\boldsymbol{\omega}}_i^\circ)$$

30:         Update policy: $\theta_i(k+1) = \theta_i(k) + \eta_k\hat{g}_i(k)$, where $\eta_k = \frac{\eta}{\sqrt{k+1}}$
31:     **end for**
32: **end for**
33: *// Phase 3: adaptation to new domain*
34: Collect few trajectories $\tau^{M+1} = \{(\mathbf{s}(t), \mathbf{a}(t), \mathbf{s}(t+1))\}_{t=0}^{T_a}$ in the new domain
35: **for** each agent $i$ **do**
36:     Infer domain factor $\hat{\boldsymbol{\omega}}_i^{M+1}$
37:     Deploy policy $\pi_i^{\theta(K)}(\cdot | \mathbf{s}_i, \hat{\boldsymbol{\omega}}_i^{M+1})$
38: **end for**

---

*Proof of Proposition 1.* Denote $\mathbf{s} = \left(\mathbf{s}^\circ_{\mathcal{N}^\kappa_i}, \mathbf{s}/\mathbf{s}^\circ_{\mathcal{N}^\kappa_i}\right)$, $\mathbf{a} = \left(\mathbf{a}_{\mathcal{N}^\kappa_i}, \mathbf{a}_{\mathcal{N}^\kappa_{-i}}\right)$; $\mathbf{s}' = \left(\mathbf{s}^\circ_{\mathcal{N}^\kappa_i}, \mathbf{s}'/\mathbf{s}^\circ_{\mathcal{N}^\kappa_i}\right)$ and $\mathbf{a}' = \left(\mathbf{a}_{\mathcal{N}^\kappa_i}, \mathbf{a}'_{\mathcal{N}^\kappa_{-i}}\right)$. Let $\rho_{t,i}$ be the distribution of $(\mathbf{s}_i(t), \mathbf{a}_i(t))$ conditioned on $(\mathbf{s}(0), \mathbf{a}(0)) = (\mathbf{s}, \mathbf{a})$ under policy $\pi$, and let $\rho'_{t,i}$ be the distribution of $(\mathbf{s}_i(t), \mathbf{a}_i(t))$ conditioned on $(\mathbf{s}(0), \mathbf{a}(0)) = (\mathbf{s}', \mathbf{a}')$ under policy $\pi$.

Due to the local dependence structure, and the localized policy structure, $\rho_{t,i}$ only depends on $\left(\mathbf{s}_{\mathcal{N}^t_i}, \mathbf{a}_{\mathcal{N}^t_i}\right)$ which is the same as $\left(\mathbf{s}'_{\mathcal{N}^t_i}, \mathbf{a}'_{\mathcal{N}^t_i}\right)$ when $t \leq \kappa$. Furthermore, by the definition of ACR, we must have $\rho_{t,i} = \rho'_{t,i}$ for all $t \leq \kappa$ per the way the initial state $(\mathbf{s}, \mathbf{a})$, $(\mathbf{s}', \mathbf{a}')$ are chosen. With these definitions, we expand the definition of $Q^\pi_i$,

$$|Q^\pi_i(\mathbf{s}, \mathbf{a}) - Q^\pi_i(\mathbf{s}', \mathbf{a}')|$$

$$\leq \sum_{t=0}^\infty \left| \mathbb{E}\left[\gamma^t r_i(\mathbf{s}_i(t), \mathbf{a}_i(t)) \mid (\mathbf{s}(0), \mathbf{a}(0)) = (\mathbf{s}, \mathbf{a})\right] - \mathbb{E}\left[\gamma^t r_i(\mathbf{s}_i(t), \mathbf{a}_i(t)) \mid (\mathbf{s}(0), \mathbf{a}(0)) = (\mathbf{s}', \mathbf{a}')\right] \right|$$

$$= \sum_{t=0}^\infty \left| \gamma^t \mathbb{E}_{(\mathbf{s}_i, \mathbf{a}_i) \sim \rho_{t,i}} r_i(\mathbf{s}_i, \mathbf{a}_i) - \gamma^t \mathbb{E}_{(\mathbf{s}_i, \mathbf{a}_i) \sim \rho'_{t,i}} r_i(\mathbf{s}_i, \mathbf{a}_i) \right|$$

$$= \sum_{t=\kappa+1}^\infty \left| \gamma^t \mathbb{E}_{(\mathbf{s}_i, \mathbf{a}_i) \sim \rho_{t,i}} r_i(\mathbf{s}_i, \mathbf{a}_i) - \gamma^t \mathbb{E}_{(\mathbf{s}_i, \mathbf{a}_i) \sim \rho'_{t,i}} r_i(\mathbf{s}_i, \mathbf{a}_i) \right|$$

$$\leq \sum_{t=\kappa+1}^\infty \gamma^t \bar{r} \mathrm{TV}\left(\rho_{t,i}, \rho'_{t,i}\right) \leq \frac{\bar{r}}{1-\gamma} \gamma^{\kappa+1},$$

where $\mathrm{TV}\left(\rho_{t,i}, \rho'_{t,i}\right) \leq 1$ is the total variation distance between $\rho_{t,i}$ and $\rho'_{t,i}$. $\square$

### E.2   Policy ACR

We begin by presenting Proposition 6 and its proof, which establishes the guarantee of Algorithm 2, and then proceed to the proof of Proposition 2.

**Proposition 6** (Compactness of policy ACR). *Let $\pi$ be an arbitrary localized policy and $\tilde{\pi}$ be the corresponding approximately compact localized policy output by Algorithm 2, for each agent $i$, and any $(\mathbf{s}, \mathbf{a}) \in \mathcal{S} \times \mathcal{A}$, we have*

$$Q^{\tilde{\pi}}_i(\mathbf{s}, \mathbf{a}) = Q^\pi_i(\mathbf{s}, \mathbf{a}).$$

To prove Proposition 6, we start with some technical lemmas. Using the definition of policy ACR (Algorithm 2), we obtain Lemma 3.

**Lemma 3.** *Fix $i \in \mathcal{N}$. For any $j \in [d^s_i]$, $s_{i,j} \in \mathbf{s}^\circ_i$ if and only if either of the following holds*

- (i) $s_{i,j}$ has an edge to $r_i$, or

- (ii) there exists $i' \in \mathcal{N}$ such that there exists a path from $s_{i,j}(t)$ to $r_{i'}(t+l)$

  $$s_{i,j}(t) \to s_{i(t+1),j(t+1)}(t+1) \to s_{i(t+2),j(t+2)}(t+2) \cdots \to s_{i(t+l),j(t+l)}(t+l) \to r_{i'}(t+l),$$

  where $i(t+l) = i'$ and $i(t+m) \in \mathcal{N}_{i(t+m-1)}$ for any $m \in [l]$ and for some $l \geq 1$. Moreover, $s_{j(t+m),i(t+m)}(t+m) \in \mathbf{s}^\circ_{i(t+m)}$ for all $m \in [l]$.

*Proof of Lemma 3.* We prove this by induction on the iterations of the while loop in Algorithm 2.

**Base case:** After initialization (lines 2-8 of Algorithm 2), $\mathbf{s}^\circ_i$ contains exactly those state components $s_{i,j}$ that have a direct edge to $r_i$ (i.e., where $\mathbf{c}^{s \to r}_{i,j} = 1$). This corresponds precisely to condition (i).

**Inductive hypothesis:** Suppose that after $k$ iterations of the while loop, for any agent $i$ and component $j$, $s_{i,j} \in \mathbf{s}^\circ_i$ if and only if either:

- $s_{i,j}$ has a direct edge to $r_i$, or

- there exists a path from $s_{i,j}(t)$ to some $r_{i'}(t+m)$ where $m \leq k$, such that all intermediate state components are in their respective $\mathbf{s}^\circ$ sets, and each consecutive agent in the path is a neighbor of the previous agent.

**Inductive step:** Consider the $(k+1)$-th iteration. In this iteration, according to Algorithm 2 (lines 10-17), we add $s_{i,j}$ to $\mathbf{s}_i^\circ$ if there is an edge from $s_{i,j}$ to some component of $\mathbf{s}_{\mathcal{N}_i}^\circ$, where $\mathbf{s}_{\mathcal{N}_i}^\circ = \cup_{i' \in N_i} \mathbf{s}_{i'}^\circ$ contains all minimal components of $i$'s neighbors from the previous iterations.

By the inductive hypothesis, for any $i' \in \mathcal{N}_i$ and any component $j'$, $s_{i',j'} \in \mathbf{s}_{i'}^\circ$ if and only if either:

- $s_{i',j'}$ has a direct edge to $r_{i'}$, or

- there exists a path from $s_{i',j'}(t)$ to some $r_{i''}(t+m)$ where $m \leq k$, with all intermediate components in their respective $\mathbf{s}^\circ$ sets.

**Forward direction:** Suppose $s_{i,j}$ is added to $\mathbf{s}_i^\circ$ in the $(k+1)$-th iteration. Then there must be an edge from $s_{i,j}$ to some component $s_{i',j'} \in \mathbf{s}_{i'}^\circ$ for some $i' \in \mathcal{N}_i$. By the inductive hypothesis, either:

Case 1: If $s_{i',j'}$ has a direct edge to $r_{i'}$, then we have a path:

$$s_{i,j}(t) \to s_{i',j'}(t+1) \to r_{i'}(t+1),$$

which is a path of length 2 ($l = 1$) from $s_{i,j}(t)$ to $r_{i'}(t+1)$, satisfying condition (ii).

Case 2: If there is a path from $s_{i',j'}(t+1)$ to $r_{i''}(t+1+m)$ for some $m \leq k$, then we have a longer path:

$$s_{i,j}(t) \to s_{i',j'}(t+1) \to \cdots \to r_{i''}(t+1+m),$$

which is a path of length $m+2$ ($l = m+1 \leq k+1$) from $s_{i,j}(t)$ to $r_{i''}(t+1+m)$, satisfying condition (ii).

In both cases, $i' \in \mathcal{N}_i$ (the neighborhood constraint is satisfied), and all intermediate state components are in their respective $\mathbf{s}^\circ$ sets by the inductive hypothesis.

**Backward direction:** Conversely, suppose there exists a path from $s_{i,j}(t)$ to $r_{i'}(t+l)$ for some $l \leq k+1$, where all intermediate components are in their respective $\mathbf{s}^\circ$ sets and each agent is a neighbor of the previous agent.

If $l = 0$, then $s_{i,j}$ has a direct edge to $r_i$ (condition (i)), so $s_{i,j} \in \mathbf{s}_i^\circ$ after initialization.

If $l \geq 1$, consider the path:

$$s_{i,j}(t) \to s_{i(t+1),j(t+1)}(t+1) \to \cdots \to r_{i'}(t+l).$$

The first transition in this path is from $s_{i,j}(t)$ to $s_{i(t+1),j(t+1)}(t+1)$, where $i(t+1) \in N_i$ (by the neighborhood constraint). By the path condition, $s_{i(t+1),j(t+1)}(t+1) \in \mathbf{s}_{i(t+1)}^\circ$.

Furthermore, there exists a path from $s_{i(t+1),j(t+1)}(t+1)$ to $r_{i'}(t+l)$ of length $l-1 \leq k$. By the inductive hypothesis, $s_{i(t+1),j(t+1)} \in \mathbf{s}_{i(t+1)}^\circ$.

Since $i(t+1) \in \mathcal{N}_i$, we have $s_{i(t+1),j(t+1)} \in \mathbf{s}_{\mathcal{N}_i}^\circ$. Therefore, there is an edge from $s_{i,j}$ to a component in $\mathbf{s}_{\mathcal{N}_i}^\circ$, which means $s_{i,j}$ will be added to $\mathbf{s}_i^\circ$ in the $(k+1)$-th iteration (or earlier).

By induction, $s_{i,j} \in \mathbf{s}_i^\circ$ if and only if either condition (i) or condition (ii) holds, completing the proof. $\qed$

**Lemma 4.** Fix $i \in \mathcal{N}$. For any $j \in [d_i^s]$, $s_{i,j} \notin \mathbf{s}_i^\circ$, if and only if each of the following holds

- (i) $s_{i,j}$ does not have an edge to $r_i$

- (ii) for any $i' \in \mathcal{N}$, there exists no path from $s_{i,j}(t)$ to $r_{i'}(t+l)$ for some $l$

*Proof of Lemma 4.* This lemma is the logical contrapositive of Lemma 3. Let's formalize this relationship.

Lemma 3 states that $s_{i,j} \in \mathbf{s}_i^\circ$ if and only if either:

- $s_{i,j}$ has an edge to $r_i$, or

- There exists $i' \in \mathcal{N}$ and there exists a path from $s_{i,j}(t)$ to $r_{i'}(t+l)$ satisfying certain constraints.

The logical form of Lemma 3 can be written as: $s_{i,j} \in \mathbf{s}_i^{\circ} \iff (A) \vee (B)$.

The contrapositive of this statement is: $s_{i,j} \notin \mathbf{s}_i^{\circ} \iff \neg((A) \vee (B))$.

Using De Morgan's laws, $\neg((A) \vee (B)) \iff \neg(A) \wedge \neg(B)$, which gives us:

$s_{i,j} \notin \mathbf{s}_i^{\circ} \iff \neg(A) \wedge \neg(B)$, where:

- $\neg(A)$: $s_{i,j}$ does not have an edge to $r_i$

- $\neg(B)$: For all $i' \in \mathcal{N}$ and for all $l \geq 1$, there exists no path from $s_{i,j}(t)$ to $r_{i'}(t+l)$ that satisfies the constraints on intermediate nodes.

These are precisely the conditions (i) and (ii) in Lemma 4. Therefore, by the logical equivalence of a statement and its contrapositive, Lemma 4 is proved. $\qquad \square$

*Proof of Proposition 6.* We will prove that the $Q$-functions are identical by showing that the distribution of rewards collected under both policies is identical. This follows if we can establish that the relevant state-action trajectories have the same distribution under both policies.

Let us define:

- $\rho_t^{\pi}(\mathbf{s}(t), \mathbf{a}(t) | \mathbf{s}(0) = \mathbf{s}, \mathbf{a}(0) = \mathbf{a})$: The probability of being in state-action pair $(\mathbf{s}(t), \mathbf{a}(t))$ at time $t$, starting from $(\mathbf{s}(0) = \mathbf{s}, \mathbf{a}(0) = \mathbf{a})$ and following policy $\pi$.

- $r_i(\mathbf{s}_i, \mathbf{a}_i)$: The reward for agent $i$ given its state-action pair.

The $Q$-function for agent $i$ under policy $\pi$ can be expressed as:

$$Q_i^{\pi}(\mathbf{s}, \mathbf{a}) = \mathbb{E}_{\pi} \left[ \sum_{t=0}^{\infty} \gamma^t r_i(\mathbf{s}_i(t), \mathbf{a}_i(t)) \mid \mathbf{s}(0) = \mathbf{s}, \mathbf{a}(0) = \mathbf{a} \right]$$

$$= \sum_{t=0}^{\infty} \gamma^t \sum_{\mathbf{s}(t), \mathbf{a}(t)} \rho_t^{\pi}(\mathbf{s}(t), \mathbf{a}_t | \mathbf{s}(0) = \mathbf{s}, \mathbf{a}(0) = \mathbf{a}) \cdot r_i(\mathbf{s}_i(t), \mathbf{a}_i(t)).$$

Similarly, we can define $Q_i^{\tilde{\pi}}(\mathbf{s}, \mathbf{a})$ for the approximately compact policy $\tilde{\pi}$.

To prove $Q_i^{\tilde{\pi}}(\mathbf{s}, \mathbf{a}) = Q_i^{\pi}(\mathbf{s}, \mathbf{a})$, it suffices to show that for all $t \geq 0$, the distribution of rewards $r_i(\mathbf{s}_i(t), \mathbf{a}_i(t))$ is identical under both policies.

From Lemma 3, we know that the reward $r_i(\mathbf{s}_i, \mathbf{a}_i)$ depends only on the components in $\mathbf{s}_i^{\circ}$ and $a_i$. This is because any state component not in $\mathbf{s}_i^{\circ}$ has no path to $r_i$ (directly or indirectly), meaning it cannot influence the reward function. Therefore, we need to prove that the joint distribution of $(\mathbf{s}_i^{\circ}(t), \mathbf{a}_i(t))$ is the same under both $\pi$ and $\tilde{\pi}$.

We prove this by induction on $t$.

**Base case** ($t = 0$): The initial state $\mathbf{s}(0) = \mathbf{s}$ is given, so $\mathbf{s}_i^{\circ}(0)$ is the same for both policies. For the initial action, we have:

Under $\pi$: $\mathbf{a}_i(0) \sim \pi_i(\cdot | \mathbf{s}_{\mathcal{N}_i}(0))$

Under $\tilde{\pi}$: $\mathbf{a}_i(0) \sim \tilde{\pi}_i(\cdot | \mathbf{s}_{\mathcal{N}_i}^{\circ}(0))$

By the definition of the approximately compact localized policy:

$$\tilde{\pi}_i(\cdot | \mathbf{s}_{\mathcal{N}_i}^{\circ}) = \pi_i(\cdot | \mathbf{s}_{\mathcal{N}_i}^{\circ}, \mathbf{s}_{\mathcal{N}_i} / \mathbf{s}_{\mathcal{N}_i}^{\circ})$$

for any fixed $\mathbf{s}_{\mathcal{N}_i}^{\circ}$ and for any configuration of the non-minimal components $\mathbf{s}_{\mathcal{N}_i} / \mathbf{s}_{\mathcal{N}_i}^{\circ}$. Therefore, $\mathbf{a}_i(0)$ has the same distribution under both policies when conditioned on the same initial state.

**Inductive hypothesis**: Assume that for some $t \geq 0$, the joint distribution of $(\mathbf{s}_i^{\circ}(t), \mathbf{a}_i(t))$ is the same under both $\pi$ and $\tilde{\pi}$ for all agents $i$.

**Inductive step**: We need to show that the joint distribution of $(\mathbf{s}_i^{\circ}(t+1), \mathbf{a}_i(t+1))$ is also the same under both policies.

For $\mathbf{s}_i^\circ(t+1)$, we analyze how it depends on the previous state and action. By the Markov assumption, $\mathbf{s}(t+1)$ depends only on $\mathbf{s}(t)$ and $\mathbf{a}(t)$. The key insight is to show that $\mathbf{s}_i^\circ(t+1)$ depends only on $\mathbf{s}_{\mathcal{N}_i}^\circ(t)$ and $\mathbf{a}_{\mathcal{N}_i}(t)$, not on any component outside $\mathbf{s}_{\mathcal{N}_i}^\circ(t)$.

Suppose, for contradiction, that there exists a component $s_{k,j}(t) \notin \mathbf{s}_{\mathcal{N}_i}^\circ(t)$ that influences some component $s_{i,j'}(t+1) \in \mathbf{s}_i^\circ(t+1)$. This means there is a causal link from $s_{k,j}(t)$ to $s_{i,j'}(t+1)$.

By Lemma 3, since $s_{i,j'}(t+1) \in \mathbf{s}_i^\circ(t+1)$, either:

- $s_{i,j'}(t+1)$ has a direct edge to $r_i$, or

- There exists a path from $s_{i,j'}(t+1)$ to some reward $r_{i'}(t+1+l)$ for some $l \geq 0$.

Either way, this creates a path from $s_{k,j}(t)$ to a reward (either directly to $r_i$ through $s_{i,j'}(t+1)$, or to $r_{i'}(t+1+l)$ via the path from $s_{i,j'}(t+1)$).

This path satisfies the constraints in Lemma 3 condition (ii), which would imply $s_{k,j}(t) \in \mathbf{s}_k^\circ(t)$. Since $k \in \mathcal{N}_i$ (there is a causal link from agent $k$ to agent $i$), we would have $s_{k,j}(t) \in \mathbf{s}_{\mathcal{N}_i}^\circ(t)$, contradicting our assumption.

Therefore, $\mathbf{s}_i^\circ(t+1)$ depends only on $\mathbf{s}_{\mathcal{N}_i}^\circ(t)$ and $\mathbf{a}_{\mathcal{N}_i}(t)$. By the inductive hypothesis, the joint distribution of $(\mathbf{s}_{\mathcal{N}_i}^\circ(t), \mathbf{a}_{\mathcal{N}_i}(t))$ is the same under both policies, so the distribution of $\mathbf{s}_i^\circ(t+1)$ is also the same.

For $\mathbf{a}_i(t+1)$, we have:

Under $\pi$: $\mathbf{a}_i(t+1) \sim \pi_i(\cdot|\mathbf{s}_{\mathcal{N}_i}(t+1))$

Under $\tilde{\pi}$: $\mathbf{a}_i(t+1) \sim \tilde{\pi}_i(\cdot|\mathbf{s}_{\mathcal{N}_i}^\circ(t+1))$

By the definition of approximately compact policy and the fact that the distribution of $\mathbf{s}_{\mathcal{N}_i}^\circ(t+1)$ is the same under both policies, $\mathbf{a}_i(t+1)$ also has the same distribution under both policies.

By induction, for all $t \geq 0$, the joint distribution of $(\mathbf{s}_i^\circ(t), \mathbf{a}_i(t))$ is the same under both $\pi$ and $\tilde{\pi}$ for all agents $i$.

Since the reward $r_i(\mathbf{s}_i(t), \mathbf{a}_i(t))$ depends only on $\mathbf{s}_i^\circ(t)$ and $\mathbf{a}_i(t)$, and these have the same distribution under both policies, the expected discounted sum of rewards is also the same.

Therefore, $Q_i^{\tilde{\pi}}(\mathbf{s}, \mathbf{a}) = Q_i^\pi(\mathbf{s}, \mathbf{a})$ for all agents $i$ and all state-action pairs $(\mathbf{s}, \mathbf{a})$. $\qquad\square$

**Corollary 1.** Let $\pi$ be an arbitrary localized policy and $\tilde{\pi}$ be the corresponding approximately compact localized policy output by Algorithm 2, for each agent $i$, and any $(\mathbf{s}, \mathbf{a}) \in \mathcal{S} \times \mathcal{A}$, the approximation error between $\tilde{Q}_i^{\tilde{\pi}}\left(\mathbf{s}_{\mathcal{N}_i^\kappa}^\circ, \mathbf{a}_{\mathcal{N}_i^\kappa}\right)$ and $Q_i^\pi(\mathbf{s}, \mathbf{a})$ can be bounded as

$$\sup_{(\mathbf{s},\mathbf{a})\in\mathcal{S}\times\mathcal{A}} \left|\tilde{Q}_i^{\tilde{\pi}}\left(\mathbf{s}_{\mathcal{N}_i^\kappa}^\circ, \mathbf{a}_{\mathcal{N}_i^\kappa}\right) - Q_i^\pi(\mathbf{s},\mathbf{a})\right| \leq \frac{\bar{r}}{1-\gamma}\gamma^{\kappa+1}.$$

*Proof.* Using Proposition 6, we have

$$\left|\tilde{Q}_i^{\tilde{\pi}}\left(\mathbf{s}_{\mathcal{N}_i^\kappa}^\circ, \mathbf{a}_{\mathcal{N}_i^\kappa}\right) - Q_i^\pi(\mathbf{s},\mathbf{a})\right| \leq \left|\tilde{Q}_i^{\tilde{\pi}}\left(\mathbf{s}_{\mathcal{N}_i^\kappa}^\circ, \mathbf{a}_{\mathcal{N}_i^\kappa}\right) - Q_i^{\tilde{\pi}}(\mathbf{s},\mathbf{a})\right| + \left|Q_i^{\tilde{\pi}}(\mathbf{s},\mathbf{a}) - Q_i^\pi(\mathbf{s},\mathbf{a})\right|$$
$$\leq \frac{\bar{r}}{1-\gamma}\gamma^{\kappa+1}.$$

$\qquad\square$

**Proposition 7** (Approximation error of policy ACR). For any policy $\pi$, let $\tilde{\pi}$ be the corresponding approximately compact policy constructed using Algorithm 3, for any $i$, we have

$$\sup_{(\mathbf{s},\mathbf{a})\in\mathcal{S}\times\mathcal{A}} \left|Q_i^{\tilde{\pi}}(\mathbf{s},\mathbf{a}) - Q_i^\pi(\mathbf{s},\mathbf{a})\right| \leq \frac{\bar{r}}{1-\gamma}\gamma^{\kappa+1}.$$

*Proof of Proposition 7.* Let us first establish precise definitions:

- Let $\mathbf{s}_i^{\circ,\infty}$ denote the minimal state representation for agent $i$ obtained from Algorithm 2 (with indefinite recursion until convergence).

- Let $\mathbf{s}_i^{\circ,\kappa}$ denote the minimal state representation for agent $i$ obtained from Algorithm 3 (with exactly $\kappa$ steps).

- Let $\pi$ be the original localized (1-hop) policy.

- Let $\tilde{\pi}^\infty$ be the approximately compact policy derived from Algorithm 2.

- Let $\tilde{\pi}^\kappa$ be the approximately compact policy derived from Algorithm 3 with $\kappa$ steps.

**Relationship between minimal representations:** By construction of the algorithms, we have $\mathbf{s}_i^{\circ,\kappa} \subseteq \mathbf{s}_i^{\circ,\infty}$ for any finite $\kappa$. This is because Algorithm 3 performs exactly $\kappa$ iterations, while Algorithm 2 continues until convergence, which may require more than $\kappa$ iterations.

**Lemma 5.** For any agent $i$, $\mathbf{s}_i^{\circ,\kappa}$ captures all state components that can influence rewards within $\kappa$ time steps.

*Proof.* By the structure of Algorithm 3, in step 1, $\mathbf{s}_i^\circ(1)$ includes all state components that directly affect the reward $r_i$. In each subsequent step $l$ (for $l = 1, 2, ..., \kappa - 1$), any state component that can affect a component in $\mathbf{s}_i^\circ(l)$ within one time step is added to $\mathbf{s}_i^\circ(l + 1)$. By induction, after $\kappa$ steps, $\mathbf{s}_i^\circ(\kappa)$ contains all state components that can affect the reward within $\kappa$ time steps. $\square$

**Policy equivalence and divergence:** From Proposition 6, we know that for the indefinite recursion case:

$$Q_i^{\tilde{\pi}^\infty}(\mathbf{s}, \mathbf{a}) = Q_i^\pi(\mathbf{s}, \mathbf{a}).$$

Therefore, to prove Proposition 7, we need to show:

$$\sup_{(\mathbf{s},\mathbf{a}) \in S \times A} |Q_i^{\tilde{\pi}^\kappa}(\mathbf{s}, \mathbf{a}) - Q_i^{\tilde{\pi}^\infty}(\mathbf{s}, \mathbf{a})| \leq \frac{\bar{r}}{1 - \gamma} \gamma^{\kappa+1}.$$

**Trajectory analysis:** For any initial state-action pair $(\mathbf{s}, \mathbf{a})$, we'll analyze the difference in trajectories under policies $\tilde{\pi}^\kappa$ and $\tilde{\pi}^\infty$.

**Lemma 6.** For any initial state $\mathbf{s}(0) = \mathbf{s}$ and action $\mathbf{a}(0) = \mathbf{a}$, the distribution of states and actions under policies $\tilde{\pi}^\kappa$ and $\tilde{\pi}^\infty$ are identical for the first $\kappa$ time steps, i.e., for $t = 0, 1, ..., \kappa$.

*Proof.* We prove this by induction.

*Base case* $(t = 0)$: By definition, $\mathbf{s}(0) = \mathbf{s}$ and $\mathbf{a}(0) = \mathbf{a}$ for both policies.

*Inductive step*: Assume the distributions are identical for time steps $0, 1, ..., t$ where $t < \kappa$. At time $t$, given identical state distributions, both policies make decisions based on the minimal state representations:

- $\tilde{\pi}^\kappa$ uses $\mathbf{s}_{\mathcal{N}_i}^{\circ,\kappa}(t)$

- $\tilde{\pi}^\infty$ uses $\mathbf{s}_{\mathcal{N}_i}^{\circ,\infty}(t)$

For any component $s_{i,j}(t)$ that affects rewards within $\kappa - t$ time steps, by Lemma 5, $s_{i,j}(t) \in \mathbf{s}_i^{\circ,\kappa}(t)$. Since $t < \kappa$, both policies have identical information about all state components that can affect rewards within the remaining time steps. Therefore, the action distributions at time $t$ are identical, which leads to identical state distributions at time $t + 1$. $\square$

**$Q$-function decomposition:** For any initial state-action pair $(\mathbf{s}, \mathbf{a})$, we can decompose the $Q$-functions as follows:

$$Q_i^{\tilde{\pi}^\kappa}(\mathbf{s}, \mathbf{a}) = \mathbb{E}_{\tilde{\pi}^\kappa}\left[\sum_{t=0}^{\kappa} \gamma^t r_i(\mathbf{s}_i(t), \mathbf{a}_i(t)) \mid \mathbf{s}(0) = \mathbf{s}, \mathbf{a}(0) = \mathbf{a}\right]$$

$$+ \mathbb{E}_{\tilde{\pi}^\kappa}\left[\sum_{t=\kappa+1}^{\infty} \gamma^t r_i(\mathbf{s}_i(t), \mathbf{a}_i(t)) \mid \mathbf{s}(0) = \mathbf{s}, \mathbf{a}(0) = \mathbf{a}\right]$$

$$Q_i^{\tilde{\pi}^\infty}(\mathbf{s}, \mathbf{a}) = \mathbb{E}_{\tilde{\pi}^\infty}\left[\sum_{t=0}^{\kappa} \gamma^t r_i(\mathbf{s}_i(t), \mathbf{a}_i(t)) \mid \mathbf{s}(0) = \mathbf{s}, \mathbf{a}(0) = \mathbf{a}\right]$$

$$+ \mathbb{E}_{\tilde{\pi}^\infty}\left[\sum_{t=\kappa+1}^{\infty} \gamma^t r_i(\mathbf{s}_i(t), \mathbf{a}_i(t)) \mid \mathbf{s}(0) = \mathbf{s}, \mathbf{a}(0) = \mathbf{a}\right].$$

By Lemma 2, the first terms are identical. Therefore:

$$|Q_i^{\tilde{\pi}^\kappa}(\mathbf{s}, \mathbf{a}) - Q_i^{\tilde{\pi}^\infty}(\mathbf{s}, \mathbf{a})| = \left|\mathbb{E}_{\tilde{\pi}^\kappa}\left[\sum_{t=\kappa+1}^{\infty} \gamma^t r_i(\mathbf{s}_i(t), \mathbf{a}_i(t)) \mid \mathbf{s}(0) = \mathbf{s}, \mathbf{a}(0) = \mathbf{a}\right]\right.$$

$$\left. - \mathbb{E}_{\tilde{\pi}^\infty}\left[\sum_{t=\kappa+1}^{\infty} \gamma^t r_i(\mathbf{s}_i(t), \mathbf{a}_i(t)) \mid \mathbf{s}(0) = \mathbf{s}, \mathbf{a}(0) = \mathbf{a}\right]\right|.$$

**Error bound:** Let $\rho_t^{\tilde{\pi}^\kappa}$ and $\rho_t^{\tilde{\pi}^\infty}$ be the distributions of $(\mathbf{s}_i(t), \mathbf{a}_i(t))$ under policies $\tilde{\pi}^\kappa$ and $\tilde{\pi}^\infty$ respectively, conditioned on $(\mathbf{s}(0), \mathbf{a}(0)) = (\mathbf{s}, \mathbf{a})$.

From Lemma 6, we know that $\rho_t^{\tilde{\pi}^\kappa} = \rho_t^{\tilde{\pi}^\infty}$ for $t \leq \kappa$.

For $t > \kappa$, we can bound the total variation distance between these distributions:

$$\text{TV}(\rho_t^{\tilde{\pi}^\kappa}, \rho_t^{\tilde{\pi}^\infty}) \leq 1.$$

Therefore:

$$|Q_i^{\tilde{\pi}^\kappa}(\mathbf{s}, \mathbf{a}) - Q_i^{\tilde{\pi}^\infty}(\mathbf{s}, \mathbf{a})| = \left|\sum_{t=\kappa+1}^{\infty} \gamma^t \left(\mathbb{E}_{(\mathbf{s}_i, \mathbf{a}_i) \sim \rho_t^{\tilde{\pi}^\kappa}}[r_i(\mathbf{s}_i, \mathbf{a}_i)] - \mathbb{E}_{(s_i, a_i) \sim \rho_t^{\tilde{\pi}^\infty}}[r_i(\mathbf{s}_i, \mathbf{a}_i)]\right)\right|$$

$$\leq \sum_{t=\kappa+1}^{\infty} \gamma^t \bar{r} \cdot \text{TV}(\rho_t^{\tilde{\pi}^\kappa}, \rho_t^{\tilde{\pi}^\infty})$$

$$\leq \sum_{t=\kappa+1}^{\infty} \gamma^t \bar{r} = \bar{r}\gamma^{\kappa+1}\frac{1}{1-\gamma}.$$

This gives us the desired bound:

$$\sup_{(\mathbf{s}, \mathbf{a}) \in S \times A} |Q_i^{\tilde{\pi}^\kappa}(\mathbf{s}, \mathbf{a}) - Q_i^{\tilde{\pi}^\infty}(\mathbf{s}, \mathbf{a})| \leq \frac{\bar{r}}{1-\gamma}\gamma^{\kappa+1}.$$

Since $Q_i^{\tilde{\pi}^\infty}(\mathbf{s}, \mathbf{a}) = Q_i^{\pi}(\mathbf{s}, \mathbf{a})$ from Proposition 6, we have:

$$\sup_{(\mathbf{s}, \mathbf{a}) \in S \times A} |Q_i^{\tilde{\pi}^\kappa}(\mathbf{s}, \mathbf{a}) - Q_i^{\pi}(\mathbf{s}, \mathbf{a})| \leq \frac{\bar{r}}{1-\gamma}\gamma^{\kappa+1}.$$

This completes the proof of Proposition 7. $\square$

*Proof of Proposition 2.* Using Proposition 7, we have

$$\left|\tilde{Q}_i^{\tilde{\pi}}\left(\mathbf{s}_{\mathcal{N}_i^\kappa}^\circ, \mathbf{a}_{\mathcal{N}_i^\kappa}\right) - Q_i^{\pi}(\mathbf{s}, \mathbf{a})\right| \leq \left|\tilde{Q}_i^{\tilde{\pi}}\left(\mathbf{s}_{\mathcal{N}_i^\kappa}^\circ, \mathbf{a}_{\mathcal{N}_i^\kappa}\right) - Q_i^{\tilde{\pi}}(\mathbf{s}, \mathbf{a})\right| + \left|Q_i^{\tilde{\pi}}(\mathbf{s}, \mathbf{a}) - Q_i^{\pi}(\mathbf{s}, \mathbf{a})\right|$$

$$\leq \frac{3\bar{r}}{1-\gamma}\gamma^{\kappa+1}.$$

$\square$

### E.3 Cross domain

Denote by $\boldsymbol{\omega}_{\mathcal{N}_i^\kappa} := (\boldsymbol{\omega}_i)_{i \in \mathcal{N}_i^\kappa}$ and $\boldsymbol{\omega}_{\mathcal{N}_{-i}^\kappa} := \boldsymbol{\omega}/\boldsymbol{\omega}_{\mathcal{N}_i^\kappa}$.

**Lemma 7** (Exponential decay property of domain-specific value functions). For any localized policy $\pi$, for any $i \in \mathcal{N}, \mathbf{s}_{\mathcal{N}_i^\kappa} \in \mathcal{S}_{\mathcal{N}_i^\kappa}, \mathbf{s}_{\mathcal{N}_{-i}^\kappa}, \mathbf{s}'_{\mathcal{N}_{-i}^\kappa} \in \mathcal{S}_{\mathcal{N}_{-i}^\kappa}, \mathbf{a}_{\mathcal{N}_i^\kappa} \in \mathcal{A}_{\mathcal{N}_i^\kappa}, \mathbf{a}_{\mathcal{N}_{-i}^\kappa}, \mathbf{a}'_{\mathcal{N}_{-i}^\kappa} \in \mathcal{A}_{\mathcal{N}_{-i}^\kappa}, \boldsymbol{\omega}_{\mathcal{N}_i^\kappa} \in \Omega_{\mathcal{N}_i^\kappa}, \boldsymbol{\omega}_{\mathcal{N}_{-i}^\kappa}, \boldsymbol{\omega}'_{\mathcal{N}_{-i}^\kappa} \in \Omega_{\mathcal{N}_{-i}^\kappa}$, the following holds

$$\left| Q_i^\pi \left( \mathbf{s}_{\mathcal{N}_i^\kappa}, \mathbf{s}_{\mathcal{N}_{-i}^\kappa}, \mathbf{a}_{\mathcal{N}_i^\kappa}, \mathbf{a}_{\mathcal{N}_{-i}^\kappa}, \boldsymbol{\omega}_{\mathcal{N}_i^\kappa}, \boldsymbol{\omega}_{\mathcal{N}_{-i}^\kappa} \right) - Q_i^\pi \left( \mathbf{s}_{\mathcal{N}_i^\kappa}, \mathbf{s}'_{\mathcal{N}_{-i}^\kappa}, \mathbf{a}_{\mathcal{N}_i^\kappa}, \mathbf{a}'_{\mathcal{N}_{-i}^\kappa}, \boldsymbol{\omega}_{\mathcal{N}_i^\kappa}, \boldsymbol{\omega}'_{\mathcal{N}_{-i}^\kappa} \right) \right| \le \frac{\bar{r}}{1 - \gamma} \gamma^\kappa.$$

*Proof.* The proof follows from that of Lemma 1 by augmenting the state space of each agent by $\mathcal{S}_i \times \Omega_i \ni (\mathbf{s}_i, \boldsymbol{\omega}_i)$, where the augmented state variable $\boldsymbol{\omega}_i$ degenerates to a deterministic state $(\boldsymbol{\omega}_i(t) = \boldsymbol{\omega}_i, \forall t)$. $\qquad\square$

#### E.3.1 Value functions

Given the generative model (Equation 6), we define $\boldsymbol{\omega}$-ACR for $Q$-functions as follows.

**Definition 5** (Value $\boldsymbol{\omega}$-ACR). Given the graphical representation of the networked system model $\mathcal{G}$ that is encoded in the binary masks $c^{\cdot \rightarrow \cdot}$, the value ACR $\mathbf{s}_{\mathcal{N}_i^\kappa}^\circ$ (Definition 1), for each agent $i$ and its $\kappa$-hop neighborhood $\mathcal{N}_i^\kappa$, we recursively define **domain-specific ACR** $\boldsymbol{\omega}_{\mathcal{N}_i^\kappa}^\circ$: the latent change factors $\boldsymbol{\omega}_{\mathcal{N}_i^\kappa}$ that either

- $\boldsymbol{\omega}_{i,j}$ has a direct link to agent $i$'s reward $r_i$, i.e., $\boldsymbol{\omega}_{i,j} = \boldsymbol{\omega}_i^r$ and $\mathbf{c}_{i,j}^{\boldsymbol{\omega} \rightarrow r_i} = 1$, or
- $\boldsymbol{\omega}_{i',j}$ has an edge to a state component $s_{i',l} \in \mathbf{s}_{\mathcal{N}_i^\kappa}^\circ$, i.e., $\mathbf{c}_{i',j,l}^{\boldsymbol{\omega} \rightarrow s} = 1$ and $s_{i',l} \in \mathbf{s}_{\mathcal{N}_i^\kappa}^\circ$.

Definition 5 also defines agent-wise ACR: $\cup_{i' \in \mathcal{N}_i^\kappa} \boldsymbol{\omega}_{i'}^\circ = \boldsymbol{\omega}_{\mathcal{N}_i^\kappa}^\circ$.

**Definition 6** (Approximately compact $\boldsymbol{\omega}$-conditioned $Q$-function). Fix $i \in \mathcal{N}$ and an arbitrary $\bar{s}_{\mathcal{N}_i^\kappa}$ and $\bar{\boldsymbol{\omega}}_{\mathcal{N}_i^\kappa}$. Define the approximately compact $\boldsymbol{\omega}$-conditioned $Q$-function as

$$\tilde{Q}_i^\pi \left( \mathbf{s}_{\mathcal{N}_i^\kappa}^\circ, \mathbf{a}_{\mathcal{N}_i^\kappa}, \boldsymbol{\omega}_{\mathcal{N}_i^\kappa}^\circ \right) := \hat{Q}_i^\pi \left( \mathbf{s}_{\mathcal{N}_i^\kappa}^\circ, \bar{\mathbf{s}}_{\mathcal{N}_i^\kappa}/\mathbf{s}_{\mathcal{N}_i^\kappa}^\circ, \mathbf{a}_{\mathcal{N}_i^\kappa}, \boldsymbol{\omega}_{\mathcal{N}_i^\kappa}^\circ, \bar{\boldsymbol{\omega}}_{\mathcal{N}_i^\kappa}/\boldsymbol{\omega}_{\mathcal{N}_i^\kappa}^\circ \right).$$

**Proposition 8** (Approximation error of value ACR). For each agent $i$, the approximation error between $\tilde{Q}_i^\pi \left( \mathbf{s}_{\mathcal{N}_i^\kappa}^\circ, \mathbf{a}_{\mathcal{N}_i^\kappa}, \boldsymbol{\omega}_{\mathcal{N}_i^\kappa}^\circ \right)$ and $Q_i^\pi (\mathbf{s}, \mathbf{a}, \boldsymbol{\omega})$ can be bounded as

$$\sup_{(\mathbf{s}, \mathbf{a}) \in \mathcal{S} \times \mathcal{A}} \left| \tilde{Q}_i^\pi \left( \mathbf{s}_{\mathcal{N}_i^\kappa}^\circ, \mathbf{a}_{\mathcal{N}_i^\kappa}, \boldsymbol{\omega}_{\mathcal{N}_i^\kappa}^\circ \right) - Q_i^\pi (\mathbf{s}, \mathbf{a}, \boldsymbol{\omega}) \right| \le \frac{2\bar{r}}{1 - \gamma} \gamma^{\kappa + 1}.$$

*Proof.* The proof follows from that of Proposition 1 by augmenting the state space of each agent by $\mathcal{S}_i \times \Omega_i \ni (\mathbf{s}_i, \boldsymbol{\omega}_i)$, where the augmented state variable $\boldsymbol{\omega}_i$ degenerates to a deterministic state $(\boldsymbol{\omega}_i(t) = \boldsymbol{\omega}_i, \forall t)$. $\qquad\square$

#### E.3.2 Policies

**Definition 7** ($\boldsymbol{\omega}$-ACR for localized policies). Given the graphical representation of the networked system model $\mathcal{G}$ that is encoded in the binary masks $\mathbf{c}$, the policy ACR $\mathbf{s}_{\mathcal{N}_i}^\circ$ obtained from Algorithm 2 or Algorithm 3, for each agent $i$ and its 1-hop neighborhood $\mathcal{N}_i$, we recursively define **domain-specific ACR** $\boldsymbol{\omega}_{\mathcal{N}_i}^\circ$: the latent factors $\boldsymbol{\omega}_{\mathcal{N}_i}$ that either

- $\boldsymbol{\omega}_{i,j}$ has a direct link to agent $i$'s reward $r_i$, i.e., $\boldsymbol{\omega}_{i,j} = \boldsymbol{\omega}_i^r$ and $\mathbf{c}_{i,j}^{\boldsymbol{\omega} \rightarrow r_i} = 1$, or
- $\boldsymbol{\omega}_{i',j}$ has an edge to a state component $s_{i',l} \in \mathbf{s}_{\mathcal{N}_i}^\circ$, i.e., $\mathbf{c}_{i',j,l}^{\boldsymbol{\omega} \rightarrow s} = 1$ and $s_{i',l} \in \mathbf{s}_{\mathcal{N}_i}^\circ$.

**Definition 8** (Approximately compact $\boldsymbol{\omega}$-conditioned policy). A domain-specific localized policy $\pi = (\pi_1, \dots, \pi_n)$ is called an approximately compact localized policy if the following holds for all $i \in \mathcal{N}, \mathbf{s}_{\mathcal{N}_i}, \mathbf{s}'_{\mathcal{N}_i} \in \mathcal{S}_{\mathcal{N}_i}, \boldsymbol{\omega}_{\mathcal{N}_i}, \boldsymbol{\omega}'_{\mathcal{N}_i}$:

$$\pi_i \left( \cdot \mid \mathbf{s}_{\mathcal{N}_i}^\circ, \mathbf{s}_{\mathcal{N}_i}/\mathbf{s}_{\mathcal{N}_i}^\circ, \boldsymbol{\omega}_{\mathcal{N}_i}^\circ, \boldsymbol{\omega}_{\mathcal{N}_i}/\boldsymbol{\omega}_{\mathcal{N}_i}^\circ \right) = \pi_i \left( \cdot \mid \mathbf{s}_{\mathcal{N}_i}^\circ, \mathbf{s}'_{\mathcal{N}_i}/\mathbf{s}_{\mathcal{N}_i}^\circ, \boldsymbol{\omega}_{\mathcal{N}_i}^\circ, \boldsymbol{\omega}'_{\mathcal{N}_i}/\boldsymbol{\omega}_{\mathcal{N}_i}^\circ \right).$$

In other words, there exists $\tilde{\pi}_i : \mathcal{S}^{\circ}_{\mathcal{N}_i} \times \Omega^{\circ}_{\mathcal{N}_i} \mapsto \Delta_{\mathcal{A}_i}$ such that $\pi_i\left(\cdot \mid \mathbf{s}^{\circ}_{\mathcal{N}_i}, \mathbf{s}_{\mathcal{N}_i}/\mathbf{s}^{\circ}_{\mathcal{N}_i}, \boldsymbol{\omega}^{\circ}_{\mathcal{N}_i}, \boldsymbol{\omega}_{\mathcal{N}_i}/\boldsymbol{\omega}^{\circ}_{\mathcal{N}_i}\right) = \tilde{\pi}_i\left(\cdot \mid \mathbf{s}^{\circ}_{\mathcal{N}_i}, \boldsymbol{\omega}^{\circ}_{\mathcal{N}_i}\right)$. For convenience, we fix an arbitrary $\mathbf{s}_{\mathcal{N}_i}/\mathbf{s}^{\circ}_{\mathcal{N}_i}, \boldsymbol{\omega}_{\mathcal{N}_i}/\boldsymbol{\omega}^{\circ}_{\mathcal{N}_i}$, and denote by $\tilde{\pi}_i\left(\cdot \mid \mathbf{s}^{\circ}_{\mathcal{N}_i}, \boldsymbol{\omega}^{\circ}_{\mathcal{N}_i}\right) := \pi_i\left(\cdot \mid \mathbf{s}^{\circ}_{\mathcal{N}_i}, \mathbf{s}_{\mathcal{N}_i}/\mathbf{s}^{\circ}_{\mathcal{N}_i}, \boldsymbol{\omega}^{\circ}_{\mathcal{N}_i}, \boldsymbol{\omega}_{\mathcal{N}_i}/\boldsymbol{\omega}^{\circ}_{\mathcal{N}_i}\right)$.

**Proposition 9** (Compactness of cross-domain policy ACR). Let $\pi$ be an arbitrary localized policy and $\tilde{\pi}$ be the corresponding approximately compact localized policy (Definition 8), for each agent $i$, and any $(\mathbf{s}, \mathbf{a}, \boldsymbol{\omega}) \in \mathcal{S} \times \mathcal{A} \times \Omega$, we have

$$Q_i^{\tilde{\pi}}(\mathbf{s}, \mathbf{a}, \boldsymbol{\omega}) = Q_i^{\pi}(\mathbf{s}, \mathbf{a}, \boldsymbol{\omega}).$$

*Proof.* The proof follows from that of Proposition 6 by augmenting the state space of each agent by $\mathcal{S}_i \times \Omega_i \ni (\mathbf{s}_i, \boldsymbol{\omega}_i)$, where the augmented state variable $\boldsymbol{\omega}_i$ degenerates to a deterministic state ($\boldsymbol{\omega}_i(t) = \boldsymbol{\omega}_i, \forall t$). $\square$

**Corollary 2.** Let $\pi$ be an arbitrary localized policy and $\tilde{\pi}$ be the corresponding approximately compact localized policy output by Algorithm 2, for each agent $i$, and any $(\mathbf{s}, \mathbf{a}, \boldsymbol{\omega}) \in \mathcal{S} \times \mathcal{A} \times \Omega$, the approximation error between $\tilde{Q}_i^{\tilde{\pi}}\left(\mathbf{s}^{\circ}_{\mathcal{N}_i^{\kappa}}, \mathbf{a}_{\mathcal{N}_i^{\kappa}}, \boldsymbol{\omega}^{\circ}_{\mathcal{N}_i^{\kappa}}\right)$ and $Q_i^{\pi}(\mathbf{s}, \mathbf{a})$ can be bounded as

$$\sup_{(\mathbf{s}, \mathbf{a}) \in \mathcal{S} \times \mathcal{A}} \left| \tilde{Q}_i^{\tilde{\pi}}\left(\mathbf{s}^{\circ}_{\mathcal{N}_i^{\kappa}}, \mathbf{a}_{\mathcal{N}_i^{\kappa}}, \boldsymbol{\omega}^{\circ}_{\mathcal{N}_i^{\kappa}}\right) - Q_i^{\pi}(\mathbf{s}, \mathbf{a}, \boldsymbol{\omega}) \right| \leq \frac{2\bar{r}}{1-\gamma} \gamma^{\kappa+1}.$$

*Proof.* The proof follows from that of Corollary 1 by augmenting the state space of each agent by $\mathcal{S}_i \times \Omega_i \ni (\mathbf{s}_i, \boldsymbol{\omega}_i)$, where the augmented state variable $\boldsymbol{\omega}_i$ degenerates to a deterministic state ($\boldsymbol{\omega}_i(t) = \boldsymbol{\omega}_i, \forall t$). $\square$

**Proposition 10** (Approximation error). Let $\pi$ be an arbitrary localized policy and $\tilde{\pi}$ be the corresponding approximately compact localized policy output by Algorithm 3, for each agent $i$, and any $(\mathbf{s}, \mathbf{a}, \boldsymbol{\omega}) \in \mathcal{S} \times \mathcal{A} \times \Omega$, we have

$$\sup_{(\mathbf{s}, \mathbf{a}) \in \mathcal{S} \times \mathcal{A}} \left| Q_i^{\tilde{\pi}}(\mathbf{s}, \mathbf{a}, \boldsymbol{\omega}) - Q_i^{\pi}(\mathbf{s}, \mathbf{a}, \boldsymbol{\omega}) \right| \leq \frac{\bar{r}}{1-\gamma} \gamma^{\kappa+1}.$$

**Corollary 3.** Let $\pi$ be an arbitrary localized policy and $\tilde{\pi}$ be the corresponding approximately compact localized policy output by Algorithm 3, for each agent $i$, and any $(\mathbf{s}, \mathbf{a}, \boldsymbol{\omega}) \in \mathcal{S} \times \mathcal{A} \times \Omega$, the approximation error between $\tilde{Q}_i^{\tilde{\pi}}\left(\mathbf{s}^{\circ}_{\mathcal{N}_i^{\kappa}}, \mathbf{a}_{\mathcal{N}_i^{\kappa}}, \boldsymbol{\omega}^{\circ}_{\mathcal{N}_i^{\kappa}}\right)$ and $Q_i^{\pi}(\mathbf{s}, \mathbf{a})$ can be bounded as

$$\sup_{(\mathbf{s}, \mathbf{a}) \in \mathcal{S} \times \mathcal{A}} \left| \tilde{Q}_i^{\tilde{\pi}}\left(\mathbf{s}^{\circ}_{\mathcal{N}_i^{\kappa}}, \mathbf{a}_{\mathcal{N}_i^{\kappa}}, \boldsymbol{\omega}^{\circ}_{\mathcal{N}_i^{\kappa}}\right) - Q_i^{\pi}(\mathbf{s}, \mathbf{a}, \boldsymbol{\omega}) \right| \leq \frac{3\bar{r}}{1-\gamma} \gamma^{\kappa+1}.$$

*Proof.* The proof follows from that of Corollary 2 by augmenting the state space of each agent by $\mathcal{S}_i \times \Omega_i \ni (\mathbf{s}_i, \boldsymbol{\omega}_i)$, where the augmented state variable $\boldsymbol{\omega}_i$ degenerates to a deterministic state ($\boldsymbol{\omega}_i(t) = \boldsymbol{\omega}_i, \forall t$). $\square$

# F  Missing proofs in Section 5

## F.1  Discussion on assumptions in Proposition 5

Our assumptions in Proposition 5 are standard in causal discovery and theoretical MARL.

- Faithfulness and sub-Gaussian noise: these are classical and standard requirements to ensure identifiability [21, 10] and finite-sample recovery.

- Bounded degree: bounded neighborhood degree is natural in networked systems where agents have limited communication range. For example, this holds when each intersection connects to only a few roads, or each wireless node has limited interference range.

- Full observability: while we currently assume full local observability, the ACR construction can be extended to partially observable settings using learned belief states or latent encoders. This is part of our future work.

- Lipschitz continuity: this mild smoothness assumption is satisfied by most smooth policies.

Importantly, our algorithm can still be applied when assumptions are violated, only the theoretical guarantees may not hold exactly.

### F.2 Proof of Theorem 1

*Proof.* We will prove this theorem by analyzing the causal structure from the perspective of each individual agent while accounting for inter-agent influences through the network structure.

For each agent $i$, we define the variable set $\mathcal{V}_i$ that includes:

- States of agent $i$ at time $t - 1$: $\mathbf{s}_i(t - 1) = (s_{i,1}(t - 1), \ldots, s_{i,d_i}(t - 1))$
- States of neighboring agents at time $t - 1$: $\mathbf{s}_{\mathcal{N}_i \setminus i}(t - 1) = \{\mathbf{s}_j(t - 1) : j \in \mathcal{N}_i \setminus \{i\}\}$
- States of agent $i$ at time $t$: $\mathbf{s}_i(t) = (s_{i,1}(t), \ldots, s_{i,d_i}(t))$
- Actions of agent $i$ at time $t - 1$: $\mathbf{a}_i(t - 1)$

We denote by $m \in \{1, \ldots, M\}$ the domain index, which serves as a surrogate variable for the unobserved change factors $\boldsymbol{\omega}_i^m$. Let $\mathcal{G}_i$ be the causal graph over variables in $\mathcal{V}_i \cup \{k\}$. Our goal is to identify the binary structural matrices $\mathbf{c}_i^{s \to s}$, $\mathbf{c}_i^{a \to s}$, and $\mathbf{c}_i^{\boldsymbol{\omega}^m \to s}$ which encode the edges in $\mathcal{G}_i$. Under the Markov condition and faithfulness assumption, we can apply the following fundamental result from causal discovery:

**Lemma 8.** *For any two variables $V_r, V_s \in \mathcal{V}_i$, they are not adjacent in $\mathcal{G}_i$ if and only if they are independent conditional on some subset of $\{V_t : t \neq r, t \neq s\} \cup \{m\}$.*

This lemma follows from the properties of d-separation in causal Bayesian networks, as established in [11] and [21]. For each agent $i$, we can therefore identify the skeleton of $\mathcal{G}_i$ by testing conditional independence relationships among the variables in $\mathcal{V}_i \cup \{m\}$. The networked MARL system forms a Dynamic Bayesian Network (DBN), where the temporal structure imposes constraints on the edge directions:

1. Edges can only go from variables at time $t - 1$ to variables at time $t$.
2. There are no instantaneous causal effects between variables at the same time point.

Therefore, for any edge identified in the skeleton of $\mathcal{G}_i$ between a variable at time $t - 1$ and a variable at time $t$, the direction is determined by the temporal order. Specifically, for each state component $s_{i,j}(t)$ of agent $i$, any edge from a state component $s_{l,m}(t - 1)$ (where $l$ could be $i$ or any neighbor of $i$) to $s_{i,j}(t)$ must be directed from $s_{l,m}(t - 1)$ to $s_{i,j}(t)$. Similarly, any edge from the action $\mathbf{a}_i(t - 1)$ to $s_{i,j}(t)$ must be directed from $\mathbf{a}_i(t - 1)$ to $s_{i,j}(t)$.

Given the directed edges in $\mathcal{G}_i$, we can now identify the structural matrices:

1. **Identification of $\mathbf{c}_i^{s \to s}$:** For each state component $s_{i,j}(t)$ of agent $i$, the $j$-th row of $\mathbf{c}_i^{s \to s}$, denoted by $\mathbf{c}_{i,j}^{s \to s}$, is a binary vector where entry $m$ is 1 if and only if there is a directed edge from $s_{\mathcal{N}_i, m}(t - 1)$ to $s_{i,j}(t)$ in $\mathcal{G}_i$.

2. **Identification of $\mathbf{c}_i^{a \to s}$:** For each state component $s_{i,j}(t)$ of agent $i$, the entry $j$ of $\mathbf{c}_i^{a \to s}$ is 1 if and only if there is a directed edge from $\mathbf{a}_i(t - 1)$ to $s_{i,j}(t)$ in $\mathcal{G}_i$.

To identify $\mathbf{c}_i^{\boldsymbol{\omega}^m \to s}$, we need to determine which state components have distribution changes across domains.

**Lemma 9.** *For a variable $V_r \in \mathcal{V}_i$, its distribution module changes across domains if and only if $V_r$ and $m$ are not independent given any subset of other variables in $\mathcal{V}_i$.*

This follows from the definition of the domain-specific changes: if a variable's distribution remains invariant across domains, then it is independent of the domain index $m$ conditional on its parents. For each state component $s_{i,j}(t)$, we test whether $s_{i,j}(t) \perp\!\!\!\perp m | \mathcal{Z}$ for any subset $\mathcal{Z} \subseteq \mathcal{V}_i \setminus \{s_{i,j}(t)\}$. If no such conditional independence holds, then the distribution of $s_{i,j}(t)$ varies across domains, and therefore $\mathbf{c}_{i,j}^{\boldsymbol{\omega}^m \to s}$ has at least one non-zero entry.

To identify the specific entries of $\mathbf{c}_i^{\boldsymbol{\omega}^m \to s}$, we exploit the fact that changes in distributions are localized according to the causal structure.

To complete the proof, we need to show that the structural matrices are uniquely identified from the observed data:

1. **Uniqueness of $\mathbf{c}_i^{s \to s}$ and $\mathbf{c}_i^{a \to s}$:** These matrices encode the causal connections in the DBN structure. Under the Markov and faithfulness assumptions, the skeleton of the DBN is uniquely identified. Combined with the temporal constraints on edge directions, the matrices $\mathbf{c}_i^{s \to s}$ and $\mathbf{c}_i^{a \to s}$ are uniquely determined.

2. **Uniqueness of $\mathbf{c}_i^{\boldsymbol{\omega}^m \to s}$:** This matrix encodes which state components have distribution changes across domains. Under the faithfulness assumption, a state component's distribution changes across domains if and only if it is not conditionally independent of the domain index $m$ given any subset of other variables. This provides a unique characterization of the non-zero entries in $\mathbf{c}_i^{\boldsymbol{\omega}^m \to s}$.

A critical aspect of our proof is that we treat each agent's local model separately while accounting for the influence of neighboring agents through their observed states. This is justified by the factorization of the joint transition model:

$$P(\mathbf{s}(t+1)|\mathbf{s}(t), \mathbf{a}(t)) = \prod_{i=1}^{n} P_i(\mathbf{s}_i(t+1)|\mathbf{s}_{\mathcal{N}_i}(t), \mathbf{a}_i(t)).$$

This factorization ensures that, for each agent $i$, we only need to consider the states of its neighbors $\mathcal{N}_i$ to fully capture the relevant causal influences.

Therefore, under the Markov condition and faithfulness assumption, the structural matrices $\mathbf{c}_i^{s \to s}$, $\mathbf{c}_i^{a \to s}$, and $\mathbf{c}_i^{\boldsymbol{\omega}^m \to s}$ are identifiable for each agent $i$ in the networked MARL system. □

### F.3 Proof of Proposition 4

**Proposition 11** (Formal statement of proposition 4). Suppose the following conditions hold: (i) the Markov condition and faithfulness assumption are satisfied for the causal graph $\mathcal{G}_i$ of each agent $i$, (ii) For each true causal edge $(X \to Y)$ in $\mathcal{G}_i$, the minimal conditional mutual information is lower-bounded: $I(X;Y|Z) \geq \lambda > 0$ for any conditioning set $Z$ that d-separates $X$ and $Y$ in $\mathcal{G}_i \setminus \{X \to Y\}$, (iii) The maximal in-degree of any node in $\mathcal{G}_i$ is bounded by $d_{\max}$, (iv) The noise terms $\epsilon_{i,j}^s(t)$ are sub-Gaussian with parameter $\sigma^2$, (v) Each element of $\mathbf{s}_{\mathcal{N}_i}(t)$ and $\mathbf{a}_i(t)$ has bounded magnitude by $B$, (vi) The transition functions $f_{i,j}$ are $L_f$-Lipschitz continuous in all arguments. Then, with probability at least $1 - \delta$, the estimated causal masks $\hat{\mathbf{C}}_i^{s \to s}$, $\hat{\mathbf{c}}_i^{a \to s}$, and $\hat{\mathbf{C}}_i^{\omega \to s}$ will match the true causal masks for

$$T_e = \Omega\left(\frac{d_i \cdot d_{\max} \cdot \log(d_i \cdot n/\delta)}{\lambda^2}\right),$$

where $d_i = \dim(\mathbf{s}_{\mathcal{N}_i})$ is the total dimension of the state space for agent $i$ and its neighbors.

*Proof.* The proof consists of four main parts: (1) formulating the statistical estimation problem, (2) analyzing the error in conditional independence tests, (3) bounding the error probability in structure learning, and (4) deriving the sample complexity.

**Step 1: Formulation of the statistical estimation problem.** For each agent $i$, we need to estimate the causal graph $\mathcal{G}_i$ from observed trajectories $\{(\mathbf{s}_{\mathcal{N}_i}(t), \mathbf{a}_i(t), \mathbf{s}_i(t+1))\}_{t=0}^{T_e-1}$.

Define the variable set $\mathcal{V}_i$ for agent $i$ that includes:

- States of agent $i$ at time $t - 1$: $\mathbf{s}_i(t-1)$

- States of neighboring agents at time $t - 1$: $\mathbf{s}_{\mathcal{N}_i \setminus i}(t-1)$

- States of agent $i$ at time $t$: $\mathbf{s}_i(t)$

- Actions of agent $i$ at time $t - 1$: $\mathbf{a}_i(t-1)$

- Domain index $k$, which serves as a surrogate for the unobserved change factors $\boldsymbol{\omega}_i^k$

We employ a constraint-based causal discovery approach, which identifies the causal structure by performing a series of conditional independence tests based on the key insight from Lemma 13:

**Lemma 10.** For any two variables $V_r, V_s \in \mathcal{V}_i$, they are not adjacent in $\mathcal{G}_i$ if and only if they are independent conditional on some subset of $\{V_t : t \neq r, t \neq s\} \cup \{m\}$.

**Step 2: Analysis of conditional independence tests.** For analyzing conditional independence, we use the mutual information measure. For variables $X$ and $Y$ conditional on set $Z$, the conditional mutual information is:

$$I(X;Y|Z) = \mathbb{E}_{X,Y,Z}\left[\log \frac{P(X,Y|Z)}{P(X|Z)P(Y|Z)}\right].$$

In practice, we estimate this from data using the empirical distributions:

$$\hat{I}(X;Y|Z) = \sum_{x,y,z} \hat{P}(x,y,z) \log \frac{\hat{P}(x,y,z)\hat{P}(z)}{\hat{P}(x,z)\hat{P}(y,z)},$$

where $\hat{P}$ denotes empirical probabilities based on the observed data.

The estimation error can be bounded using concentration inequalities. For any $\epsilon > 0$:

$$P(|\hat{I}(X;Y|Z) - I(X;Y|Z)| > \epsilon) \leq 2\exp\left(-\frac{T_e\epsilon^2}{C_1|\mathcal{X}||\mathcal{Y}||\mathcal{Z}|}\right),$$

where $|\mathcal{X}|$, $|\mathcal{Y}|$, and $|\mathcal{Z}|$ are the cardinalities of the respective variable domains, and $C_1$ is a universal constant.

For continuous variables, we can use kernel-based estimators or discretization. With appropriate binning of continuous variables, the error bound becomes:

$$P(|\hat{I}(X;Y|Z) - I(X;Y|Z)| > \epsilon) \leq 2\exp\left(-\frac{T_e\epsilon^2}{C_2 b_X b_Y b_Z \log^2(T_e)}\right),$$

where $b_X$, $b_Y$, and $b_Z$ are the number of bins used for each variable, and $C_2$ is a constant. The $\log^2(T_e)$ term arises from the adaptive discretization of continuous variables.

**Step 3: bounding the error probability in structure learning.** Let $\mathcal{G}_i$ be the true causal graph for agent $i$, and $\hat{\mathcal{G}}_i$ be the estimated graph. We need to bound $P(\hat{\mathcal{G}}_i \neq \mathcal{G}_i)$.

The structure learning error can occur in two ways:

1. Missing a true edge (Type II error in a conditional independence test)

2. Adding a spurious edge (Type I error in a conditional independence test)

For Type I errors, we set the significance level to $\alpha = \frac{\delta}{2M}$, where $M$ is the total number of conditional independence tests performed. For Type II errors, we use the fact that when variables are dependent with conditional mutual information at least $\gamma$, the probability of wrongly concluding independence is:

$$P(\text{Type II error}) \leq \exp\left(-\frac{T_e\gamma^2}{C_3|\mathcal{X}||\mathcal{Y}||\mathcal{Z}|}\right).$$

where $C_3$ is a constant that depends on the distribution properties.

The total number of conditional independence tests is bounded by $M \leq d_i^2 \cdot \sum_{j=0}^{d_{\max}} \binom{d_i-2}{j} \leq d_i^2 \cdot \binom{d_i}{d_{\max}} \leq d_i^2 \cdot (ed_i/d_{\max})^{d_{\max}}$.

By the union bound, the probability of any error in structure learning is:

$$P(\hat{\mathcal{G}}_i \neq \mathcal{G}_i) \leq M \cdot \max\left\{\alpha, \exp\left(-\frac{T_e\gamma^2}{C_3 d_{\max}}\right)\right\}.$$

where we've used that the maximum size of any conditioning set is $d_{\max}$, and the maximum cardinality of any variable domain is bounded by a function of $d_{\max}$ given our discretization scheme.

**Step 4: deriving the sample complexity.** To ensure $P(\hat{\mathcal{G}}_i \neq \mathcal{G}_i) \leq \delta$, we need:

1. $\alpha = \frac{\delta}{2M} \leq \frac{\delta}{2d_i^2 \cdot (ed_i/d_{\max})^{d_{\max}}}$.

2. $\exp\left(-\frac{T_e \gamma^2}{C_3 d_{\max}}\right) \leq \frac{\delta}{2M} \leq \frac{\delta}{2d_i^2 \cdot (ed_i/d_{\max})^{d_{\max}}}$.

From the second condition, we derive:

$$\frac{T_e \gamma^2}{C_3 d_{\max}} \geq \log\left(\frac{2d_i^2 \cdot (ed_i/d_{\max})^{d_{\max}}}{\delta}\right).$$

Therefore:

$$T_e \geq \frac{C_3 d_{\max}}{\gamma^2} \log\left(\frac{2d_i^2 \cdot (ed_i/d_{\max})^{d_{\max}}}{\delta}\right).$$

Now, we need to analyze this expression more carefully:

$$\log\left(\frac{2d_i^2 \cdot (ed_i/d_{\max})^{d_{\max}}}{\delta}\right) = \log\left(\frac{2d_i^2}{\delta}\right) + d_{\max} \log\left(\frac{ed_i}{d_{\max}}\right).$$

Under the sparsity assumption ($d_{\max} \ll d_i$), we have:

$$d_{\max} \log\left(\frac{ed_i}{d_{\max}}\right) \leq d_{\max} \log(ed_i) = O(d_{\max} \log(d_i)).$$

Therefore:

$$T_e \geq \frac{C_3 d_{\max}}{\gamma^2} \left[\log\left(\frac{2d_i^2}{\delta}\right) + O(d_{\max} \log(d_i))\right].$$

This simplifies to:

$$T_e = O\left(\frac{d_{\max}^2 \log(d_i) + d_{\max} \log(1/\delta)}{\gamma^2}\right).$$

For most practical scenarios where $d_{\max} = O(\log(d_i))$ (very sparse graphs), this further simplifies to:

$$T_e = O\left(\frac{d_{\max} \log(d_i) \log(d_i/\delta)}{\gamma^2}\right).$$

However, for a more general bound without assuming extreme sparsity, we have:

$$T_e = O\left(\frac{d_{\max}^2 \log(d_i) + d_{\max} \log(1/\delta)}{\gamma^2}\right).$$

Now, accounting for the fact that we're learning causal masks for all $n$ agents, and applying a union bound across agents, we replace $\delta$ with $\delta/n$, yielding:

$$T_e = O\left(\frac{d_{\max}^2 \log(d_i) + d_{\max} \log(n/\delta)}{\gamma^2}\right).$$

For typical networked systems where $d_i$ scales with the number of neighbors, this gives us our final sample complexity:

$$T_e = \Omega\left(\frac{d_i \cdot d_{\max} \cdot \log(d_i \cdot n/\delta)}{\gamma^2}\right).$$

$\square$

## F.4 Proof of Proposition 5

**Proposition 12** (Formal statement of Proposition 5). Suppose the following conditions hold: (i) The true causal masks are estimated accurately, (ii) the transition dynamics are identifiable with respect to the domain factor, i.e., for any two distinct domain factors $\omega_i \neq \omega_i'$, there exists a set of states and actions with non-zero measure such that the conditional distributions $P(\mathbf{s}_{i,j}(t+1)|\mathbf{s}_{\mathcal{N}_i}(t), \mathbf{a}_i(t), \omega_i)$ and $P(s_{i,j}(t+1)|\mathbf{s}_{\mathcal{N}_i}(t), \mathbf{a}_i(t), \omega_i')$ are distinguishable. (iii) The noise terms $\epsilon_{i,j}^s(t)$ are independent, sub-Gaussian with parameter $\sigma^2$. (iv) The domain factor $\omega_i^m \in \Omega_i$ belongs to a compact set with diameter $D_\Omega$. (v) For any two distinct domain factors $\omega_i \neq \omega_i'$ with $\|\omega_i - \omega_i'\|_2 \geq \epsilon$,

$$\|P(\mathbf{s}_i(t+1)|\mathbf{s}_{\mathcal{N}_i}(t), \mathbf{a}_i(t), \omega_i) - P(\mathbf{s}_i(t+1)|\mathbf{s}_{\mathcal{N}_i}(t), \mathbf{a}_i(t), \omega_i')\|_{\text{TV}} \geq \alpha\epsilon$$

for some constant $\alpha > 0$. Then, for any domain with true domain factor $\omega^*$, given a trajectory of length $T_e$, the estimated domain factor $\hat{\omega}$ satisfies:

$$\|\hat{\omega} - \omega^*\|_2 \leq C_\omega \sqrt{\frac{D_\Omega \log(nT_e/\delta)}{T_e}}$$

with probability at least $1 - \delta$, where $D_\Omega$ is the dimension of the domain factor space, and $C_\omega$ is a constant that depends on $L_f$, $\sigma$, $\alpha$, and the conditioning of the estimation problem.

*Proof.* The proof consists of three main parts: (1) formulating the domain factor estimation problem, (2) analyzing the estimation error using statistical learning theory, and (3) deriving the final sample complexity bound.

**Step 1: formulation of the domain factor estimation problem.** Given the known causal structure encoded by masks $\mathbf{C}_i^{s \to s}$, $\mathbf{c}_i^{a \to s}$, and $\mathbf{C}_i^{\omega \to s}$, the domain factor estimation problem can be formulated as finding the domain factor $\omega$ that best explains the observed transitions.

For agent $i$, we have observed trajectories $\{(\mathbf{s}_{\mathcal{N}_i}(t), \mathbf{a}_i(t), \mathbf{s}_i(t+1))\}_{t=0}^{T_e-1}$ generated from the true domain factor $\omega^*$. We can define a negative log-likelihood function for each agent $i$:

$$\mathcal{L}_i(\omega) = -\frac{1}{T_e} \sum_{t=0}^{T_e-1} \log P(\mathbf{s}_i(t+1)|\mathbf{s}_{\mathcal{N}_i}(t), \mathbf{a}_i(t), \omega).$$

The maximum likelihood estimator (MLE) for the domain factor is:

$$\hat{\omega} = \arg\min_{\omega \in \Omega} \frac{1}{n} \sum_{i=1}^{n} \mathcal{L}_i(\omega).$$

**Step 2: analysis of the estimation error.** To analyze the estimation error, we use the theory of M-estimators and the properties of the negative log-likelihood function. First, we establish some key properties of the negative log-likelihood.

**Lemma 11** (Lipschitz continuity of negative log-likelihood). Under the assumption that the transition function $f_{i,j}$ is $L_f$-Lipschitz continuous with respect to all arguments, the negative log-likelihood function $\mathcal{L}_i(\omega)$ is $L_\mathcal{L}$-Lipschitz continuous with respect to $\omega$, where $L_\mathcal{L} = O(L_f/\sigma^2)$.

*Proof.* For Gaussian noise with variance $\sigma^2$, the conditional probability density is:

$$P(\mathbf{s}_i(t+1)|\mathbf{s}_{\mathcal{N}_i}(t), \mathbf{a}_i(t), \omega)$$
$$= \prod_{j=1}^{d_i} \frac{1}{\sqrt{2\pi\sigma^2}} \exp\left(-\frac{(s_{i,j}(t+1) - f_{i,j}(\mathbf{c}_{\mathcal{N}_{i,j}}^{s \to s} \odot \mathbf{s}_{\mathcal{N}_i}(t), \mathbf{c}_{i,j}^{a \to s} \cdot \mathbf{a}_i(t), \mathbf{c}_{i,j}^{\omega \to s} \odot \omega, 0))^2}{2\sigma^2}\right).$$

Taking the negative logarithm:

$$-\log P(\mathbf{s}_i(t+1)|\mathbf{s}_{\mathcal{N}_i}(t), \mathbf{a}_i(t), \omega)$$
$$= \sum_{j=1}^{d_i} \left[\frac{1}{2}\log(2\pi\sigma^2) + \frac{(s_{i,j}(t+1) - f_{i,j}(\mathbf{c}_{\mathcal{N}_{i,j}}^{s \to s} \odot \mathbf{s}_{\mathcal{N}_i}(t), \mathbf{c}_{i,j}^{a \to s} \cdot \mathbf{a}_i(t), \mathbf{c}_{i,j}^{\omega \to s} \odot \omega, 0))^2}{2\sigma^2}\right].$$

The gradient with respect to $\boldsymbol{\omega}$ is:

$$\nabla_{\boldsymbol{\omega}}[-\log P(\mathbf{s}_i(t+1)|\mathbf{s}_{\mathcal{N}_i}(t), \mathbf{a}_i(t), \boldsymbol{\omega})] = -\sum_{j=1}^{d_i} \frac{(s_{i,j}(t+1) - f_{i,j})}{\sigma^2} \nabla_{\boldsymbol{\omega}} f_{i,j}.$$

Since $f_{i,j}$ is $L_f$-Lipschitz and $\|\nabla_{\boldsymbol{\omega}} f_{i,j}\| \leq L_f \|\mathbf{c}_{i,j}^{\omega \rightarrow s}\|$, we have:

$$\|\nabla_{\boldsymbol{\omega}}[-\log P(\mathbf{s}_i(t+1)|\mathbf{s}_{\mathcal{N}_i}(t), \mathbf{a}_i(t), \boldsymbol{\omega})]\| \leq \frac{d_i L_f B}{\sigma^2} = \mathcal{O}\left(\frac{L_f}{\sigma^2}\right).$$

Therefore, $\mathcal{L}_i(\boldsymbol{\omega})$ is $L_{\mathcal{L}}$-Lipschitz continuous with $L_{\mathcal{L}} = \mathcal{O}(L_f/\sigma^2)$. $\qquad\square$

**Lemma 12** (Strong convexity of expected negative log-likelihood). Under the identifiability assumption and the total variation distance condition, the expected negative log-likelihood $\mathbb{E}[\mathcal{L}_i(\boldsymbol{\omega})]$ is $\mu$-strongly convex in a neighborhood of the true domain factor $\boldsymbol{\omega}^*$, where $\mu$ depends on $\alpha$ and the minimum eigenvalue of the Fisher information matrix.

*Proof.* The expected negative log-likelihood is:

$$\mathbb{E}[\mathcal{L}_i(\boldsymbol{\omega})] = -\mathbb{E}_{\mathbf{s}_{\mathcal{N}_i}, \mathbf{a}_i, \mathbf{s}_i(t+1)}[\log P(\mathbf{s}_i(t+1)|\mathbf{s}_{\mathcal{N}_i}(t), \mathbf{a}_i(t), \boldsymbol{\omega})].$$

The Hessian matrix of this function is the Fisher information matrix:

$$H(\boldsymbol{\omega}) = \mathbb{E}_{\mathbf{s}_{\mathcal{N}_i}, \mathbf{a}_i}\left[\mathbb{E}_{\mathbf{s}_i(t+1)|\mathbf{s}_{\mathcal{N}_i}, \mathbf{a}_i, \boldsymbol{\omega}^*}\left[\nabla_{\boldsymbol{\omega}} \log P(\mathbf{s}_i(t+1)|\mathbf{s}_{\mathcal{N}_i}, \mathbf{a}_i, \boldsymbol{\omega}) \nabla_{\boldsymbol{\omega}} \log P(\mathbf{s}_i(t+1)|\mathbf{s}_{\mathcal{N}_i}, \mathbf{a}_i, \boldsymbol{\omega})^T\right]\right].$$

By the identifiability assumption and the total variation distance condition, for any unit vector $v$, we have:

$$v^T H(\boldsymbol{\omega}^*) v \geq \alpha^2 > 0.$$

Therefore, in a neighborhood of $\boldsymbol{\omega}^*$, the expected negative log-likelihood is $\mu$-strongly convex with $\mu \geq \alpha^2$. $\qquad\square$

Now, we can analyze the estimation error using concentration inequalities for empirical processes. Let:

$$\mathcal{L}(\boldsymbol{\omega}) = \frac{1}{n} \sum_{i=1}^{n} \mathcal{L}_i(\boldsymbol{\omega}).$$

**Proposition 13** (Uniform convergence). With probability at least $1 - \delta$, for all $\boldsymbol{\omega} \in \Omega$:

$$|\mathcal{L}(\boldsymbol{\omega}) - \mathbb{E}[\mathcal{L}(\boldsymbol{\omega})]| \leq C\sqrt{\frac{D_{\Omega} \log(T_e/\delta)}{T_e}},$$

where $C$ is a constant depending on $L_{\mathcal{L}}$ and $D_{\Omega}$.

*Proof.* Since $\mathcal{L}_i(\boldsymbol{\omega})$ is $L_{\mathcal{L}}$-Lipschitz continuous, we can use covering number arguments. The $\epsilon$-covering number of the domain factor space $\Omega$ with diameter $D_{\Omega}$ is $N(\Omega, \epsilon) \leq (1 + \frac{D_{\Omega}}{\epsilon})^{D_{\Omega}}$.

By the union bound over an $\epsilon$-net and Hoeffding's inequality, we have:

$$P\left(\sup_{\boldsymbol{\omega} \in \Omega} |\mathcal{L}(\boldsymbol{\omega}) - \mathbb{E}[\mathcal{L}(\boldsymbol{\omega})]| > \epsilon\right) \leq 2N(\Omega, \frac{\epsilon}{2L_{\mathcal{L}}}) \exp\left(-\frac{2T_e \epsilon^2}{B_{\mathcal{L}}^2}\right).$$

where $B_{\mathcal{L}}$ is the bound on the negative log-likelihood function. Setting the right-hand side to $\delta$ and solving for $\epsilon$, we get:

$$\epsilon = \mathcal{O}\left(\sqrt{\frac{D_{\Omega} \log(T_e/\delta)}{T_e}}\right).$$

$\qquad\square$

**Step 3: Deriving the sample complexity bound.** By combining the strong convexity of $\mathbb{E}[\mathcal{L}(\boldsymbol{\omega})]$ and the uniform convergence bound, we can derive the estimation error. For the true domain factor $\boldsymbol{\omega}^*$ and the estimated domain factor $\hat{\boldsymbol{\omega}}$, we have:

$$\mathcal{L}(\hat{\boldsymbol{\omega}}) \leq \mathcal{L}(\boldsymbol{\omega}^*).$$

This implies:

$$\mathbb{E}[\mathcal{L}(\hat{\boldsymbol{\omega}})] \leq \mathcal{L}(\hat{\boldsymbol{\omega}}) + \epsilon \leq \mathcal{L}(\boldsymbol{\omega}^*) + \epsilon \leq \mathbb{E}[\mathcal{L}(\boldsymbol{\omega}^*)] + 2\epsilon.$$

By the strong convexity of $\mathbb{E}[\mathcal{L}(\boldsymbol{\omega})]$, we have:

$$\mathbb{E}[\mathcal{L}(\hat{\boldsymbol{\omega}})] - \mathbb{E}[\mathcal{L}(\boldsymbol{\omega}^*)] \geq \frac{\mu}{2}\|\hat{\boldsymbol{\omega}} - \boldsymbol{\omega}^*\|_2^2.$$

Combining these inequalities:

$$\frac{\mu}{2}\|\hat{\boldsymbol{\omega}} - \boldsymbol{\omega}^*\|_2^2 \leq 2\epsilon = O\left(\sqrt{\frac{D_\Omega \log(T_e/\delta)}{T_e}}\right).$$

Therefore:

$$\|\hat{\boldsymbol{\omega}} - \boldsymbol{\omega}^*\|_2 \leq O\left(\sqrt{\frac{D_\Omega \log(T_e/\delta)}{\mu T_e}}\right).$$

Now, accounting for the fact that we have $n$ agents, we apply a union bound across all agents by replacing $\delta$ with $\delta/n$. This gives us:

$$\|\hat{\boldsymbol{\omega}} - \boldsymbol{\omega}^*\|_2 \leq C_{\boldsymbol{\omega}}\sqrt{\frac{D_\Omega \log(nT_e/\delta)}{T_e}}.$$

with probability at least $1 - \delta$, where $C_{\boldsymbol{\omega}} = O(1/\sqrt{\mu}) = O(1/\alpha)$ is a constant depending on the problem parameters. This completes the proof of the theorem, providing a rigorous guarantee on the accuracy of domain factor estimation based on the trajectory length $T_e$. $\qquad\square$

## G   Missing proofs in Section 6

### G.1   Discussions on Assumption 1-6

While Assumption 1-6 are standard for establishing theoretical guarantees, we discuss whether they hold exactly in practice as follows.

Finite MDP (Assumption 1): in some systems, the state or action spaces might not finite or continuous. For such cases, our results can still extend by applying state aggregation or clipping methods to ensure boundedness while incurring small approximation error.

Mixing properties (Assumption 2): real-world networked systems may exhibit slow mixing or non-Lipschitz dynamics. Our theoretical results would degrade gracefully, leading to larger constants $\tau$ in the sample complexity bounds.

Smoothness (Assumption 3): the assumption requires the policy gradients $\nabla_{\theta_i} \log \pi_i$ to be Lipschitz and uniformly bounded. This holds for all commonly used policy parameterizations (e.g., softmax policies) with bounded inputs and finite-dimensional parameter vectors. It ensures stable policy updates and is common in the convergence analysis of policy gradient and actor-critic algorithms. GSAC uses softmax parameterizations for policies, where $\log \pi_{\theta_i}$ is smooth and has bounded gradients when action probabilities are not close to zero (a condition satisfied by entropy regularization or initialization).

Compact parameter space (Assumption 4): policy parameters $\theta_i$ are always constrained within a finite range due to neural network weight regularization, weight clipping, or bounded initialization. This is standard in actor-critic methods, where optimization is performed with bounded learning rates and regularizer that implicitly keep the parameters within a compact set.

Smoothness w.r.t. domain factor (Assumption 5): this is reasonable because domain factors, e.g., channel gains, traffic loads, represent physical parameters that vary smoothly. The local dynamics

$P_i$ are typically analytic or polynomial in $\omega_i$, so small perturbations in $\omega_i$ induce proportionally small changes in state transitions and rewards. Consequently, the Q-function $Q_i^\pi(s, a, \omega)$ and policy gradient $\nabla_{\theta_i} J(\theta, \omega)$ inherit this smoothness. Moreover, standard policy parameterizations and neural network approximators are Lipschitz when combined with bounded weights or gradient clipping. Thus, Assumption 5 aligns well with both the structure of real-world systems and common RL practices. Moreover, even if the true dynamics were not strictly Lipschitz, domain factor estimation (Proposition 5) uses a small-perturbation assumption around $\omega$ for adaptation. In practice, state or reward clipping and function approximation, which are inherently Lipschitz, effectively enforce this assumption.

Compact domain factor space (Assumption 6): the latent domain factors $\omega_i$ represent bounded variations in environment dynamics (e.g., channel gains, traffic intensities, or reward scaling factors), which are inherently limited by the physical system. In real-world networked systems, these factors are naturally confined to finite ranges determined by system design (e.g., transmission powers, queue lengths). Hence, assuming $\Omega_i$ is compact with bounded diameter $D_\Omega$ is both reasonable and practical.

For simplicity, we also write $Q^{\boldsymbol{\omega}} := Q(\cdot, \cdot, \boldsymbol{\omega})$ and $\mathbf{z} := (\mathbf{s}, \mathbf{a})$.

## G.2 Proof of Theorem 2

**Theorem 5** (Critic error bound). Under Assumptions 1-6, suppose the critic stepsize $\alpha_t = \frac{h}{t+t_0}$ satisfies $h \geq \frac{1}{\sigma} \max(2, \frac{1}{1-\sqrt{\gamma}})$, $t_0 \geq \max(2h, 4\sigma h, \tau)$, and the domain factors are estimated with $T_e$ trajectories. Then, inside outer loop iteration $k$ with domain factor $\boldsymbol{\omega}^{m(k)}$, for each $i \in \mathcal{N}$, with probability at least $1 - \delta$:

$$\left| Q_i(\mathbf{s}, \mathbf{a}, \boldsymbol{\omega}) - \hat{Q}_i^T(\mathbf{s}_{\mathcal{N}_i^\kappa}, \mathbf{a}_{\mathcal{N}_i^\kappa}, \hat{\boldsymbol{\omega}}_{\mathcal{N}_i^\kappa}) \right| \leq \frac{C_a}{\sqrt{T + t_0}} + \frac{C_a'}{T + t_0} + \frac{2c\rho^{\kappa+1}}{(1-\gamma)^2} + C_{\boldsymbol{\omega}}' \sqrt{\frac{\log(nT_e/\delta)}{T_e}}.$$

where $C_a := \frac{6\bar{\epsilon}}{1-\sqrt{\gamma}} \sqrt{\frac{\tau h}{\sigma} [\log(\frac{2\tau T^2}{\delta}) + f(\kappa) \log SA]}$, $C_a' := \frac{2}{1-\sqrt{\gamma}} \max(\frac{16\bar{\epsilon}h\tau}{\sigma}, \frac{2\bar{r}}{1-\gamma}(\tau + t_0))$, with $\bar{\epsilon} := 4\frac{\bar{r}}{1-\gamma} + 2\bar{r}$, and $C_{\boldsymbol{\omega}}' := L_Q C_{\boldsymbol{\omega}} \sqrt{D_\Omega}$.

*Proof sketch of Theorem 2.* The proof follows a similar structure to the Theorem 5 in [24], with modifications to account for domain factors. First, we decompose

$$Q_i(\mathbf{z}, \boldsymbol{\omega}) - \hat{Q}_i^T(\mathbf{z}_{\mathcal{N}_i^\kappa}, \hat{\boldsymbol{\omega}}_{\mathcal{N}_i^\kappa}) = Q_i(\mathbf{z}, \boldsymbol{\omega}) - Q_i(\mathbf{z}, \hat{\boldsymbol{\omega}}) + Q_i(\mathbf{z}, \hat{\boldsymbol{\omega}}) - \hat{Q}_i^T(\mathbf{z}_{\mathcal{N}_i^\kappa}, \hat{\boldsymbol{\omega}}_{\mathcal{N}_i^\kappa}).$$

The first difference term accounts for the error due to domain factor estimation. Using the domain factor identifiability in Proposition 5 and the Lipschitz continuity of $Q$ with respect to $\boldsymbol{\omega}$ (Assumption 5 (ii)), we have

$$Q_i(\mathbf{z}, \boldsymbol{\omega}) - Q_i(\mathbf{z}, \hat{\boldsymbol{\omega}}) \leq L_Q \|\boldsymbol{\omega} - \hat{\boldsymbol{\omega}}\|_2 \leq L_Q \delta_{\boldsymbol{\omega}}(T_e).$$

For $\hat{\boldsymbol{\omega}} = \hat{\boldsymbol{\omega}}^{m(k)}$, the critic update follows:

$$\hat{Q}_i^t(\mathbf{s}_{\mathcal{N}_i^\kappa}(t-1), \mathbf{a}_{\mathcal{N}_i^\kappa}(t-1), \hat{\boldsymbol{\omega}}_{\mathcal{N}_i^\kappa}) = (1 - \alpha_{t-1})\hat{Q}_i^{t-1}(\mathbf{s}_{\mathcal{N}_i^\kappa}(t-1), \mathbf{a}_{\mathcal{N}_i^\kappa}(t-1), \hat{\boldsymbol{\omega}}_{\mathcal{N}_i^\kappa})$$
$$+ \alpha_{t-1}(r_i(t-1) + \gamma \hat{Q}_i^{t-1}(\mathbf{s}_{\mathcal{N}_i^\kappa}(t), \mathbf{a}_{\mathcal{N}_i^\kappa}(t), \hat{\boldsymbol{\omega}}_{\mathcal{N}_i^\kappa})).$$

The steps from the original proof apply with the augmentation of $\boldsymbol{\omega}_{\mathcal{N}_i^\kappa}^m$ to the state-action representation. The key difference is that we need to account for the error in domain factor estimation. Let $\xi_t := \sup_{(\mathbf{z}) \in \mathcal{S} \times \mathcal{A}} |Q_i(\mathbf{z}, \hat{\boldsymbol{\omega}}) - \hat{Q}_i^t(\mathbf{z}_{\mathcal{N}_i^\kappa}, \hat{\boldsymbol{\omega}}_{\mathcal{N}_i^\kappa})|$.

Following the decomposition in the original proof (Lemma 7 in [24]), we have:

$$\xi_t \leq \tilde{\beta}_{\tau-1,t}\xi_\tau + \gamma \sup_{z_{\mathcal{N}_i^\kappa}} \sum_{k=\tau}^{t-1} b_{k,t}(\mathbf{z}_{\mathcal{N}_i^\kappa})\xi_k + \frac{2c\rho^{\kappa+1}}{1-\gamma} + \left\| \sum_{k=\tau}^{t-1} \alpha_k \tilde{B}_{k,t}\epsilon_k \right\|_\infty + \left\| \sum_{k=\tau}^{t-1} \alpha_k \tilde{B}_{k,t}\phi_k \right\|_\infty.$$

Following the same induction argument as in the original proof and combining all terms, we get the stated bound. $\square$

## G.3 Proof of Theorem 3

**Theorem 6** (Convergence). Under Assumptions 1-6, for any $\delta \in (0,1)$, $K \geq 3$, suppose the critic stepsize $\alpha_t = \frac{h}{t+t_0}$ and $\eta_k = \frac{\eta}{\sqrt{k+1}}$ with $\eta \leq \frac{1}{6L'}$. For sufficiently large the inner loop length $T$ such that $T+1 \geq \frac{\log(c(1-\gamma)/\bar{r})+(\kappa+1)\log_\gamma \rho}{\log \gamma}$ and $\frac{C_a(\delta/2nK,T)}{\sqrt{T+t_0}} + \frac{C'_a}{T+t_0} \leq \frac{2c\rho^{\kappa+1}}{(1-\gamma)^2}$, Then, with probability at least $1 - \delta$, when domain factors are estimated with $T_e$ trajectories:

$$
\frac{\sum_{k=0}^{K-1}\eta_k\|\nabla J(\theta(k))\|^2}{\sum_{k=0}^{K-1}\eta_k} \leq \frac{\frac{2\bar{r}}{\eta(1-\gamma)} + \frac{8\bar{r}^2L^2}{(1-\gamma)^4}\sqrt{\log K \log \frac{4}{\delta}} + \frac{240\bar{r}^2L'L^2}{(1-\gamma)^4}\eta\log K}{\sqrt{K+1}}
$$
$$
+ \frac{12L^2c\bar{r}}{(1-\gamma)^5}\rho^{\kappa+1} + \frac{2\bar{r}LL_JC_{\boldsymbol{\omega}}}{(1-\gamma)^2}\sqrt{\frac{\log(4nM/\delta)}{T_e}} + \frac{2C_5(\bar{r}L+L_\theta)d^\theta}{(1-\gamma)^2}\sqrt{\frac{\log(4M/\delta)}{M}}.
$$

*Proof sketch of Theorem 3.* We adapt the proof of Theorem 4 in [24] to account for domain factors. The key difference is that the policy gradient now depends on the domain factors, and we need to account for the error in domain factor estimation.

Note that for any $\theta \in \Theta$, we have

$$
\nabla_{\theta_i}\log\pi^\theta(\mathbf{a}|\mathbf{s},\boldsymbol{\omega}) = \nabla_{\theta_i}\sum_{j\in\mathcal{N}}\log\pi_j^{\theta_j}(\mathbf{a}_j|\mathbf{s}_j,\boldsymbol{\omega}_j) = \nabla_{\theta_i}\log\pi_i^{\theta_i}(\mathbf{a}_i|\mathbf{s}_i,\boldsymbol{\omega}_j).
$$

Let us define the approximation of the policy gradient and the true policy gradient

$$
\hat{g}_i(k) = \sum_{t=0}^{T}\gamma^t\frac{1}{n}\sum_{j\in\mathcal{N}_i^\kappa}\hat{Q}_j^T(\mathbf{s}_{\mathcal{N}_j^\kappa}(t),\mathbf{a}_{\mathcal{N}_j^\kappa}(t),\hat{\boldsymbol{\omega}}_{\mathcal{N}_j^\kappa}^{m(k)})\nabla_{\theta_i}\log\pi_i^{\theta_i(k)}(\mathbf{a}_i(t)|\mathbf{s}_i(t),\hat{\boldsymbol{\omega}}_i^{m(k)})
$$

$$
g_i(k) := \sum_{t=0}^{T}\gamma^t\frac{1}{n}\sum_{j\in\mathcal{N}_i^\kappa}Q_j^{\theta(k)}(\mathbf{s}(t),\mathbf{a}(t),\hat{\boldsymbol{\omega}}^{m(k)})\nabla_{\theta_i}\log\pi_i^{\theta_i(k)}(\mathbf{a}_i(t)|\mathbf{s}_i(t),\hat{\boldsymbol{\omega}}_i^{m(k)})
$$

$$
h_i(k) := \sum_{m=1}^{M}\frac{1}{M}\sum_{t=0}^{T}\mathbb{E}_{\mathbf{s}\sim\rho_t^{\theta(k)},\mathbf{a}\sim\pi^{\theta(k)}(\cdot|\mathbf{s},\hat{\boldsymbol{\omega}}^m)}\left[\gamma^t\frac{1}{n}\sum_{j\in\mathcal{N}_i^\kappa}Q_j^{\theta(k)}(\mathbf{s},\mathbf{a},\hat{\boldsymbol{\omega}}^m)\nabla_{\theta_i}\log\pi_i^{\theta_i(k)}(\mathbf{a}_i|\mathbf{s}_i,\hat{\boldsymbol{\omega}}_i^m)\right]
$$

$$
\nabla_{\theta_i}J(\theta(k),\hat{\mathcal{D}}^M) := \sum_{m=1}^{M}\frac{1}{M}\sum_{t=0}^{\infty}\mathbb{E}_{\mathbf{s}\sim\rho_t^{\theta(k)},\mathbf{a}\sim\pi^{\theta(k)}(\cdot|\mathbf{s},\hat{\boldsymbol{\omega}}^m)}\left[\gamma^tQ^{\theta(k)}(\mathbf{s},\mathbf{a},\hat{\boldsymbol{\omega}}^m)\nabla_{\theta_i}\log\pi^{\theta(k)}(\mathbf{a}|\mathbf{s},\hat{\boldsymbol{\omega}}^m)\right]
$$
$$
= \sum_{m=1}^{M}\frac{1}{M}\nabla_{\theta_i}J(\theta(k),\hat{\boldsymbol{\omega}}^m)
$$

$$
\nabla_{\theta_i}J(\theta(k),\mathcal{D}^M) := \sum_{m=1}^{M}\frac{1}{M}\sum_{t=0}^{\infty}\mathbb{E}_{\mathbf{s}\sim\rho_t^{\theta(k)},\mathbf{a}\sim\pi^{\theta(k)}(\cdot|\mathbf{s},\boldsymbol{\omega}^m)}\left[\gamma^tQ^{\theta(k)}(\mathbf{s},\mathbf{a},\boldsymbol{\omega}^m)\nabla_{\theta_i}\log\pi^{\theta(k)}(\mathbf{a}|\mathbf{s},\boldsymbol{\omega}^m)\right]
$$
$$
= \sum_{m=1}^{M}\frac{1}{M}\nabla_{\theta_i}J(\theta(k),\boldsymbol{\omega}^m)
$$

$$
\nabla_{\theta_i}J(\theta(k),\mathcal{D}) := \mathbb{E}_{\boldsymbol{\omega}\sim\mathcal{D}}\sum_{t=0}^{\infty}\mathbb{E}_{\mathbf{s}\sim\rho_t^{\theta(k)},\mathbf{a}\sim\pi^{\theta(k)}(\cdot|\mathbf{s},\boldsymbol{\omega})}\left[\gamma^tQ^{\theta(k)}(\mathbf{s},\mathbf{a},\boldsymbol{\omega})\nabla_{\theta_i}\log\pi^{\theta(k)}(\mathbf{a}|\mathbf{s},\boldsymbol{\omega})\right]
$$
$$
= \mathbb{E}_{\boldsymbol{\omega}\sim\mathcal{D}}\nabla_{\theta_i}J(\theta(k),\boldsymbol{\omega}) = \nabla_{\theta_i}\mathbb{E}_{\boldsymbol{\omega}\sim\mathcal{D}}J(\theta(k),\boldsymbol{\omega}) = \nabla_{\theta_i}J(\theta(k)).
$$

**Lemma 13.** The following holds almost surely,

$$
\max(\|\hat{g}(k)\|, \|g(k)\|, \|h(k)\|, \|\nabla_\theta J(\theta(k),\hat{\mathcal{D}}^M)\|, \|\nabla_\theta J(\theta(k),\mathcal{D}^M)\|, \|\nabla_\theta J(\theta(k))\|) \leq \frac{\bar{r}L}{(1-\gamma)^2}.
$$

We decompose the error between $\hat{g}(k)$ and the true gradient as follows

$$
\hat{g}(k) = \underbrace{\hat{g}(k) - g(k)}_{e^1(k)} + \underbrace{g(k) - h(k)}_{e^2(k)} + \underbrace{h(k) - \nabla_\theta J(\theta(k), \hat{\mathcal{D}}^M)}_{e^3(k)} + \underbrace{\nabla_\theta J(\theta(k), \hat{\mathcal{D}}^M) - \nabla_\theta J(\theta(k), \mathcal{D}^M)}_{e^4(k)}
$$
$$
+ \underbrace{\nabla_\theta J(\theta(k), \mathcal{D}^M) - \nabla_{\theta_i} J(\theta(k), \mathcal{D})}_{e^5(k)} + \nabla_\theta J(\theta(k)).
$$

$$(12)$$

We will bound each term separately and combine these bounds to prove Theorem 3.

**Bounds on $e^1(k)$.** The following lemmas follows similar steps as in the proof [24].

**Lemma 14** (Lemma 14, [24]). When $T$ is large enough s.t.

$$
\frac{C_a\left(\frac{\delta}{2nM}, T\right)}{\sqrt{T + t_0}} + \frac{C_a'}{T + t_0} \leq \frac{2c\rho^{\kappa+1}}{(1-\gamma)^2}, \quad \text{where}
$$

$$
C_a(\delta, T) = \frac{6\bar{\epsilon}}{1 - \sqrt{\gamma}}\sqrt{\frac{\tau h}{\sigma}}\left[\log\left(\frac{2\tau T^2}{\delta}\right) + f(\kappa)\log SA\right], \quad C_a' = \frac{2}{1 - \sqrt{\gamma}}\max\left(\frac{16\bar{\epsilon}h\tau}{\sigma}, \frac{2\bar{r}}{1-\gamma}(\tau + t_0)\right),
$$

with $\bar{\epsilon} = 4\frac{\bar{r}}{1-\gamma} + 2\bar{r}$, then we have with probability at least $1 - \frac{\delta}{4}$,

$$
\sup_{0 \leq k \leq K-1} \|e^1(k)\| \leq \frac{4cL\rho^{\kappa+1}}{(1-\gamma)^3}.
$$

**Bounds on $e^2(k)$.** Let $\mathcal{G}_k$ be the $\sigma$-algebra generated by the trajectories in the first $k$ outer-loop iterations. In particular, let $\mathcal{G}_0$ be the $\sigma$-algebra generated the trajectories in the first phase of Algorithm 1. Note that $\boldsymbol{\omega}^{1:M}$ and $\hat{\boldsymbol{\omega}}^{1:M}$ are $\mathcal{G}_0$-measurable. In addition, $\theta(k)$ is $\mathcal{G}_{k-1}$-measurable, and so is $h_i(k)$. Further, by the way that the trajectory $\{(s(t), a(t))\}_{t=0}^T$ is generated, we have $\mathbb{E}[g(k)|\mathcal{G}_{k-1}] = h(k)$. As such, $\eta_k\langle \nabla J(\theta(k)), e^2(k)\rangle$ is a martingale difference sequence w.r.t. $\mathcal{G}_k$, and we have the following bound which is a direct consequence of Azuma-Hoeffding bound.

**Lemma 15** (Adapted Lemma 15 [24]). With probability at least $1 - \delta/4$, we have

$$
\left|\sum_{k=0}^{K-1} \eta_k\langle \nabla J(\theta(k)), e^2(k)\rangle\right| \leq \frac{2\bar{r}^2 L^2}{(1-\gamma)^4}\sqrt{2\sum_{k=0}^{K-1} \eta_k^2 \log\frac{8}{\delta}}.
$$

**Bounds on $e^3(k)$.**

**Lemma 16** (Adapted Lemma 16, [24]). When $T + 1 \geq \dfrac{\log\left(\dfrac{c(1-\gamma)}{\bar{r}}\right) + (\kappa + 1)\log\rho}{\log\gamma}$, we have almost surely,

$$
\|e^3(k)\| \leq 2\frac{Lc}{(1-\gamma)}\rho^{\kappa+1}.
$$

**Bounds on $e^4(k)$.** Since the error between $\hat{\boldsymbol{\omega}}^m$ and $\boldsymbol{\omega}^m$ is bounded as $\|\hat{\boldsymbol{\omega}}^m - \boldsymbol{\omega}^m\|_2 \leq \delta_{\boldsymbol{\omega}}(T_e) = C_{\boldsymbol{\omega}}\sqrt{\frac{\log(n/\delta)}{T_e}}$ with probability at least $1 - \delta$. Using Assumption 5 (iii), we have with probability at least $1 - \delta/4$

$$
\forall k, e^4(k) = \nabla_\theta J(\theta(k), \hat{\mathcal{D}}^M) - \nabla_\theta J(\theta(k), \mathcal{D}^M) \leq L_J \sum_{m=1}^M \frac{1}{M}\|\hat{\boldsymbol{\omega}}^m - \boldsymbol{\omega}^m\|_2
$$

$$(13)$$

$$
\leq L_J C_{\boldsymbol{\omega}}\sqrt{\frac{\log(4nM/\delta)}{T_e}}.
$$

**Bounds on $e^5(k)$.** We first state the uniform concentration bound for vector-valued functions.

**Lemma 17.** Let $(x_1, x_2, \ldots, x_M)$ be i.i.d. samples from a distribution $D$ on $\mathcal{X}$. Let $\Theta \subset \mathbb{R}^p$, and consider $f : \mathcal{X} \times \Theta \to \mathbb{R}^d$ with the following properties: (i) For all $j = 1, \ldots, d$, $|f_j(x, \theta)| \leq B$ for all $(x, \theta)$; (ii) For all $x \in \mathcal{X}$ and $j = 1, \ldots, d$, $|f_j(x, \theta) - f_j(x, \theta')| \leq L \cdot \|\theta - \theta'\|_2$. Then, with probability at least $1 - \delta/4$,

$$\sup_{\theta \in \Theta} \left\| \frac{1}{M} \sum_{i=1}^{M} f(x_i, \theta) - \mathbb{E}[f(X, \theta)] \right\|_2 = \mathcal{O}\left( (B + L) \sqrt{\frac{dp \log M + \log(4/\delta)}{M}} \right),$$

where the hidden constant is universal.

Observe that for $\mathbb{E}\left[ \nabla_\theta J(\theta, \mathcal{D}^M) \right] = \nabla_\theta J(\theta)$. Using Lemma 17, we have with probability at least $1 - \delta/4$,

$$\sup_k \left\| e^5(k) \right\| \leq \sup_{k, \theta \in \Theta} \left\| \nabla_\theta J(\theta, \mathcal{D}^M) - \nabla_\theta J(\theta) \right\|$$

$$\leq \mathcal{O}\left( \left( \frac{\bar{r} L}{(1 - \gamma)^2} + L_\theta \right) d^\theta \sqrt{\frac{\log(4M/\delta)}{M}} \right) \tag{14}$$

$$\leq \mathcal{O}\left( \frac{(\bar{r} L + L_\theta) d^\theta}{(1 - \gamma)^2} \sqrt{\frac{\log(4M/\delta)}{M}} \right)$$

where the second inequality is due to Lemma 13.

**Putting all bounds together.** With the above bounds on all $e^i(k)$, we are now ready to prove the Theorem 3. Since $\nabla J(\theta)$ is $L'$-Lipschitz continuous, we have

$$J(\theta(k+1)) \geq J(\theta(k)) + \langle \nabla J(\theta(k)), \theta(k+1) - \theta(k) \rangle - \frac{L'}{2} \|\theta(k+1) - \theta(k)\|^2$$
$$= J(\theta(k)) + \eta_k \langle \nabla J(\theta(k)), \hat{g}(k) \rangle - \frac{L' \eta_k^2}{2} \|\hat{g}(k)\|^2. \tag{15}$$

Using the decomposition of $\hat{g}(k)$ in Equation 12, we get,

$$\|\hat{g}(k)\|^2 \leq 6\|e^1(k)\|^2 + 6\|e^2(k)\|^2 + 6\|e^3(k)\|^2 + 6\|e^4(k)\|^2 + 6\|e^5(k)\|^2 + 6\|\nabla J(\theta(k))\|^2.$$

We bound $\langle \nabla J(\theta(k)), \hat{g}(k) \rangle$ as follows

$$\langle \nabla J(\theta(k)), \hat{g}(k) \rangle = \|\nabla J(\theta(k))\|^2 + \langle \nabla J(\theta(k)), \sum_{j=1}^{5} e^j(k) \rangle \geq \|\nabla J(\theta(k))\|^2 + \langle \nabla J(\theta(k)), e^2(k) \rangle$$
$$- \|\nabla J(\theta(k))\|(\|e^1(k)\| + \|e^3(k)\| + \|e^4(k)\| + \|e^5(k)\|).$$

Plugging the above bounds on $\|\hat{g}(k)\|^2$ and $\langle \nabla J(\theta(k)), \hat{g}(k) \rangle$ into Equation 15, we have

$$J(\theta(k+1)) \geq J(\theta(k)) + (\eta_k - 3L'\eta_k^2)\|\nabla J(\theta(k))\|^2 + \eta_k \epsilon_{k,0} - \eta_k \epsilon_{k,1} - \eta_k^2 \epsilon_{k,2},$$

where $\epsilon_{k,0} := \langle \nabla J(\theta(k)), e^2(k) \rangle$, $\epsilon_{k,1} := \|\nabla J(\theta(k))\|(\|e^1(k)\| + \|e^3(k)\| + \|e^4(k)\| + \|e^5(k)\|)$, $\epsilon_{k,2} := 3L' \sum_{j=1}^{5} \|e^j(k)\|^2$. It follows from telescoping sum that

$$J(\theta(K)) - J(\theta(0)) \geq \sum_{k=0}^{K-1} (\eta_k - 3L'\eta_k^2)\|\nabla J(\theta(k))\|^2 + \sum_{k=0}^{K-1} \eta_k \epsilon_{k,0} - \sum_{k=0}^{K-1} \eta_k \epsilon_{k,1} - \sum_{k=0}^{K-1} \eta_k^2 \epsilon_{k,2}$$

$$\geq \sum_{k=0}^{K-1} \frac{1}{2}\eta_k\|\nabla J(\theta(k))\|^2 + \sum_{k=0}^{K-1} \eta_k \epsilon_{k,0} - \sum_{k=0}^{K-1} \eta_k \epsilon_{k,1} - \sum_{k=0}^{K-1} \eta_k^2 \epsilon_{k,2},$$

where we use $\eta_k - 3L'\eta_k^2 = \eta_k(1 - 3L'\eta_k) \geq \frac{1}{2}\eta_k$ given that $\eta_k \leq \frac{1}{6L'}$. After rearranging, we get

$$\sum_{k=0}^{K-1} \frac{1}{2}\eta_k\|\nabla J(\theta(k))\|^2 \leq J(\theta(K)) - J(\theta(0)) - \sum_{k=0}^{K-1} \eta_k \epsilon_{k,0} + \sum_{k=0}^{K-1} \eta_k \epsilon_{k,1} + \sum_{k=0}^{K-1} \eta_k^2 \epsilon_{k,2}. \tag{16}$$

We now apply our bounds on each $e^j(k)$. By Lemma 15, we have with probability $1 - \frac{\delta}{2}$,

$$\sum_{k=0}^{K-1} \eta_k \epsilon_{k,0} \leq \frac{2\bar{r}^2 L^2}{(1-\gamma)^4} \sqrt{2 \sum_{k=0}^{K-1} \eta_k^2 \log \frac{4}{\delta}}. \tag{17}$$

By Lemma 14, Lemma 16, Equation 13 and Equation 14, we have with probability $1 - \frac{\delta}{2}$,

$$
\begin{aligned}
\sup_{k \leq K-1} \epsilon_{k,1} &\leq \frac{\bar{r}L}{(1-\gamma)^2} \left( \sup_{k \leq K-1} \|e^1(k)\| + \sup_{k \leq K-1} \|e^3(k)\| + \sup_{k \leq K-1} \|e^4(k)\| + \sup_{k \leq K-1} \|e^5(k)\| \right) \\
&\leq \frac{\bar{r}L}{(1-\gamma)^2} \left( \frac{4cL\rho^{\kappa+1}}{(1-\gamma)^3} + 2\frac{Lc}{(1-\gamma)}\rho^{\kappa+1} + L_J C_{\boldsymbol{\omega}}\sqrt{\frac{\log(4nM/\delta)}{T_e}} + C_5 \frac{(\bar{r}L + L_\theta)d^\theta}{(1-\gamma)^2}\sqrt{\frac{\log(4M/\delta)}{M}} \right) \\
&\leq \frac{6L^2 c\bar{r}}{(1-\gamma)^5}\rho^{\kappa+1} + \frac{\bar{r}LL_J C_{\boldsymbol{\omega}}}{(1-\gamma)^2}\sqrt{\frac{\log(4nM/\delta)}{T_e}} + \frac{C_5(\bar{r}L + L_\theta)d^\theta}{(1-\gamma)^2}\sqrt{\frac{\log(4M/\delta)}{M}}.
\end{aligned}
\tag{18}
$$

By Lemma 13, we have almost surely,

$$\max_{1 \leq j \leq 5}(\|e^j(k)\|\|\|) \leq 2\frac{\bar{r}L}{(1-\gamma)^2},$$

and hence,

$$\sup_{k \leq K-1} \epsilon_{k,2} = 3L' \sum_{j=1}^{5} \|e^j(m)\|^2 \leq \frac{60\bar{r}^2 L' L^2}{(1-\gamma)^4}.$$

Using a union bound, we have with probability $1 - \delta$, all three events hold, thus

$$
\begin{aligned}
&\frac{\sum_{k=0}^{K-1} \eta_k \|\nabla J(\theta(k))\|^2}{\sum_{k=0}^{K-1} \eta_k} \\
&\leq \frac{2(J(\theta(K)) - J(\theta(0))) + 2\left|\sum_{k=0}^{K-1} \eta_k \epsilon_{k,0}\right| + 2\sup_k \epsilon_{k,2} \sum_{k=0}^{K-1} \eta_k^2}{\sum_{k=0}^{K-1} \eta_k} + 2\sup_k \epsilon_{k,1} \\
&\leq \frac{2(J(\theta(K)) - J(\theta(0))) + \frac{4\bar{r}^2 L^2}{(1-\gamma)^4}\sqrt{2\sum_{k=0}^{K-1} \eta_k^2 \log \frac{4}{\delta}} + \frac{120\bar{r}^2 L' L^2}{(1-\gamma)^4}\sum_{k=0}^{K-1} \eta_k^2}{\sum_{k=0}^{K-1} \eta_k} \\
&\quad + \frac{12L^2 c\bar{r}}{(1-\gamma)^5}\rho^{\kappa+1} + \frac{2\bar{r}LL_J C_{\boldsymbol{\omega}}}{(1-\gamma)^2}\sqrt{\frac{\log(4nM/\delta)}{T_e}} + \frac{2C_5(\bar{r}L + L_\theta)d^\theta}{(1-\gamma)^2}\sqrt{\frac{\log(4M/\delta)}{M}}.
\end{aligned}
$$

Since $\eta_k = \frac{\eta}{\sqrt{k+1}}$, we have

$$\sum_{k=0}^{K-1} \eta_k > 2\eta(\sqrt{K+1}-1) \geq \eta\sqrt{K+1} \quad \text{and} \quad \sum_{k=0}^{K-1} \eta_k^2 < \eta^2(1+\log K) < 2\eta^2 \log K \quad \text{(using } K \geq 3\text{)}.$$

Further we use the bound $J(\theta(K)) \leq \frac{\bar{r}}{1-\gamma}$ and $J(\theta(0)) \geq 0$ almost surely. Combining these results, we get with probability $1 - \delta$,

$$
\begin{aligned}
\frac{\sum_{k=0}^{K-1} \eta_k \|\nabla J(\theta(k))\|^2}{\sum_{k=0}^{K-1} \eta_k} &\leq \frac{\frac{2\bar{r}}{\eta(1-\gamma)} + \frac{8\bar{r}^2 L^2}{(1-\gamma)^4}\sqrt{\log K \log \frac{4}{\delta}} + \frac{240\bar{r}^2 L' L^2}{(1-\gamma)^4}\eta \log K}{\sqrt{K+1}} \\
&\quad + \frac{12L^2 c\bar{r}}{(1-\gamma)^5}\rho^{\kappa+1} + \frac{2\bar{r}LL_J C_{\boldsymbol{\omega}}}{(1-\gamma)^2}\sqrt{\frac{\log(4nM/\delta)}{T_e}} + \frac{2C_5(\bar{r}L + L_\theta)d^\theta}{(1-\gamma)^2}\sqrt{\frac{\log(4M/\delta)}{M}}.
\end{aligned}
$$

$\square$

### G.3.1 Empirical convergence

**Corollary 4** (Empirical convergence). Under the same setup of Theorem 3, with probability at least $1 - \delta$, when domain factors are estimated with $T_e$ trajectories:

$$\frac{\sum_{k=0}^{K-1} \eta_k \|\nabla J(\theta(k), \mathcal{D}^M)\|^2}{\sum_{k=0}^{K-1} \eta_k} \leq \frac{\frac{2\bar{r}}{\eta(1-\gamma)} + \frac{8\bar{r}^2 L^2}{(1-\gamma)^4}\sqrt{\log K \log \frac{4}{\delta}} + \frac{240\bar{r}^2 L' L^2}{(1-\gamma)^4}\eta \log K}{\sqrt{K+1}}$$

$$+ \frac{12 L^2 c\bar{r}}{(1-\gamma)^5}\rho^{\kappa+1} + \frac{2\bar{r} L L_J C_{\boldsymbol{\omega}}}{(1-\gamma)^2}\sqrt{\frac{\log(4nM/\delta)}{T_e}}$$

$$\frac{\sum_{k=0}^{K-1} \eta_k \|\nabla J(\theta(k), \hat{\mathcal{D}}^M)\|^2}{\sum_{k=0}^{K-1} \eta_k} \leq \frac{\frac{2\bar{r}}{\eta(1-\gamma)} + \frac{8\bar{r}^2 L^2}{(1-\gamma)^4}\sqrt{\log K \log \frac{4}{\delta}} + \frac{240\bar{r}^2 L' L^2}{(1-\gamma)^4}\eta \log K}{\sqrt{K+1}}$$

$$+ \frac{12 L^2 c\bar{r}}{(1-\gamma)^5}\rho^{\kappa+1}.$$

### G.4 Proof of Theorem 4

*Proof.* We use the Lipschitz continuity of $J(\theta, \boldsymbol{\omega}'; \boldsymbol{\omega})$ with respect to $\boldsymbol{\omega}'$

$$J(\theta(K), \hat{\boldsymbol{\omega}}^{M+1}; \boldsymbol{\omega}^{M+1}) - J(\theta(K), \boldsymbol{\omega}^{M+1}; \boldsymbol{\omega}^{M+1}) \geq -L_{\boldsymbol{\omega}'} \left\|\hat{\boldsymbol{\omega}}^{M+1} - \boldsymbol{\omega}^{M+1}\right\|$$

$$\geq -L_{\boldsymbol{\omega}'} C_{\boldsymbol{\omega}}\sqrt{\frac{\log(n/\delta)}{T_a}}.$$

It follows that

$$\mathbb{E}\left[J(\theta(K), \hat{\boldsymbol{\omega}}^{M+1}; \boldsymbol{\omega}^{M+1})|\theta(K)\right] \geq \mathbb{E}\left[J(\theta(K), \boldsymbol{\omega}^{M+1})|\theta(K)\right] - L_{\boldsymbol{\omega}'} C_{\boldsymbol{\omega}}\sqrt{\frac{\log(n/\delta)}{T_a}}$$

$$= J(\theta(K)) - L_{\boldsymbol{\omega}'} C_{\boldsymbol{\omega}}\sqrt{\frac{\log(n/\delta)}{T_a}}.$$

$\square$

## H Comprehensive Experiments

We begin by clarifying the distinction between the *meta-RL* and *multi-task RL* paradigms to highlight the key differences between our algorithmic design and baseline methods.

**Clarification: Meta-RL vs. Multi-task RL** The goal of multi-task RL [37] is to learn a single task-conditioned policy that maximizes the average expected return across all source domains drawn from a domain distribution. However, multi-task RL does not provide any guarantee of rapid few-shot adaptation to new, unseen domains. In contrast, meta-RL explicitly aims to *learn to adapt*: it leverages the set of source domains to train a policy that can be efficiently adapted to a new domain using only a small number of trajectories at test time.

Therefore, GSAC falls under the meta-RL framework. Our objective is to achieve fast adaptation in a new domain with limited data. Specifically, we train a shared, domain-factor-conditioned policy across source domains and, during adaptation, estimate the domain factors $\hat{\omega}$ from only a few trajectories to enable immediate policy deployment without additional fine-tuning.

### H.1 Wireless communication

**Environment setup.** The network consists of $n$ users positioned on a 2D grid. Each user maintains a queue of packets with a fixed deadline $d_i = 2$. At every timestep, a new packet arrives at user $i$ with probability $p_i \sim \text{Unif}[0, 1]$. The user either transmits the earliest packet to a randomly selected

access point (AP) $y_i(t) \in Y_i$, or remains idle. A transmission to AP $y_i(t)$ succeeds with probability $q_i \sim \text{Unif}[0,1]$ if no other user simultaneously transmits to the same AP; otherwise, a collision occurs and no transmission succeeds. Each successful transmission yields a reward of 1.

The local state of each user $\mathbf{s}_i(t) = (b_{i,1}(t), \ldots, b_{i,d_i}(t), z_{i,1}(t), z_{i,2}(t))$. $b_i(t) \in \{0,1\}^{d_i}$ encodes queue status by deadline bins (e.g., $b_i(t) = (1,1,0)$ means the first two deadline bins are occupied while the third is empty). $z_{i,1}(t)$ and $z_{i,2}(t)$ are irrelevant components to decision quality. $z_{i,1}(t)$ is the channel quality indicator, which does not influence the packet success probability $q_i$, as $q_i$ depends solely on collisions. $z_{i,2}(t)$ is the grid coordinates, which adds no useful information because the connectivity graph already encodes all relevant interactions.

The action space is $\mathcal{A}_i = Y_i \cup \{\texttt{null}\}$, where $\texttt{null}$ denotes the idle action. The interaction graph is defined by overlapping access point sets: two users are neighbors if they share at least one AP. This induces local dependencies, which define each agent's $\kappa$-hop neighborhood for state, action, and domain-factor inputs.

Figure 4 visualizes the communication graph induced by different grid sizes: $3 \times 3$, $4 \times 4$, and $5 \times 5$. Each user is placed at a grid cell and is connected to APs located at the corners of grid blocks. Users that share an AP form communication links, resulting in a structured interaction graph. This setup highlights the challenge of scalability in networked MARL, as larger grids yield significantly higher local complexity and increased neighborhood sizes. Figure 5 illustrates the local neighborhood (i.e., $\kappa$-hop graph connectivity) for a representative agent in each grid size. These visualizations demonstrate how local dependencies expand with increasing grid sizes. The number of neighbors within $\kappa = 1$ grows with the grid size, resulting in higher-dimensional input spaces for both state and domain-factor features.

**Benefits of ACR.** The ACR procedure automatically prunes such irrelevant or redundant variables, retaining only those that causally influence the local reward or dynamics. This yields a non-trivial reduction in local dimensionality. For example, consider a $4 \times 4$ grid with $d_i = 2$ and a neighborhood size $|N_i| = 5$:

- Raw (truncation-only): $|N_i| \times (d_i + 2) = 5 \times 4 = 20$ features.
- With ACR: $|N_i| \times d_i = 5 \times 2 = 10$ features.

This 50% shrinkage directly translates into:

- Smaller critics and lower compute: reduced local state–action support and table size.
- Faster learning: lower variance in actor updates and tighter constants in our bounds.
- Accuracy preserved: approximation error remains controlled (see Proposition 1-2).
- Improved sample efficiency: by effectively increasing the minimum visitation $\sigma$ and reducing the mixing constant $\tau$ (Assumption 2), ACRs tighten the constants in the convergence guarantee (Theorem 2).

**Evaluation protocol.** We generate $M = 3$ source domains by sampling domain factors $p_i$ from $\{0.2, 0.5, 0.8\}$, while the target domain uses $p_{\text{target}} = 0.65$ unless otherwise specified. In each domain, $\texttt{GSAC}$ is trained for $K$ outer iterations (variable depending on convergence), with a time horizon of $T = 10$. We use a constant stepsize of $\alpha = 0.1$ for the critic and $\eta = 0.01$ for the actor. The softmax temperature is set to $\tau = 0.5$, and the discount factor is $\gamma = 0.95$. For domain factor estimation, we collect $T_e = 20$ trajectories per domain. To evaluate adaptation in the target domain, we compare our algorithm against three baselines: $\texttt{SAC-MTL}$, $\texttt{SAC-FT}$, $\texttt{SAC-LFS}$.

**Domain shift sensitivity.** To assess generalization under varying domain factors, we vary the target packet arrival rate with $p_{\text{target}} \in \{0.6, 0.65, 0.7\}$. As shown in Figure 7, $\texttt{GSAC}$ consistently adapts to new domain settings with minimal performance degradation, even when the target domain differs significantly from the training domains. Notably, it outperforms all three baselines across all configurations. Moreover, Figures 2 and 7 demonstrate that $\texttt{GSAC}$ can rapidly adapt to new environments using only a few trajectories.

Across all settings, $\texttt{GSAC}$ demonstrates: (i) consistent improvement over training iterations; (ii) fast adaptation from a few trajectories; and (iii) robust performance under domain shifts. These findings

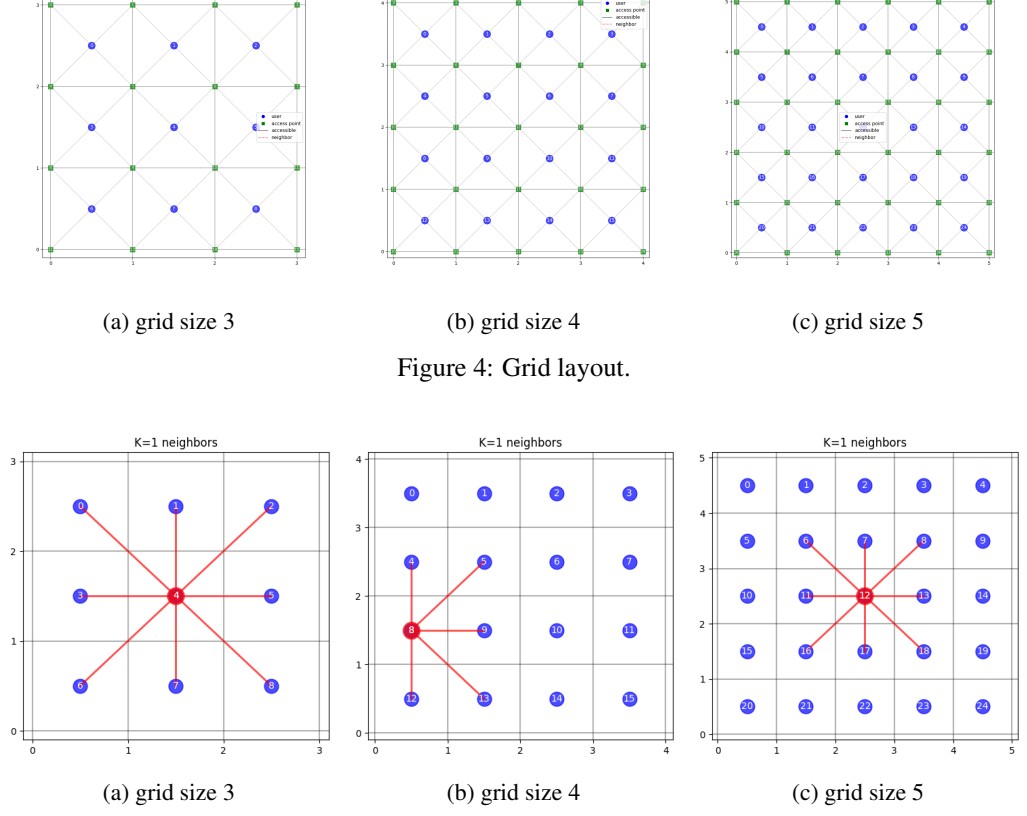

<p align="center">(a) grid size 3       (b) grid size 4       (c) grid size 5</p>

<p align="center">Figure 4: Grid layout.</p>

<p align="center">(a) grid size 3       (b) grid size 4       (c) grid size 5</p>

<p align="center">Figure 5: Neighborhood illustration.</p>

support our theoretical claims on scalability and generalizability in large-scale networked MARL systems.

## H.2 Traffic control

**Environment setup.** The environment consists of a network of $n$ interconnected road links forming a directed graph. Each node $i$ represents a road link, and the edges define feasible turning movements between links. The local state of link $i$ is denoted by $\mathbf{s}_i(t) = (x_{ij}(t))_{j \in \mathcal{N}_i^{\text{out}}}$, where $x_{ij}(t) \in [S] = \{0, 1, \ldots, S\}$ represents the number of vehicles on link $i$ intending to turn to neighboring link $j$. Accordingly, the local state space is $\mathcal{S}_i = [S]^{|\mathcal{N}_i^{\text{out}}|}$.

The local action $a_i(t) = (y_{ij}(t))_{j \in \mathcal{N}_i^{\text{out}}}$ is a binary traffic-signal tuple, where $y_{ij}(t) \in \{0, 1\}$ controls whether the turn movement $(i \to j)$ is allowed at time $t$. Hence, the local action space is $\mathcal{A}_i = \{0, 1\}^{|\mathcal{N}_i^{\text{out}}|}$. When $y_{ij}(t) = 1$, a random number of queued vehicles $C_{ij}(t)$ will depart link $i$ and flow into link $j$, subject to capacity and queue constraints. Meanwhile, link $i$ receives incoming flows from other connected links $k \in \mathcal{N}_i^{\text{in}}$, where a random fraction $R_{ij}(t)$ of the inflow is assigned to each downstream queue $x_{ij}(t)$. The resulting dynamics are

$$x_{ij}(t+1) = [x_{ij}(t) - \min(C_{ij}(t)y_{ij}(t), x_{ij}(t))]_0^S + \sum_{k \in \mathcal{N}_i^{\text{in}}} \min(C_{ki}(t)y_{ki}(t), x_{ki}(t))R_{ij}(t), \quad (19)$$

where $[x]_0^S = \max(\min(x, S), 0)$. The random variables $C_{ij}(t)$ and $R_{ij}(t)$ are drawn i.i.d. from fixed distributions, defining stochastic yet locally dependent transitions. This structure ensures that $\mathbf{s}_i(t+1)$ depends only on the states and actions within link $i$'s neighborhood, consistent with the local interaction model in Section 2.1.

<p align="center">43</p>

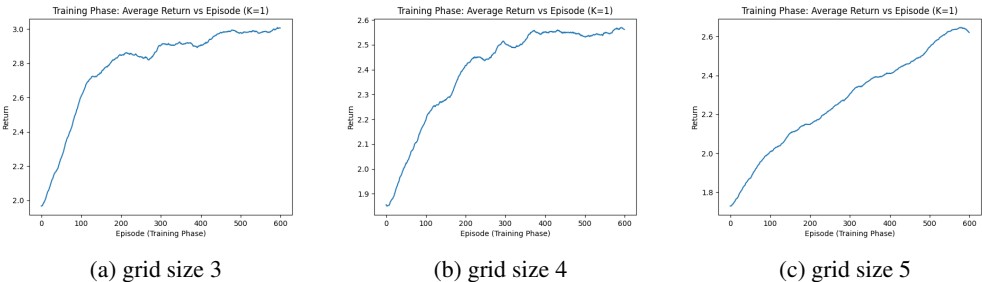

(a) grid size 3         (b) grid size 4         (c) grid size 5

Figure 6: GSAC Training for different grid sizes.

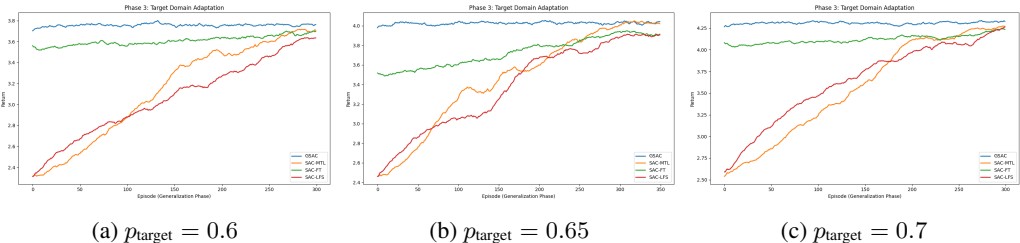

(a) $p_{\text{target}} = 0.6$      (b) $p_{\text{target}} = 0.65$      (c) $p_{\text{target}} = 0.7$

Figure 7: Adaptation comparison for different target domains in wireless communication benchmark.

The local reward is defined as the negative queue length, quantifying congestion at each link:

$$r_i(\mathbf{s}_i, a_i) = - \sum_{j \in \mathcal{N}_i^{\text{out}}} x_{ij}. \tag{20}$$

Intuitively, the objective is to minimize network-wide congestion by controlling local signals. Policies such as the max-pressure controller typically rely on known traffic-flow statistics, while our learning framework operates in a model-free fashion, adapting from observed data without prior knowledge of $C_{ij}$ or $R_{ij}$ distributions.

**Evaluation protocol.** We evaluate performance under $M = 3$ source domains, each defined by different vehicle flow intensities sampled from $\{\text{low}, \text{medium}, \text{high}\}$ levels of $C_{ij}(t)$. The target domain adopts an intermediate traffic level unless otherwise stated. Each experiment runs for $T = 50$ timesteps, with discount factor $\gamma = 0.95$ and learning rates $\alpha = 0.1$ for the critic and $\eta = 0.01$ for the actor. For domain factor estimation, $T_e = 20$ trajectories are collected per domain.

To assess adaptation, we compare GSAC against a SAC–LFS baseline trained directly in the target domain. We also vary the network topology by adjusting the number of intersections $n \in \{9, 16, 25\}$, corresponding to $3 \times 3$, $4 \times 4$, and $5 \times 5$ grid layouts. As is shown in Figure 8, GSAC achieves faster adaptation, lower queue lengths, and improved stability across all configurations.

**Domain shift sensitivity.** To analyze robustness, we test target domains with altered inflow distributions, varying $C_{ij}$ by $\pm 20\%$ relative to training domains. As is shown in Figure 9, GSAC consistently adapts to these changes with minimal degradation in performance, whereas LFS requires substantially more data to achieve comparable results. These findings confirm the scalability and generalizability of our model-free learning framework in stochastic, networked traffic environments.

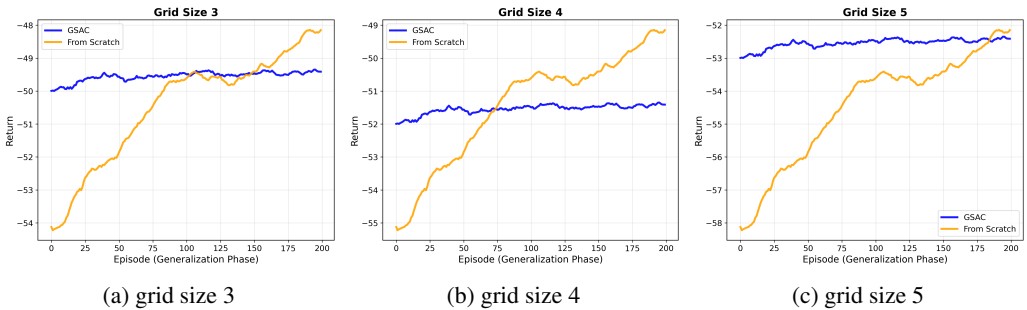

(a) grid size 3          (b) grid size 4          (c) grid size 5

Figure 8: Adaptation comparison for different grid sizes in traffic control benchmark.

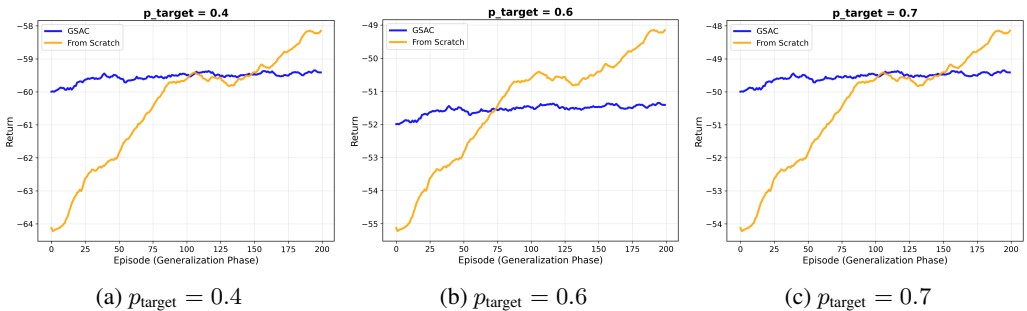

(a) $p_{\text{target}} = 0.4$       (b) $p_{\text{target}} = 0.6$       (c) $p_{\text{target}} = 0.7$

Figure 9: Adaptation comparison for different target domains in traffic control benchmark..

