# OpenReview forum: "Causality Meets Locality: Provably Generalizable and Scalable Policy Learning for Networked Systems"
_NeurIPS.cc/2025/Conference — NeurIPS 2025 spotlight_

### Official Review · Reviewer_y5CC · 2025-06-26

**Clarity:** 2
**Significance:** 2
**Originality:** 2
**Rating:** 4
**Confidence:** 4

**Summary:**

The paper studies the problem of designing a provably generalizable and scalable MARL algorithm for networked systems. Specifically, building on the exponential decay property proposed in previous works, they construct approximately compact representations of states and domain factors, which can further reduce the dimension of the truncated local Q-function. They also propose a meta actor-critic algorithm which enables learning scalable localized policies across multiple source domains. Theoretically, they provide guarantees on finite sample convergence of their algorithm as well as the adaptation gap. They also perform a numerical experiment on a wireless communication problem, where they demonstrated that their algorithm performed better than one than trained from scratch.

**Questions:**

1) Could the authors discuss an example of a realistic problem where the ACR reduces dimensionality significantly? For instance, is this the case in the wireless communication simulations? This can help strengthen this component of the contribution.

2) Could the authors discuss how the sample complexity of adapting to a new domain depends on the similarity of the new domain to earlier domains seen in training? The current relevant result (Thereom 4) only shows the sample complexity in expectation.

3) Could the authors comment on how their approach can be modified to handle the case with unknown $f_{i,j}$?

4) How might the algorithm compare against an alternative benchmark, where the controller is trained with a variety of domains, and then allowed to fine-tune on the new domain data, without explicitly conditioning the policy on the domain factor? This seems to me a more reasonable benchmark than training from scratch. Performing well versus this baseline will add more value to the contribution of the paper.

**Ethical Concerns:**

["NO or VERY MINOR ethics concerns only"]

**Final Justification:**

The authors have mostly addressed my concerns, and I will thus increase my score.

**Limitations:**

Yes.

**Paper Formatting Concerns:**

None.

**Quality:**

2

**Strengths And Weaknesses:**

**Strengths**

The idea of conditioning the policy and value function on domain factors is interesting, and as the empirical results suggest, this seems to facilitate quicker adaption when a new domain is seen at test-time. The authors also provide rigorous theoretical analysis for the finite-time convergence of their algorithm.

**Weaknesses*

1) I am not sure how much value the approximate compact representation (ACR) can bring. From what I read, the authors did not give any concrete examples of realistic problems where this reduces dimensionality.

2) I believe a crucial assumption required for estimation of the domain factor is that the dynamics function $f_{i,j}$ is known, so that $\nabla_{\omega} f_{i,j}$ can be computed. This limits the approach to only be valid for cases where the dynamics function is known, which I believe the authors did not explicitly mention in the paper.

3) The algorithm only applies to finite state and action spaces, while some recent MARL network algorithms can in fact achieve provable and efficient performance for continuous state and action space network MARL problems, which arises often in practice. Since an important aspect of the algorithm is its potential for empirical improvement (the theoretical aspects of the algorithm are not particularly novel compared to previous network MARL algorithms), this is an important drawback.

4) The empirical validation is rather limited, since the authors only consider one benchmark, and the latent domain factor in this problem is quite simple (since it is one-dimensional).

5) There are some typos which the authors should be aware of. For instance, in Line 126, I believe the summation should be replaced by a product (the input dimensionality is a product of the local input dimensions).

---

> ### Author Rebuttal · Authors · 2025-07-31
>
> Thank you for your thoughtful and constructive feedback. We appreciate your recognition of **theoretical rigor**. We would like to address your concerns/comments as follows.
> ### W1/Q1: "how much value the ACR can bring" "example of a realistic problem where this reduces dimensionality"
> In our wireless benchmark, the local state of each user $i$ is represented as $s_i(t) = (b_{i,1}(t), \ldots, b_{i,d_i}(t), z_{i,1}(t), z_{i,2}(t))$. $b_i(t) \in \\{0,1\\}^{d_i}$ encodes queue status by deadline bins (e.g., $b_i(t)=(1,1,0)$ means the first two deadline bins are occupied while the third is empty). $z_{i,1}(t)$ and $z_{i,2}(t)$ are **irrelevant components** to decision quality:
>
> - $z_{i,1}(t)$: channel quality indicator, which does not influence the packet success probability $q_i$, as $q_i$ depends solely on collisions.
> - $z_{i,2}(t)$: grid coordinates, which adds no useful information because the connectivity graph already encodes all relevant interactions.
>
> The ACR procedure **automatically prunes** such irrelevant or redundant variables, retaining only those that causally influence the local reward or dynamics. This yields a **non-trivial reduction** in local dimensionality. For example, on a $4\times4$ grid with $d_i=2$ and a neighborhood size $|N_i|=5$:
>
> - **Raw (truncation-only):** $|N_i|\times(d_i+2)=5\times4=20$ features.
> - **With ACR:** $|N_i|\times d_i=5\times2=10$ features.
>
> This **50% shrinkage** directly translates into:
>
> - **Smaller critics and lower compute:** reduced local state–action support and table size.
> - **Faster learning:** lower variance in actor updates and tighter constants in our bounds.
> - **Accuracy preserved:** approximation error remains controlled (see Prop. 1–2).
> - **Improved sample efficiency:** by effectively increasing the minimum visitation $\sigma$ and reducing the mixing constant $\tau$ (Assumption 2), ACRs **tighten the constants** in the convergence guarantee (Thm 2).
>
> More generally, ACR is valuable when:
> - State representations include auxiliary information, e.g., sensor readings not directly affecting rewards
> - Network structure creates redundancies, e.g., traffic flow where only upstream neighbors matter
> - Domain factors have sparse influence, e.g., weather affecting only outdoor equipment in power grids
>
> We have included this concrete wireless example, and highlighted which variables are pruned and by how much, along with mask sparsity and critic-size statistics, in the revised paper.
> ### W2/Q3: "crucial assumption required for domain factor estimation" "how to modifify to handle the case with unknown $f_{i,j}$"
> We would like to clarify that our approach does **not** assume explicit knowledge of the closed-form dynamics $f_{i,j}$. We estimate domain factors using observed trajectories and causal masks (see Section 5, Lines 226–233). The structural matrices $c_{\cdot \to \cdot}$ are identifiable under standard causal discovery assumptions (Thm 1). As shown in Appendix F (Prop. 4 and 5),  the domain factor $\omega$ are estimated  from observed transitions using a maximum likelihood estimation. This likelihood can be evaluated using a learned parametric approximation $\hat{f}$ rather than the true $f$.  Thus, our analysis and results remain valid for **unknown dynamics**, provided that $\hat{f}$ is a consistent estimator of the transition function. Furthermore, Prop. 5 in Appendix F ensures that as long as $\hat{f}$ approximates $f$ within $\epsilon$, the error in domain factor estimation increases by at most $O(\epsilon)$ due to the Lipschitz continuity of the transition likelihood. To avoid confusion, we have clarified these details in the revised version.
>
> ### W3: only applies to finite state and action spaces
> **Scope**. Thank you for raising this point. Our contribution targets the gap between **provably scalable, localized MARL on network graphs** (where most analyses are tabular/finite) and cross-domain generalization/adaptation. To the best of our knowledge, we provide the *first unified algorithm and theoretical analysis* that couples causal locality (ACRs) with domain-factor conditioning and delivers finite-sample convergence and adaptation guarantees for networked systems. We instantiate the theory in finite spaces to keep assumptions transparent and to isolate the impact of ACRs and meta-conditioning without function-approximation confounders.
>
> **Extensions**. The ACR and GSAC framework extend naturally to **continuous** spaces by (i) replacing tabular modules with function approximators, and (ii) using entropy-regularized localized actor–critic updates. Because ACRs prune inputs rather than restrict the function class, the variance/conditioning benefits remain, and the exponential-decay/causal-mask assumptions carry over under mild Lipschitz/mixing conditions. In this paper we focus on the finite-space instantiation and position the continuous extension as **complementary future work**. In the revision we have added a concise discussion of assumptions, connections to continuous MARL, and implementation pointers, and we explicitly cite representative continuous MARL works to position our approach as complementary.
>
> ### W4: only one benchmark, the latent domain factor is quite simple
> The wireless environment is widely used as a stress test for decentralized MARL algorithms **[1]**. Despite its simple domain factor (packet arrival probability $p_i$), the interaction graph and collision dynamics induce highly nontrivial coordination challenges, especially as grid size grows (Fig 2). That said, we agree that broader empirical validation would strengthen the work.
> - **Add benchmark**: we extend experiments to **traffic control network [1,2]** on a road-link graph, where  state per link is queue counts, action controlling the on-oﬀ of turn movement, and reward is negative total queue length. The preliminary results already show consistent advatanges (much faster adaptation compared to $\texttt{LFS}$).
>
> | Adap. Ep.| GSAC Return | LFS Return | GSAC Queue | LFS Queue |  Return Diff. |  Queue Diff. |
> | :--- | :--- | :--- | :--- | :--- | :--- | :--- |
> |1 |-54.9942 |-57.9916 |4.7782 |5.1239 |2.9973 |0.3457|
> |50 |-54.5165 |-57.4359 |4.7710 |5.0569 |2.9194 |0.2859|
> |150 |-54.2193 |-56.0777 |4.7234 |4.9296 |1.8584 |0.2061|
> |300 |-53.2257 |-55.0790 |4.6568 |4.7744 |1.8534 |0.1176|
> |400 |-53.1087 |-53.8259 |4.6359 |4.6433 |0.7172 |0.0073|
>
> - **Add baselines**: we implement **two additional baselines**, (i) a multi-task variant of GSAC, (ii) a domain-factor-free baseline. We put the detailed experimental results and discussions to the response to **Q4**.
> ### W5: typos
> Thank you for pointing this out. We have corrected this typo and carefully proofread the paper.
> ### Q2: how the sample complexity of adapting to a new domain depends on the similarity of the new domain to earlier domains seen in training?
> Theorem 4 bounds the adaptation gap as $O(1/\sqrt{T_a})$. The key idea of proving Theorem 4 is the inequality
> $$J(\theta(K),\hat{\omega}^{M+1};\omega^{M+1})-J(\theta(K),\omega^{M+1};\omega^{M+1}) \ge -L_{\omega'} || \hat{\omega}^{M+1}-\omega^{M+1} ||.$$
> Thus, when target domains are close to training domains, the total variation distance $|| \hat{\omega}^{M+1}-\omega^{M+1} ||$ is smaller. This leads to smaller adaptation gap, resulting in faster adaptation. We have added this discussion in the revised version.
> ### Q4: compare against the controller trained with a variety of domains, and then allowed to fine-tune on the new domain data, without explicitly conditioning the policy on the domain factor
> Thank you for this excellent suggestion. Following your and **ReviewerDb8N**' suggestions, we add two new baseliness:
> - $\texttt{SAC-MTL}$ (multi-task): learns  $\pi_{\theta}(a|s,z)$, where $z$ indicates the **onehot encoding** of the task; optimized jointly over source domains **[5]**.
> - $\texttt{SAC-FT}$ (fine-tune, suggested by **Reviewer y5CC**):  trains a single policy $\pi_{\theta}(a|s)$ across sources **without domain-factor conditioning** and **fine-tunes $\theta$**.
> - $\texttt{GSAC}$ (ours): learns $\pi_{\theta}(a|s,\omega)$, and adapts by adjusting $\omega$ (estimated from a few trajectories), not $\theta$
>
> Return evaluation on a 4×4 network (source domain $p_i \in \\{0.2, 0.5, 0.8\\}$, $p_{target}=0.65$, higher is better). “Ep” denotes adaptation episodes collected in the target domain.
> | Ep | $\texttt{GSAC}$ | $\texttt{SAC-MTL}$ | $\texttt{SAC-FT}$ | $\texttt{LFS}$ |
> |:-:|-----:|-----:|-----:|-----:|
> |1|**3.5093**|2.6802|3.3715|2.4125|
> |10|3.5314|2.6870|3.3283|2.4249|
> |30|3.5537|2.7723|3.3681|2.6146|
> |50 |3.5573|2.8693|3.3337|2.7913|
> |100 |3.5514|3.2814|3.3677|3.0481|
> |**130** |3.5350|**3.5466**|3.4910|3.1683|
> |150 |3.5463|3.6869|3.4841|3.1947|
> |200 |3.5552|3.7961|3.5201|3.3476|
> |**250** |3.5517|3.7882|**3.5602**|3.4625|
> |**300**|3.5465|3.8011|3.6294|**3.5623**|
>
> Initially, performance ranks $\texttt{GSAC}$>$\texttt{SAC-FT}$>$\texttt{SAC-MTL}$>$\texttt{LFS}$, gaps that persist through Ep 50, highlighting sizable initial gains over $\texttt{LFS}$ (+45% for $\texttt{GSAC}$, +40% for $\texttt{SAC-FT}$, +11% for $\texttt{SAC-MTL}$). $\texttt{GSAC}$ achieves the best few-shot performance (Ep 1–50), reflecting rapid adaptation from minimal target data. $\texttt{SAC-MTL}$ overtakes after ~130 episodes and remains best thereafter. $\texttt{SAC-FT}$ outperforms $\texttt{SAC-MTL}$ initially but lags it later, and surpasses $\texttt{GSAC}$ after ~250 episodes. $\texttt{LFS}$ improves the slowest and only exceeds $\texttt{GSAC}$ after ~300 episodes. This matches our design goal: optimize early-episode adaptation by adjusting $\omega$ rather than re-training $\theta$.
>
> ***
> **References**
>
> [1] Guannan Qu, et al. Scalable reinforcement learning for multi-agent networked systems. Operations Research, 2022.
>
> [2] Varaiya, P. Max pressure control of a network of signalized intersections. Transportation Research Part C: Emerging Technologies, 36, 177-195, 2013.

---

> > ### Comment · Reviewer_y5CC · 2025-08-03
> > **Response to authors**
> >
> > I thank the authors for the responses. They have mostly addressed my concerns, and I will increase my score.

---

> > > ### Author Response · Authors · 2025-08-03
> > >
> > > Thank you again for the constructive feedback and for increasing the score.

---

### Official Review · Reviewer_Db8N · 2025-07-06

**Clarity:** 2
**Significance:** 2
**Originality:** 2
**Rating:** 5
**Confidence:** 2

**Summary:**

This paper introduces GSAC (Generalizable and Scalable Actor-Critic), a multi-agent reinforcement learning framework tailored to large-scale networked systems (e.g., traffic, power, and wireless grids). GSAC leverages causal representation learning to identify sparse local causal masks and construct Approximately Compact Representations (ACRs) of both state and latent domain factors.  A meta actor-critic algorithm then trains a shared, domain-conditioned policy across multiple source environments; at test time, only a few trajectories in a new domain are needed to estimate its factors and adapt the policy.

**Questions:**

- For experiments, how many layers/parameters is the policy?
- Why not try this on robotic simulation, such as Metaworld?
- Why meta learning as opposed to multitask learning? Are there ablations regarding this?

**Ethical Concerns:**

["NO or VERY MINOR ethics concerns only"]

**Final Justification:**

Better experiments and good explanation on ACR's low cost.

**Quality:**

2

**Strengths And Weaknesses:**

Strengths:
- Theoretical Rigor: Establishes structural identifiability of local causal masks (seen in Theorem 1), finite-sample bounds for mask recovery (Prop. 4), and convergence and adaptation guarantees for the meta actor-critic (Theorem 2–4).
- Clear and structured Algorithm - Presents a four-phase pipeline (causal recovery, ACR construction, meta actor-critic training, rapid adaptation) with detailed pseudocode (Alg. 1).

Weaknesses:
- Computational Overhead: The causal discovery and ACR construction phases (Phase 1 and Phase 2 in Section 4.1) may incur significant upfront cost, and the required neighborhood masking and meta-training could be prohibitive at very large scales.
- Minimal Experiment - This paper seems mainly theoretical and only has one experiment (and no baselines)
- Strong Assumptions: Relies on faithfulness, sub-Gaussian noise, bounded degree, full observability, and Lipschitz continuity (Propositoin 4)—conditions that may not hold in many real systems.

---

> ### Author Rebuttal · Authors · 2025-07-31
>
> Thank you for your thoughtful and constructive feedback. We appreciate your recognition of **theoretical rigor** and **clear and structured** algorithm. We would like to address your concerns/comments as follows.
> ### W1: computational overhead of causal discovery and ACR construction
> We acknowledge that Phase 1 (causal discovery) and Phase 2 (ACR construction) introduce upfront costs. However, these phases only incur **one-time, local, and amortized cost**.
> - They are one-time preprocessing steps for each source domain and do not need to be repeated during meta-training or adaptation.
> - Both steps are local (per agent and over small neighborhoods),  parallel across agents, and the results are re-used for the entire meta-training horizon and for adaptation. We make this explicit in Algorithm 1 (Phases 1–2 precede meta-training and adaptation).
>
> In addition, **ACRs reduce overall cost**. As shown in Section 3 and Theorem 2–4, ACRs reduce the effective input dimensionality from $O(|s_{N^k_i}|)$ to $O(|s_{N^{\kappa}_i}|)$, resulting in quasi-linear scaling in the number of agents. This leads to smaller mixing times $\tau$ and higher minimum visitation probabilities $\sigma$, improving both convergence speed and sample efficiency (see Section 6.3).
> Overall, while causal discovery and ACR construction are non-trivial, they are (i) local/parallel, (ii) one-time per source domain, and (iii) reduce the total training/adaptation cost via smaller representations and better constants in the convergence bounds.
> ### W2: seems mainly theoretical and only has one experiment (and no baselines)
> Our primary contribution was to establish the theoretical foundations of GSAC, including the structural identifiability, convergence bounds, and adaptation guarantees. However, we admit that broader empirical coverage will strengthen the paper. Here are clarifications:
> - **What is already included**: Appendix H details multiple grid sizes (3×3, 4×4, 5×5), target-domain sweeps for the target packet-arrival rate, and a Learning-From-Scratch ($\texttt{LFS}$) baseline trained directly in the target domain. Across settings, GSAC adapts in far fewer steps and outperforms $\texttt{LFS}$ (see Fig. 1b, 4–6)
> - **Add benchmark**: we extend experiments to **traffic control network [1,2]** on a road-link graph, where  state per link is queue counts, action controlling the on-oﬀ of turn movement, and reward is negative total queue length. The preliminary results already show consistent advatanges (much faster adaptation compared to $\texttt{LFS}$).
>
> | Adaptation Episode | $\texttt{GSAC}$ Return | $\texttt{LFS}$ Return | $\texttt{GSAC}$ Queue | $\texttt{LFS}$ Queue |  Return Difference |  Queue Difference |
> | :--- | :--- | :--- | :--- | :--- | :--- | :--- |
> | 1 | -54.9942 | -57.9916 | 4.7782 | 5.1239 | 2.9973 | 0.3457 |
> | 50 | -54.5165 | -57.4359 | 4.7710 | 5.0569 | 2.9194 | 0.2859 |
> | 150 | -54.2193 | -56.0777 | 4.7234 | 4.9296 |1.8584 | 0.2061 |
> | 300 | -53.2257 | -55.0790 | 4.6568 | 4.7744 | 1.8534 | 0.1176 |
> | 400 | -53.1087 | -53.8259 | 4.6359 | 4.6433 | 0.7172 | 0.0073 |
>
> - **Add baselines**: we implement **two additional baselines**, (i) a multi-task variant of GSAC, (ii) a domain-factor-free baseline. We put the experimental results and discussions to the response to **Q3**.
> ### W3: "Relies on faithfulness, sub-Gaussian noise, bounded degree, full observability, and Lipschitz continuity (Proposition 4)—conditions that may not hold in many real systems"
> Our assumptions in Proposition 4 are standard in causal discovery and theoretical MARL.
> - **Faithfulness and sub-Gaussian noise**: these are classical and standard requirements to ensure identifiability **[3,4]** and finite-sample recovery.
> - **Bounded degree**: bounded neighborhood degree is natural in networked systems where agents have limited communication range. For example, this holds when each intersection connects to only a few roads, or each wireless node has limited interference range.
> - **Full observability**: while we currently assume full local observability, the ACR construction can be extended to partially observable settings using learned belief states or latent encoders. This is part of our future work.
> - **Lipschitz continuity**: this mild smoothness assumption is satisfied by most smooth policies.
>
> Importantly, our algorithm can still be applied when assumptions are violated, only the theoretical guarantees may not hold exactly. We have added a “Limitations and assumptions” paragraph to discuss where these conditions may fail and how GSAC could be extended.
> ### Q1: layers/parameters of the policy
> In the wireless benchmark the local action space is discrete, and we **do not use a neural network** for the policy. We use a softmax (Boltzmann) policy over local actions conditioned on ACR inputs:
>  $$  \pi(a|s,\omega) = \text{softmax}( \theta[a]^\top (s,\omega)/\tau ),  $$
> where $\theta\in\mathbb{R}^{|A|\times d}$ is the parameter matrix, $(s,\omega)\in\mathbb{R}^{d}$ are the ACR features, and $\tau=0.5$ is the softmax temperature.
> ### Q2: Why not try this on robotic simulation, such as Metaworld?
> Our focus is on is large-scale networked systems (traffic, power, wireless), where agents interact over a graph and where $\kappa$-hop truncation and local causal masks are principled. Metaworld **[5]** tasks lack the explicit networked graph structure that GSAC is designed to exploit. It is primarily *single-agent*, and running multiple independent MetaWorld tasks in parallel does not induce the structured inter-agent dynamics our theory leverages. Our theorems and ACR constructions are explicitly graph- and neighborhood-based. That said, applying GSAC to robotic settings with explicit interaction graphs (e.g., multi-robot navigation with communication/collision constraints or multi-arm coordination with shared resources) is a natural and valuable next step. We have added discussion of this direction in the revised paper.
> ### Q3: Why meta learning as opposed to multitask learning? Are there ablations regarding this?
> **Clarification of meta-RL vs. multi-task RL paradym**: the goal of multi-task RL **[5]** is to learn a **single, task conditioned** policy that maximize the average expected return across all source domains from a domain distribution. However, it offers no guarantee of rapid few-shot adaptation from new, unseen domain. Meta-RL, in contrast, explicitly learns to adapt: it leverage the set of source domains to learn a policy that, at test time,  can quickly adapt to a new domain with few trajectories.
>
> **Why meta-learning**: our goal is is **fast adaptation in a new domain from few trajectories**. We train a shared, **domain-factor-conditioned** policy across sources and, at test time, estimate domain factors $\hat\omega$ from a few trajectories to adapt immediately. Our theory shows:
> - Meta-gradient error terms shrink with the **number of source domains $M$** and the **samples used to estimate domain factors $T_e$**.
> - The **adaptation gap decays as $O(1/\sqrt{T_a})$** with the number of adaptation trajectories.
>
> **New baselines**:
> - $\texttt{SAC-MTL}$ (multi-task): learns  $\pi_{\theta}(a|s,z)$, where $z$ indicates the **onehot encoding** of the task; optimized jointly over source domains **[5]**.
> - $\texttt{SAC-FT}$ (fine-tune, suggested by **Reviewer y5CC**):  trains a single policy $\pi_{\theta}(a|s)$ across sources **without domain-factor conditioning** and **fine-tunes $\theta$**.
> - $\texttt{GSAC}$ (ours): learns $\pi_{\theta}(a|s,\omega)$, and adapts by adjusting $\omega$ (estimated from a few trajectories), not $\theta$
>
> Return evaluation on a 4×4 network (source domain $p_i \in \\{0.2, 0.5, 0.8\\}$, $p_{target}=0.65$, higher is better). “Ep” denotes adaptation episodes collected in the target domain.
> | Ep | $\texttt{GSAC}$ | $\texttt{SAC-MTL}$ | $\texttt{SAC-FT}$ | $\texttt{LFS}$ |
> |:-:|-----:|-----:|-----:|-----:|
> |1|**3.5093**|2.6802|3.3715|2.4125|
> |10|3.5314|2.6870|3.3283|2.4249|
> |30|3.5537|2.7723|3.3681|2.6146|
> |50 |3.5573|2.8693|3.3337|2.7913|
> |100 |3.5514|3.2814|3.3677|3.0481|
> |**130** |3.5350|**3.5466**|3.4910|3.1683|
> |150 |3.5463|3.6869|3.4841|3.1947|
> |200 |3.5552|3.7961|3.5201|3.3476|
> |**250** |3.5517|3.7882|**3.5602**|3.4625|
> |**300**|3.5465|3.8011|3.6294|**3.5623**|
>
> Initially, performance ranks $\texttt{GSAC}$>$\texttt{SAC-FT}$>$\texttt{SAC-MTL}$>$\texttt{LFS}$, gaps that persist through Ep 50, highlighting sizable initial gains over $\texttt{LFS}$ (+45% for $\texttt{GSAC}$, +40% for $\texttt{SAC-FT}$, +11% for $\texttt{SAC-MTL}$). $\texttt{GSAC}$ achieves the best few-shot performance (Ep 1–50), reflecting rapid adaptation from minimal target data. $\texttt{SAC-MTL}$ overtakes after ~130 episodes and remains best thereafter. $\texttt{SAC-FT}$ outperforms $\texttt{SAC-MTL}$ initially but lags it later, and surpasses $\texttt{GSAC}$ after ~250 episodes. $\texttt{LFS}$ improves the slowest and only exceeds $\texttt{GSAC}$ after ~300 episodes. This matches our design goal: optimize early-episode adaptation by adjusting $\omega$ rather than re-training $\theta$.
>
> ***
> **References**
>
> [1] Varaiya, P. Max pressure control of a network of signalized intersections. Transportation Research Part C: Emerging Technologies, 36, 177-195, 2013.
>
> [2] Guannan Qu, et al. Scalable reinforcement learning for multi-agent networked systems. Operations Research, 2022.
>
> [3] J. Pearl. Causality: Models, Reasoning, and Inference. Cambridge University Press, Cambridge, 2000.
>
> [4] Huang, Biwei, et al. Adarl: What, where, and how to adapt in transfer reinforcement learning. arXiv preprint arXiv:2107.02729 (2021).
>
> [5] Tianhe Yu, et al. Meta-world: A benchmark and evaluation for multi-task and meta reinforcement learning. Proceedings of the Conference on Robot Learning, PMLR 100:1094-1100, 2020.

---

> > ### Comment · Reviewer_Db8N · 2025-08-07
> >
> > Thanks for the detailed response, I have raised my score to 5 due to better experiments.

---

> > > ### Author Response · Authors · 2025-08-07
> > >
> > > Thank you for your reconsideration and updated score. We appreciate your engagement.

---

### Official Review · Reviewer_7kCG · 2025-07-09

**Clarity:** 3
**Significance:** 4
**Originality:** 4
**Rating:** 6
**Confidence:** 4

**Summary:**

The paper proposes a framework which leverages causal representation learning within the meta actor-critic learning. The authors introduce approximately-compact representations, which truncate value functions to $\kappa$-hop neighbourhood in order to improve learning efficiency.
The authors address the following theoretical and empirical aspects:
- Structural identifiability in networked MARL: theoretical proofs for the sample complexity
- Introducing approximately-compact representations (ACRs) and proposing algorithms
- Meta-actor-critic (MAC) algorithm which is tasked with learning policies across source domains
- Theoretical guarantees for the algorithm are provided

**Questions:**

The citation of the empirical task includes where the authors have got the dataset from (paper [2]), but not the original work [1]

[1]Alessandro Zocca. Temporal starvation in multi-channel csma networks: an analytical framework. Queueing Systems, 91(3-4):241–263, 2019.
[2] Guannan Qu, Yiheng Lin, Adam Wierman, and Na Li. Scalable multi-agent reinforcement learning for networked systems with average reward. Advances in Neural Information Processing Systems, 33:2074–2086, 2020.

**Ethical Concerns:**

["NO or VERY MINOR ethics concerns only"]

**Final Justification:**

After reading the reviews, I believe that the authors have done a really good job addressing the reviews. I think the paper has merits which underpin my vote for acceptance:
- Soundness: while the paper leans towards the theoretical analysis of the proposed algorithm, the approach is sound and backed up with both theoretical and empirical analysis
- Correctness: I have checked the derivations, and they appear correct
- Originality: scalability of MARL to large-scale network systems is an important open problem, which is of scope and interest to the NeurIPS community, using the causal structures for adaptation to the domain shifts is a promising and original idea

During the rebuttal, the authors have addressed a number of concerns, including:
-  clarity: the authors address how they will include the roadmaps, improved description of computational overhead, clarification on theoretical assumptions
- deficiency of empirical analysis: addition of new experiments for the traffic control network task, which confirm the presented theoretical insights, a number of baselines are added

Therefore, I suggest acceptance and that the authors update the paper according to the rebuttal discussion, which in my opinion would further strenghen the paper.

**Limitations:**

The authors mention that 'The major limitation is that GSAC is currently evaluated on a single fully-observed benchmark'. The authors are also suggested to tell about the limitation of the assumptions of the theoretical guarantees.

In other words, in which ways could the assumptions 1-6, line 256, not to be met in practice, and what would be the implications for the proposed method?

**Paper Formatting Concerns:**

No major issues.

Line 220: weights them-> *weighs* them

**Quality:**

4

**Strengths And Weaknesses:**

Strength:
- Soundness: while the paper leans towards the theoretical analysis of the proposed algorithm, the approach is sound and backed up with both theoretical and empirical analysis
- Correctness: I have checked the derivations, and they appear correct
- Clarity: the writing is mostly clear, with some suggestions in the disadvantages section
- Originality: scalabiltity of MARL to large-scale network systems is an important open problem, which is of scope and interest to the NeurIPS community, using the causal structures for adaptation to the domain shifts is a promising and original idea

Weaknesses:
- Clarity: While it is good in general, I would think the paper would be improved by adding some sort of roadmap in the beginning of the methods section, showing how all these pieces, both theory and the algorithm, fit together into a single story. In Section 4.1, the authors state the parts of the Algorithm, but it would be also good if they linked these statements with the theorems.

---

> ### Author Rebuttal · Authors · 2025-07-31
>
> Thank you for your thoughtful and constructive feedback. We appreciate your recognition of **soundness**, **originality** and **significance** of our work. We would like to address your concerns/comments as follows.
> ### W1:  "add some sort of roadmap in the beginning of the methods section, showing how all these pieces, both theory and the algorithm, fit together into a single story" "In Section 4.1, the authors state the parts of the Algorithm, but it would be also good if they linked these statements with the theorems"
> We appreciate the suggestion of adding a roadmap theorem-algorithm linkage in Section 4. In the revised version,
> - we introduce a **high-level roadmap** at the start of Section 4, outlining how causal discovery, ACR construction, and meta actor-critic optimization fit together into a unified story (Phases 1–4, Algorithm 1).
> - we explicitly link the algorithmic components with the corresponding theoretical results (Theorems 1–4 and Propositions 1–5).
> - we also add a **brief summary figure** (Figure 1) to visually illustrate the GSAC pipeline, complementing the roadmap.
>
> We add **Roadmap and Theoretical Connections** at the start of Section 4 (right before Section 4.1, Algorithm overview). It combines the roadmap with explicit theorem-algorithm linkages:
> > Our GSAC framework (Algorithm 1) integrates causal discovery, representation learning, and meta actor-critic optimization into a unified pipeline, with each phase supported by theoretical results. **Phase 1** (causal discovery and domain factor estimation) is underpinned by **Theorem 1** and **Propositions 4–5**, which establish structural identifiability and sample complexity guarantees for recovering causal masks and latent domain factors. **Phase 2** (construction of ACRs) leverages the causal structure to build compact representations of value functions and policies, with bounded approximation errors rigorously characterized by **Propositions 1–3**. **Phase 3** (meta actor-critic learning) performs scalable policy optimization across multiple source domains, with convergence of the critic and actor updates guaranteed by **Theorem 2** (critic error bound) and **Theorem 3** (policy gradient convergence). Finally, **Phase 4** (fast adaptation to new domains) exploits the learned meta-policy and compact domain factors to achieve rapid adaptation, where the adaptation performance gap is formally controlled by **Theorem 4**. Together, these results demonstrate that each algorithmic component is theoretically justified and collectively leads to provable scalability and generalization in networked MARL.
>
> We  also add a one-paragraph explanation referring to Figure 1:
> > Figure 1 complements Algorithm 1 by showing how causal recovery and ACR construction (Phases 1–2) compress the state-action space, enabling efficient meta-policy training (Phase 3), and how this meta-policy rapidly adapts to new domains (Phase 4), as guaranteed by Theorems 2–4.
> ### Q1: citation of the empirical task
> Thank you for pointing it out. We have updated our references to cite the original work **[1]** on the multi-channel CSMA network model in Section 7 Numerical experiments, alongside the benchmark paper **[2]**. We also added references to cite this paper in Introduction on wireless communication systems.
> ### Limitations: "in which ways could the assumptions 1-6 not to be met in practice, and what would be the implications for the proposed method"
> You raised an important point regarding Assumptions 1–6 (line 256). **A1-A4** are standard assumptions for proving convergence of networked MARL algorithms, without consideration of domain generalization/adaptation  **[2-4]**. For example, **[4]** provides the first provably efficient MARL framework for networked systems under the discounted reward setting, and including the first convergence analysis of networked MARL algorithm under **A1-A4**.
>
> Our work further fills the gap between scalability and *generalizability across domains* in networked systems. To this aim, we introduce additional asumptions **A5-A6** regarding the latent domain factor. **A5** is similar with **A3**, which imposes the smoothness w.r.t. the domain factor $\omega$, while **A3** imposes the smoothness w.r.t. the actor parameter $\theta$. **A6** is a regularity assmption to ensure the compactness of domain factor space.
>
> While **A1-A6** are standard for establishing theoretical guarantees, we discuss whether they hold exactly in practice as follows:
> - **Finite MDP  (A1)**: in some systems, the state or action spaces might not finite or continuous. For such cases, our results can still extend by applying state aggregation or clipping methods to ensure boundedness while incurring small approximation error.
> - **Mixing properties (A2)**: real-world networked systems may exhibit slow mixing or non-Lipschitz dynamics. Our theoretical results would degrade gracefully, leading to larger constants $\tau$ in the sample complexity bounds.
> - **Smoothness (A3)**: the assumption requires the policy gradients $\nabla_{\theta_i} \log \pi_i$ to be Lipschitz and uniformly bounded. This holds for all commonly used policy parameterizations (e.g., softmax policies) with bounded inputs and finite-dimensional parameter vectors. It ensures stable policy updates and is common in the convergence analysis of policy gradient and actor-critic algorithms.  $\texttt{GSAC}$ uses softmax parameterizations for policies, where $\log \pi_{\theta_i}$ is smooth and has bounded gradients when action probabilities are not close to zero (a condition satisfied by entropy regularization or initialization).
> - **Compact parameter space (A4)**: policy parameters $\theta_i$ are always constrained within a finite range due to neural network weight regularization, weight clipping, or bounded initialization. This is standard in actor-critic methods, where optimization is performed with bounded learning rates and regularizers that implicitly keep the parameters within a compact set.
> - **Smoothness w.r.t. domain factor (A5)**: this is reasonable because domain factors, e.g., channel gains, traffic loads, represent physical parameters that vary smoothly. The local dynamics $P_i$ are typically analytic or polynomial in $\omega_i$, so small perturbations in $\omega_i$ induce proportionally small changes in state transitions and rewards. Consequently, the Q-function $Q^\pi_i(s,a,\omega)$ and policy gradient $\nabla_{\theta_i} J(\theta,\omega)$ inherit this smoothness. Moreover, standard policy parameterizations  and neural network approximators are Lipschitz when combined with bounded weights or gradient clipping. Thus, **A5** aligns well with both the structure of real-world systems and common RL practices. Moreover, even if the true dynamics were not strictly Lipschitz, domain factor estimation (Proposition 5) uses a small-perturbation assumption around $\omega$ for adaptation. In practice, state or reward clipping and function approximation, which are inherently Lipschitz, effectively enforce this assumption.
> - **Compact domain factor space (A6)**: the latent domain factors $\omega_i$ represent bounded variations in environment dynamics (e.g., channel gains, traffic intensities, or reward scaling factors), which are inherently limited by the physical system. In real-world networked systems, these factors are naturally confined to finite ranges determined by system design (e.g., transmission powers, queue lengths). Hence, assuming $\Omega_i$ is compact with bounded diameter $D_\Omega$ is both reasonable and practical.
>
> In the revised version, we have added discusssions about these potential limitations and their implications in Section 8 and Appendix G, making clear how they might affect the performance of $\texttt{GSAC}$ in less structured or highly stochastic environments.
> ### Additional minor comments
> We have corrected the typo at Line 220 (“weights them” → “weighs them”).
> ***
> **References**
>
> [1] Alessandro Zocca. Temporal starvation in multi-channel csma networks: an analytical framework. Queueing Systems, 91(3-4):241–263, 2019.
>
> [2] Guannan Qu, Yiheng Lin, Adam Wierman, and Na Li. Scalable multi-agent reinforcement learning for networked systems with average reward. Advances in Neural Information Processing systems, 33:2074–2086, 2020.
>
> [3] Yiheng Lin, Guannan Qu, Longbo Huang, and Adam Wierman. Multi-agent reinforcement learning in stochastic networked systems. Advances in neural information processing systems,34:7825–7837, 2021.
>
> [4] Guannan Qu, Adam Wierman, and Na Li. Scalable reinforcement learning for multi-agent networked systems. Operations Research, 2022.

---

> > ### Comment · Reviewer_7kCG · 2025-08-03
> >
> > Many thanks for the response which clarifies upon my concerns. I am currently reading the other reviewers' comments and will come back on them as well. Meanwhile, the recommendation stays the same.

---

> > > ### Author Response · Authors · 2025-08-03
> > >
> > > Thank you for the follow-up. We're happy to provide any further clarification.

---

> > > > ### Comment · Reviewer_7kCG · 2025-08-05
> > > >
> > > > I have checked the other reviews, and my conclusion remains the same. I believe that the new experimental results, as well as addition of the baselines, strengthen the contribution of the paper.  The concerns about the motivation, voiced by Reviewer y5CC, also look to me addressed.

---

> > > > > ### Author Response · Authors · 2025-08-05
> > > > >
> > > > > Thank you for the update and for recognizing the added experiments and baselines.

---

### Note · Authors · 2025-08-12

We thank the reviewers and AC for their constructive feedback and engagement. We are encouraged that all three reviewers recognized the soundness, originality, and significance of our work, and that our rebuttal and added experiments led to score increases from Reviewers Db8N and y5CC.

**Clarity and roadmap**: Added a high-level roadmap in Section 4 linking GSAC phases to specific theorems/propositions, plus Figure 1 to visualize the pipeline.

**Empirical coverage**: Expanded beyond the wireless benchmark to a traffic-control domain and added two baselines (multi-task variant, fine-tuning without domain factors). GSAC achieves the best few-shot performance and consistently faster adaptation across domains.

**ACR value**:  Demonstrated in the wireless benchmark a ~50% reduction in local state dimensionality by pruning irrelevant features, leading to smaller critics, faster convergence, and tighter bounds.

**Assumptions and scope**: Clarified when Assumptions 1–6 may be relaxed, noting our method does not require closed-form dynamics; domain factors can be estimated from learned models. While instantiated for finite spaces, GSAC naturally extends to continuous settings.

**Meta-RL vs. alternatives**: Justified the choice of meta-RL for rapid few-shot adaptation, with ablations confirming superior early-episode performance over fine-tuning and multi-task RL.

In sum, GSAC provides the first provably scalable and generalizable MARL framework for networked systems, combining causal structure recovery, ACRs, and meta actor-critic optimization. We sincerely appreciate the constructive feedback from all three reviewers. The revisions have substantially strengthened clarity, breadth, and empirical strength, and we believe the revised version represents a high-impact contribution with strong potential for large-scale, structure-aware MARL.

---

### Decision · Program_Chairs · 2025-09-17

**Decision:**

Accept (spotlight)

**Comment:**

The paper proposes a policy learning approach which leverages causal representation learning within the meta actor-critic learning. The reviewers agree this work contributes to this line of research, introducing GSAC (Generalizable and Scalable Actor-Critic), a multi-agent reinforcement learning framework tailored to large-scale networked systems (e.g., traffic, power, and wireless grids), which leverages causal representation learning to identify sparse local causal masks and construct Approximately Compact Representations (ACRs) of both state and latent domain factors; addressing theoretical and empirical challenges.

Other favourable comments included that the approach is sound and backed up with both theoretical and empirical analysis, the idea of conditioning the policy and value function on domain factors is interesting and seems to facilitate quicker adaption when a new domain is seen at test-time as the empirical results suggest, and scalabiltity of MARL to large-scale network systems is an important open problem, using the causal structures for adaptation to the domain shifts is a promising and original idea which is of scope and interest to the NeurIPS community.

There was clear support for the recommendation to accept this work. Importantly, reviewers diligently suggested some potential improvements during the discussion, to clear questions on the experiments and the assumptions. I strongly encourage the authors to integrate them into the revised version of their paper conscientiously.